# Global Emissions Inventory from Open Biomass Burning (GEIOBB): Utilizing Fengyun-3D global fire spot monitoring data

Yang Liu[1,2], Jie Chen[3,4], Yusheng Shi[1], Wei Zheng[3,4], Tianchan Shan[3,4], Gang Wang[5]

[1]State Key Laboratory of Remote Sensing Science, Aerospace Information Research Institute, Chinese Academy of Sciences, Beijing, 100101, China
[2]College of Resources and Environment, University of Chinese Academy of Sciences, Beijing, 101408, China
[3]Innovation Center for FengYun Meteorological Satellite, National Satellite Meteorological Center (National Center for Space Weather), China Meteorological Administration, Beijing 100081, China
[4]Key Laboratory of Radiometric Calibration and Validation for Environmental Satellites, National Satellite Meteorological Center (National Center for Space Weather), China Meteorological Administration, Beijing 100081, China
[5]Guangzhou Meteorological Satellite Ground Station, National Satellite Meteorological Center (National Center for Space Weather), China Meteorological Administration, Guangzhou 510640, China

*Correspondence to*: Yusheng Shi (shiys@aircas.ac.cn), Tianchan Shan (shantc9165@163.com)

**Abstract.** Open biomass burning (OBB) significantly affects regional and global air quality, the climate, and human health. The burning of forests, shrublands, grasslands, peatlands, and croplands influences OBB. A global emission inventory based on satellite fire detection enables an accurate estimation of OBB emissions. In this study, we developed a global high-resolution ($1 \times 1$ km) daily OBB emission inventory using the Chinese Fengyun-3D satellite's global fire spot monitoring data, satellite-derived biomass data, vegetation index-derived spatiotemporally variable combustion efficiencies, and land-type-based emission factors. The average annual estimated OBB emissions for 2020–2022 were 2,586.88 Tg C, 8841.45 Tg $CO_2$, 382.96 Tg CO, 15.83 Tg $CH_4$, 18.42 Tg $NO_X$, 4.07 Tg $SO_2$, 18.68 Tg OC, 3.77 Tg BC, 5.24 Tg $NH_3$, 15.85 Tg $NO_2$, 42.46 Tg $PM_{2.5}$ and 56.03 Tg $PM_{10}$. Specifically, taking carbon emissions as an example, the average annual estimated OBB for 2020–2022 were 72.71 (Boreal North America; BONA), 165.73 (Temperate North America, TENA), 34.11 (Central America; CEAM), 42.93 (Northern Hemisphere South America; NHSA), 520.55 (Southern Hemisphere South America; SHSA), 13.02 (Europe; EURO), 8.37 (Middle East; MIDE), 394.25 (Northern Hemisphere Africa; NHAF), 847.03 (Southern Hemisphere Africa; SHAF), 167.35 (Boreal Asia; BOAS), 27.93 (Central Asia; CEAS), 197.29 (Southeast Asia; SEAS), 13.20 (Equatorial Asia; EQAS), and 82.38 (Australia and New Zealand; AUST) Tg C/year. Overall, savanna grassland burning contributed the largest proportion of the annual total carbon emissions (1,209.12 Tg C/year; 46.74%), followed by woody savanna/shrubs (33.04%) and tropical forests (12.11%). SHAF was found to produce the most carbon emissions globally (847.04 Tg C/year), followed by SHSA (525.56 Tg C/year), NHAF (394.26 Tg C/year), and SEAS (197.30 Tg C/year). More specifically, savanna grassland burning was predominant in SHAF (55.00%, 465.86 Tg C/year), SHSA (43.39%, 225.86 Tg C/year), and NHAF (76.14%, 300.21 Tg C/year), while woody savanna/shrub fires were dominant in SEAS (51.48%, 101.57 Tg C/year). Furthermore, carbon emissions exhibited significant seasonal variability, peaking in September 2020, and August of 2021 and 2022, with an average of 441.32 Tg C/month, substantially higher than the monthly average of 215.57 Tg C/month. Our comprehensive

high-resolution inventory of OBB emissions provides valuable insights for enhancing the accuracy of air quality modeling, atmospheric transport and biogeochemical cycle studies. The GEIOBB dataset can be downloaded at http://figshare.com with the following identifier DOI: https://doi.org/10.6084/m9.figshare.24793623 (Liu et al., 2023).

## 1 Introduction

Open biomass burning (OBB) releases significant amounts of trace gases (CO, $NO_X$, NMVOC, $SO_2$, and $NH_3$), particulate
matter ($PM_{2.5}$, $PM_{10}$), and greenhouse gases ($CH_4$ and $CO_2$), which are major atmospheric pollutants (Mehmood et al., 2022a) and have profound impacts on the global carbon cycle, climate, and air quality, thus exerting a significant influence on the global environment and human health (Wu et al., 2022). The burning of forests, shrublands, grasslands, crop residues, and peatland constitutes the major types of fires worldwide (van der Werf et al., 2017). These open burning activities severely affect air quality and ecosystems (Anon, 2017), with high degrees of sporadicity and spatiotemporal clustering (Liu et al., 2014;
Murdiyarso and Lebel, 2007; Senande-Rivera et al., 2022). In addition, some regions worldwide are experiencing a notable increase in fire incidents (Kolden et al., 2024; Richardson et al., 2022) such as the Amazon rainforest (Pivello, 2011), Australian bush (Jegasothy et al., 2023), and the United States (You and Xu, 2023), where large-scale fire incidents occur periodically and frequently (Kolden et al., 2024). Therefore, accurately estimating these emissions is crucial for devising effective environmental policies and safeguarding human health and quality of life, thereby providing significant support for
a sustainable future.

Previous studies have investigated numerous methods for estimating biomass burning emissions (Ito and Penner, 2004; Wiedinmyer et al., 2006). The burned-area-based fire emission estimation method, which is based on the burned area, available biomass fuels burned in the fields, fuel-related combustion efficiency, and emission factors, has demonstrated good accuracy in quantifying larger fire events. This method has been widely used in databases such as the Global Fire Emissions Database
(GFED) (van der Werf et al., 2017) and the Fire INventory from NCAR (FINN) (Wiedinmyer et al., 2023). However, this method relies heavily on the fire-detection precision, particularly for small fires. A method based on fire radiative power (FRP) can enhance the detection and quantification of small fire events by measuring the energy released during combustion (Filizzola et al., 2023). However, these approaches can overestimate emissions from localized fire events, which are intense, small-scale fires that may not reflect wider fire activity (Nguyen et al., 2023). For example, Fire Emissions and Energy Research (FEER),
based on FRP, reported that the global total particulate matter emissions were approximately 55% higher than those estimated by the GFED (Ichoku and Ellison, 2014). Similarly, the Global Fire Assimilation System (GFAS) using FRP estimated global and regional combustion values exceeding those of the GFED by approximately 126 Tg C/year during 2003-2008 (Kaiser et al., 2012). However, all these methods rely on MODIS active fire products.

Similar to the MERSI-2 instrument, the Fengyun-3D (FY-3D) satellite has spatial resolutions of 250 m (0.47–0.86 μm and
10.80–12.02 μm) and 1000 m (1.38–8.55 μm) at the nadir (Yin et al., 2020), which is more advantageous in detecting and monitoring various active fire events compared with MODIS (Zheng et al., 2023). Furthermore, the Global Fire Monitoring

(GFR) product with FY-3D employs optimized automatic identification algorithms for fire spots (Shan and Zheng, 2022), leading to improved fire point detection accuracy. Thus, it has an overall accuracy rate of 79.43% and exclusion omission error accuracy of 88.50%, surpassing the capabilities of MODIS satellite products (Chen et al., 2022; Xian et al., 2021), based on field-collected references from China throughout 2020. Cross-verification between MODIS and FY-3D showed the highest consistency (over 80%) in Africa and Asia, whereas the consistency in America, Europe, and Oceania exceeded 70% (Chen et al., 2022). The number of fire spots in July, August, and September was higher, with a mean consistency of over 85% between MODIS and FY-3D fire products (Chen et al., 2022). Although the Landsat Fire and Thermal Anomaly (LFTA) product has a finer spatial resolution, its lower temporal resolution limits its global coverage to only 16 days; thus, large numbers of fires with short durations are missed. The shorter revisit time of FY-3D allows for monitoring more fires lasting for one day, which are expected to yield reliable estimates of OBB emissions.

Fuel loading (F) represents the ground biomass of the fire-affected pixels. Many studies have adopted a static approach to F (Chang and Song, 2010; Puliafito et al., 2020; Shi et al., 2020; Zhou et al., 2017), assigning constant values based on regional land-cover types. This methodology overlooks the inherent spatial and temporal variability of F within each land type, which changes continuously and dynamically (Wiedinmyer et al., 2011). The combustion factor (CF), which denotes the ratio of consumed fuel to total available fuels, is typically a linear variable within a specific range when considering the fuel status and humidity conditions (van der Werf et al., 2006; Wiedinmyer et al., 2011). However, this approach of calculating CF leads to increased uncertainty in biomass estimation and poor quantification of the extent of combustion during fire events, thereby affecting OBB emissions assessment (Shi et al., 2020). To address these issues, this study employed observational and satellite-based aboveground biomass (AGB) and CF based on time-series vegetation index data derived from satellite products. The CF considers moisture-related factors, enabling the calculation of the spatiotemporal variance in combustion efficiency across diverse land types.

This study aimed to develop a high-resolution daily OBB emissions inventory (including carbon (C), carbon dioxide ($CO_2$), carbon monoxide (CO), methane ($CH_4$), nitrogen oxides ($NO_X$), sulfur dioxide ($SO_2$), particulate organic carbon (OC), particulate black carbon (BC), ammonia ($NH_3$), nitrogen dioxide ($NO_2$), $PM_{2.5}$, and $PM_{10}$) and analyze the various types of fire events along with their emission patterns across 14 distinct regions. To estimate the OBB emissions from forests, savannas/shrublands, grasslands, and peatlands, we utilized the updated FY-3D GFR product based on the continuous spatiotemporal dynamics of AGB, spatially and temporally variable combustion efficiencies, and emission factors specific to different land types. Our comprehensive high-resolution inventory of OBB emissions represents a valuable asset for applications in air quality modeling, atmospheric transport simulations, and biogeochemical cycling studies. This provides a robust framework for in-depth understanding and analysis of the environmental implications of OBB on a global scale.

## 2 Materials and Methods

The Global Emissions Inventory from Open Biomass Burning (GEIOBB) (1 km daily) was estimated using the burned area method based on the framework described by Wiedinmyer et al. (2006) and Shi et al. (2015). GEIOBB includes OBB emissions based on burned areas retrieved from active fire data from the FY-3D satellite, available biomass from satellite and ground measurements, CF scaled by tree cover (TC) and the Normalized Difference Vegetation Index (NDVI), and land cover (LC)-based emission factors. The GEIOBB is obtained by calculating the product of the above terms.

$$E_i(x) = B(x,t) \times F(x) \times CF(x) \times EF(i), \tag{1}$$

where $E_i$ ($g$) represents pollutant type $i$ emissions at location $x$, which is equal to the product of burning area $B$ ($m^2$) at time $t$ and location $x$, biomass $F$ ($kg/m^2$) at location $x$, $CF$ (expressed as a *fraction*), and the emission factor $EF$ ($g/kg$) for pollutant type $i$.

### 2.1 FY-3D global fire spot monitoring data based burned area (B)

The Fengyun-3 series of satellites is a second-generation Chinese polar-orbiting meteorological satellite system. The FY-3D satellite was the fourth in the FY-3 series. It was launched on November 15, 2017, at an altitude of 836 km, and the data became accessible in May 2020 (Li et al., 2017). FY-3D completes 14 orbital observations of the Earth's surface on a global scale twice daily. The MERSI-2 instrument onboard FY-3D was greatly improved from the MERSI-1 instrument onboard FY-3C, with high onboard accuracy and lunar calibration capabilities. Compared with MODIS, FY-3D fire products have been optimized in terms of auxiliary parameters, fire identification, and re-identification. First, FY-3D introduces an adaptive threshold using automatic identification algorithms for fire spot detection, which calculates the background temperature as the mean temperature of all the background pixels within each 3×3 window. If fewer than 20% of the pixels are identified as cloudless, the window size is expanded to 5×5, continuing up to 51×51 in order to accommodate more data (Chen et al., 2022). This approach eliminates the limitations in the MODIS and VIIRS algorithms, which set T4 to greater than fixed 360 K (320 K at night) and the variable moving window size to a maximum of 21×21 (Giglio et al., 2016). Second, FY-3D uses a re-identification index that reflects varying geographical latitudes and underlying surface types, together with the effects of clouds, water, and bare land (Zheng et al., 2020). Based on the initially identified fire spots, FY-3D employed the re-identification index to further remove false fire spots at cloud edges, water body edges and other high-reflection underlying surfaces (Chen et al., 2022). The integration of multiple influencing factors increases the fire detection accuracy. For example, the influences of factory thermal anomalies and high reflectance of photovoltaic power plants are removed. Finally, FY-3D employs a far-infrared band with a high resolution of 250 m, which has a higher resolution than MODIS (1 km) (Zheng et al., 2023). The far-infrared band has a higher sensitivity to large fires or high-brightness fire events and can distinguish differences against background brightness temperatures (Zheng and Chen, 2020). These characteristics are essential for the accurate identification of fire spots, thereby enhancing the fire detection precision of satellites (Chen et al., 2022). Overall, the FY-3D GFR product has an accuracy of 94.01% globally, as calculated using fire detection after eliminating errors based on visual checks conducted

using SMART (Visual Check) in 2019. It has accuracies of 94.61, 94.12, 90.63, 91.76, and 92.69% for Southern Central Africa, Eastern Central South America, Siberia, Australia, and the Indo-Chinese Peninsula, respectively (Chen et al., 2022). Specifically, owing to the removal of the underlying surface interference in China, FY-3D has accuracies of 79.43% and 88.50% for accuracy (omitted fire) and accuracy without omission (misidentified fire) (Chen et al., 2022). These accuracies were determined by comparing the results of a large-scale field experiment conducted jointly by the State Grid Corporation of China and China Meteorological Administration with the GFR product, including omitted and misidentified fire (Chen et al., 2022). This comprehensive assessment took place throughout 2020 across five provinces in China—Guangdong, Guangxi, Yunnan, Guizhou, and Hainan—utilizing a combination of real-time satellite data and ground-truth validation to evaluate the suitability of these fire detection products. These accuracies are significantly higher than those achieved by MODIS, which are 74.23 and 79.69%, respectively (Chen et al., 2022).

The location, timing and burned area of the fire events used in the GEIOBB were determined globally using the FY-3D GFR product (Chen et al., 2022). Processed fire event detection data Fengyun Satellite Remote Sensing Data Service Network of National Satellite Meteorological Centre (http://satellite.nsmc.org.cn/PortalSite/Default.aspx), which estimated the actual area of fire spots based on radiation in different infrared channels. When the mid-infrared channel was not saturated, it was used to estimate the sub-pixel fire spot area and temperature. Otherwise, a far-infrared channel was employed for the estimation (Zheng and Chen, 2020). These data offer daily fire detection at a 1-km resolution, including the location, time, burned area, and confidence level (Liu and Shi, 2023). Furthermore, multiple counts of the same fire may have been recorded on a single day, leading to data duplication. To address this issue, we performed a global identification and removed multiple daily detections of the same fire pixels and data with confidence levels below 20%. Specifically, we removed single daily fire detections within a 1-km radius of another fire detection. Thus, only one fire per 1 km$^2$ of a hotspot could be counted per day and was reset on the next day (Wiedinmyer et al., 2023).

**2.2 Fuel loading (F)**

Previous studies based on burned areas have distinguished F by categorizing it according to regions of different fire types (Wiedinmyer et al., 2011). The data generated by this method have some discontinuities, which may lead to large deviations at the boundaries of different areas; this is unreasonable and does not reflect the spatial distribution pattern of F. Ground observation data are more accurate and reliable, but are limited by the sparse distribution of observation stations, preventing comprehensive global coverage. In contrast, satellite data cover the entire globe and provide worldwide surface parameters, thereby enabling biomass estimation. However, their accuracy and usability are limited by factors such as their temporal and spatial resolutions and cloud cover. Therefore, combining ground observations with satellite data is an effective solution. This fusion method combines the high accuracy of ground observation data with the wide coverage of satellite data to generate global biomass products. Using this method, it is possible to overcome the limitations of using a single data source, thereby enhancing the accuracy of biomass estimations.

This study used multi-source data, including NDVI, TC, and AGB, to assess the terrestrial biomass. NDVI data were obtained using the MODIS Combined 16-Day NDVI fusion product available on the Google Earth Engine platform. AGB shows a strong linear correlation with TC and NDVI (Yao et al., 2017). The TC data were derived from the MOD44B product (DiMiceli et al., 2022) generated based on MODIS onboard the Terra satellite (https://lpdaac.usgs.gov/products/mod44bv061/), which provides a continuous global vegetation field at 250m resolution for each year from 2000 to the present. AGB data were obtained from the Global Aboveground and Belowground Biomass Carbon Density Maps for the Year 2010 product (https://daac.ornl.gov/cgi-bin/dsviewer.pl?ds_id=1763) provided by Spawn and Gibbs (2020). This dataset uses thousands of satellite data points and ground measurements to produce a biomass map with a 1-km resolution (Spawn and Gibbs, 2020). A combination of 2118 other ground measurements and Lidar data to validate observations, and showed that the fused map had a root mean-square error (RMSE) that was 15–21% lower than those reported by Saatchi et al. (2011) and Baccini et al. (2012). We used the AGB for 2010, annual TC, and NDVI data, and linearly stretched the fuel loading for other years.

$$F(x,t) = \left( \frac{NDVI_{now} + TC_{now}}{NDVI_{2010} + TC_{2010}} \right) * AGB \tag{2}$$

where $NDVI_{now}$ is the mean value of the month before a single fire event, $NDVI_{2010}$ is the mean value of $NDVI$ in 2010, $TC_{now}$ is the tree cover in the year of the fire incident, $TC_{2010}$ is the tree cover in 2010, and $AGB$ is the aboveground biomass in 2010.

## 2.3 Combustion factor (CF)

The CF is mainly defined as the percentage of fuel consumed during individual fire events, which primarily depends on the type of fuel and humidity. Typically, the CF is set as a linear variable within a specific range, which may lead to biases in emission estimations and generate significant uncertainties. Although some studies used TC to quantify CF and explain its spatial and temporal variations (Bray et al., 2018; Qiu et al., 2016; Wiedinmyer et al., 2006; Wu et al., 2018), previous research has mainly focused on areas with herbaceous vegetation cover, where the TC ranges from 40% to 60%. They assumed that the CF remained consistent across other land types, such as farmlands, forests, and grasslands. The fire type at the location of the fire event has a major influence on OBB. We used International Geosphere-Biosphere Programme (IGBP)-categorized data from the MODIS land cover type (LCT) information (Friedl and Sulla-Menashe, 2022) (MCD12Q1, https://lpdaac.usgs.gov/products/mcd12q1v061/). We reclassified the original 17 classifications into 7 categories to better differentiate fire types; grasslands and savannas (V1), woody savannas or shrubs (V2), tropical forests (V3), temperate forests (V4), boreal forests (V5), temperate evergreen forests (V6), and crops (V7); this was to allow for better matching in the calculation and subsequent analysis processes. In the GEIOBB, the CF of all fires in each grid cell was allocated as a function of TC, fire type, and NDVI (Ito and Penner, 2004). We segmented the reclassification results into 4 categories to calculate the CF. Specifically, we amalgamated the reclassification outcomes of V3, V4, V5, and V6 into forest types, designated V1 as grassland, V2 as woodland, and V7 as cropland (the specific classification method is detailed in Supplementary Information (SI) Table S1).

For woodland fires, CF is highly correlated with $TC$ (Ito and Penner, 2004):

$$CF_{woodland} = EXP(-0.013 \times TC). \qquad (3)$$

For grassland fires, a change in the NDVI is usually associated with the occurrence of fires, especially in dry seasons or in areas prone to wildfires. Generally, a decrease in NDVI may indicate deteriorating vegetation health, which increases the risk of fires because dry or withered vegetation is more prone to burning. We introduced the vegetation condition index ($VCI$) to determine the fuel moisture conditions, which were used to measure the vegetation drought conditions by calculating contemporaneous changes in NDVI as a metric for assessing the contemporaneous conditions of vegetation. We supplemented our research based on Ito and Penner (2004) by replacing the percentage of green grass from the total grass with the $VCI$, which was computed using the $NDVI$ with a time interval of 16 d at a spatial resolution of 1 km for the period of 2020–2022. In addition, we introduced a compensatory term to mitigate the impact of tree cover on grassland fires.

$$VCI = \frac{NDVI_{now} - NDVI_{min}}{NDVI_{max} - NDVI_{min}}, \qquad (4)$$

$$CF_{grassland} = (0.9 - TC) \times (-2.13 \times VCI + 1.38) + TC. \qquad (5)$$

where $NDVI_{now}$ is the mean value of the month before a single fire event, $NDVI_{max}$ is the maximum value of $NDVI$ for the same period in the previous three years of the fire event, and $NDVI_{min}$ is the minimum value of $NDVI$ for the same period in the previous three years of the fire event.

For forest fires, we used moisture category factors ($MCF$) to measure forest moisture and conducted an analysis based on the partitioning of MCF values (very dry: 0.33, dry: 0.5, moderate: 1, moist: 2, wet: 2, and very wet: 5) provided by Anderson et al. (2004). We used the VCI as a criterion for assessing wetness and dryness and discovered that it approximately conformed to the power function distribution characteristics of $VCI$. Subsequently, a power function fitting was performed ($R^2 = 0.94$), through which we determined the $CF$.

$$MCF = 0.1759 \times e^{3.5181 \times VCI}, \qquad (6)$$

$$CF_{forest} = (1 - e^{-1})^{MCF}. \qquad (7)$$

Most fires in croplands are artificially active, resulting in full combustion processes that are not designed for woody fuels. Therefore, we set the $CF$ for crops to 0.98, which is the upper limit proposed by Wiedinmyer (2006).

## 2.4 Emission factor (EF)

EFs are used to convert dry matter burned into trace gas and aerosol emissions, which denotes the number of pollutants released per unit of fuel burned. The measurements of EFs in different regions for grasslands and savannas, woody savannas or shrubs, tropical forests, temperate forests, temperate evergreen forests, and crops were reviewed and tabulated by Akagi et al. (2011), whereas those for boreal forest fires were obtained from the averages reported by Akagi et al. (2011) and Urbanski (2014). The EFs for maize, sugar, and rice crop fires were taken from the averages reported by Akagi et al. (2011), Fang et al. (2017), Liu et al. (2016), Santiago-De La Rosa et al. (2018), and Stockwell et al. (2015). The BC EFs of BC for crop fires were sourced from Kanabkaew and Kim Oanh (2011) and those for wheat fires were obtained from Cao et al. (2008). In addition, the

emission factors of NO$_2$, PM$_{2.5}$, and PM$_{10}$ for the crop fire were derived from Li et al. (2007), and the EF from the crop was the average of maize, sugar, rice, and wheat. The EFs values are presented in Table 1.

## 3 Results and Discussions

### 3.1 Spatial map of OBB emission estimates

We estimated global OBB emissions using GEIOBB, and the average annual values for 2020–2022 were 2586.88 Tg C, 3.77 Tg BC, 15.83 Tg CH$_4$, 382.96 Tg CO, 8841.45 Tg CO$_2$, 5.24 Tg NH$_3$, 15.85 Tg NO$_2$, 18.42 Tg NO$_X$, 18.68 Tg OC, 56.03 Tg PM10, 42.46 Tg PM2.5, and 4.07 Tg SO$_2$ (Table 2). Taking carbon as an example, the annual carbon emissions from the OBB were estimated for the period 2020–2022 (Figure 1), and the total OBB emissions reached 7760.63 Tg C. The average annual carbon emissions during this period were 2586.88 Tg. Overall, clear spatial variations in the OBB carbon emissions were observed across Africa and certain regions of the Americas and Asia. In Central and Southern America, elevated emissions were observed in central and northeastern Brazil, northern Bolivia, northern Paraguay, eastern Mexico, and Honduras. In Africa, substantial OBB emissions originate from Central Africa (excluding the Democratic Republic of the Congo), the northern regions of West Africa, and the southern regions of East Africa, where most $1 \times 1$ km grid cells exhibit annual average carbon emissions exceeding 50 g C/m². Elevated carbon emissions were observed in Southeast Asia (Indo-Chinese Peninsula), with significant emissions detected in western and eastern Myanmar, northern Laos, eastern Cambodia, southern Nepal, and parts of northern India. Notable carbon emissions were also observed in equatorial Asia, South Sumatra, South Kalimantan, and southern Papua New Guinea.

We divided the world into 14 regions for analysis and discussion; the geographical regions were the same as those used by van der Werf et al. (2017) (Figure 2(a)). As delineated by the reclassification in Figure 2(b), savanna grasslands emerged as the predominant LCT worldwide, encompassing 53.30% of the total area. This type primarily occurs in South America, Africa, and Asia. Following closely is woody savanna accounting for 19.74% of the global coverage. They are predominantly situated in Boreal Asia, Australia, selected areas of southern Africa, and parts of North America. The third most prevalent type was tropical forest, comprising 9.03% of the total area, mainly distributed in South America, particularly within the Amazon Rainforest, regions adjacent to the African equator, and Southeast Asia. Other LCTs, such as temperate forest, boreal forest, temperate evergreen forest, and crops, are less extensively spread and exhibit a more dispersed distribution.

This study then quantified the estimated global average annual OBB carbon emissions from different regions and fire types during 2020–2022 (Table 3). Southern Hemisphere Africa (SHAF) was found to be the primary source of global OBB carbon emissions (847.04 Tg; 32.74%); this trend also held true for other pollutants. Southern Hemisphere South America (SHSA) and Northern Hemisphere Africa (NHAF) ranked second and third, accounting for 20.12% (520.55 Tg) and 15.24% (394.26 Tg), respectively. The contributions of each fire type to the global OBB carbon emissions were then quantified. Savanna grasslands were the largest contributor (1209.12 Tg, 46.74%), followed by woody savanna/shrubs (854.71 Tg, 33.04%), tropical forest (313.32 Tg, 12.11%), temperate forest (92.65 Tg, 3.58%), crop (58.06 Tg, 2.24%), temperate evergreen forest

(41.65 Tg, 1.61%), and boreal forest (17.37Tg, 0.67%). According to GFED4.1s, the annual average carbon emissions from wildfires in SHAF, SHSA, and NHAF during 2020–2022 were 1271.63 Tg/year, accounting for approximately 64.55% of the global total OBB carbon emissions. Their research findings are similar to the results of this study, which recorded 1761.84 Tg, equivalent to 68.10% of the total.

Specifically, the contributions of the seven fire types to OBB carbon emissions varied dramatically across continents (van der Werf et al., 2010). In SHAF, the primary sources of OBB were savanna grasslands and woody savanna or shrubs, contributing 465.85 (54.99%) and 324.08 Tg/year (38.26%), respectively, consistent with Nguyen et al. (2023). Unlike SHAF, OBB in SHSA primarily originated from savanna grasslands and tropical forests (Shi et al., 2015), contributing 225.86 (43.38%) and 177.17 Tg/year (34.03%) to the region's carbon emissions, respectively. This variation could be associated with the ecological

and climatic conditions unique to each region (Sahu and Sheel, 2014; Santana et al., 2016). South America hosts the world's largest rainforests and is known for its rich biodiversity and biomass (Fagua and Ramsey, 2019). However, they are severely threatened human-induced deforestation and forest fires (Chen et al., 2013). Studies indicate that forest fires and human activities, such as deforestation and land-use changes, are the main drivers of increased carbon emissions from OBB in this region (Cochrane and Laurance, 2002; Nepstad et al., 1999). In the NHAF, the predominant source of OBB was savanna

grasslands (Roberts et al., 2009), contributing 76.14% to the region's total biomass-burning carbon emissions, averaging 300.21 Tg/year. This may be related to the arid climate and low forest cover in the region (De Sales et al., 2016; Ichoku et al., 2016). Previous research has shown that climate change and human activities, such as grazing and agricultural expansion, are the major factors in this region (Flannigan et al., 2009; Scholes and Andreae, 2000).

Fire events in savanna grasslands remain a major source for most pollutants generated by global OBB, whereas crops contribute

relatively less (Figure 3). However, with respect to BC and $NH_3$, fire events in woody savanna/shrubs have become the primary contributors (BC, 59.40%; NH3, 39.33%). Furthermore, when considering the different regions, the primary sources of pollutants from OBB vary. For instance, fire events in woody savanna/shrubs were the primary sources in the BONA, SEAS, and EQAS regions, whereas crop-related fire events mainly occurred in the EURO, MIDE, CEAS, and SEAS regions.

### 3.2 Temporal variations in OBB carbon emissions

The monthly carbon emissions at both the global and regional levels are illustrated in Figure 4. Overall, global OBB carbon emissions experienced notable shifts, with considerable monthly variations from 2020 to 2022, and peak emissions were observed in August 2021 (729.37 Tg). Global OBB carbon emissions were 2,861.05 Tg in 2020, rising slightly to 2,991.15 Tg in 2021, but showing a significant decline to 1,908.41 Tg in 2022. Monthly and seasonal variations in the OBB carbon emissions from each region exhibited substantial differences. Of the 14 regions, the annual contribution of SHAF, the largest

global contributor of OBB carbon emissions (32.74%), increased by 2.70% per year, with the peak emission of 283.59 Tg occurring in August 2021. SHAF has emerged as a primary contributor to global OBB carbon emissions owing to its substantial biomass and escalating human activities. Abundant biomass, including dense vegetation and rich forest resources, provides ample fuel for carbon emissions that are exacerbated by intensifying human activities (Chen et al., 2017). In August, specific

meteorological conditions, such as high temperatures and low humidity facilitated the increased combustibility of biomass,
resulting in a peak in carbon emissions (Shea et al., 1996). Although the SHAF region consistently remained the largest
contributor to global OBB carbon emissions during 2020–2022, its annual emissions remained relatively stable, with minor
fluctuations. Conversely, emissions from SHSA decreased at a rate of 105.22 Tg per year from 2020 to 2022, with peak
monthly emissions over the 3 years reaching 184.63, 222.12, and 123.98, respectively, size and status of emissions consistent
with Griffin et al. (2023). Annual C emissions in NHAF also declined, decreasing by 55.44 Tg over the 3 years, with its
emissions accounting for the lowest percentage at 13.76% in 2021.

Cumulatively, SHAF, SHSA, and NHAF represent almost 70% of the global OBB carbon emissions, a testament to the
profound intertwining of their native ecosystems, land utilization, and climatic influences on biomass combustion (Roy et al.,
2022). Deeper exploration revealed that the SHAF, which is endowed with vast stretches of savannahs and grasslands,
undergoes intermittent dry periods (Hoffmann and Jackson, 2000). This climatic pattern, combined with entrenched
agricultural customs like slash-and-burn, renders the region prone to wildfires (Lourenco et al., 2022). In the SHSA, which
covers significant portions of the Amazon rainforest, rampant deforestation often involves controlled burning (Kröger and
Nygren, 2020). Unfortunately, these sometimes escalate beyond the control level, adding substantially to emissions figures
(Eufemia et al., 2022). In contrast, the NHAF's shifting land-use paradigms, coupled with increasingly recurrent droughts—
potentially a byproduct of global warming—intensify frequency of fires in the area (Machete and Dintwe, 2023).

Examination of monthly emissions data revealed significant regional disparities. For example, every January, the NHAF,
influenced by its monsoon cycles (Martin and Thorncroft, 2014), consistently emerges as the primary contributor to biomass
carbon emissions, accounting for contributions of 50.74, 81.16, and 67.66% across the 3 years, as reported by Tsivlidou et al.
(2022). By March, SEAS witnessed a surge in emissions, largely due to shifts in forestry practices (Shi et al., 2014), with
contributions escalating to 50.82, 57.78, and 40.67% in subsequent years (Pletcher et al., 2022), respectively. The peak biomass
carbon emissions in 2020 occurred in September, reaching 500.62 Tg. However, the peaks in 2021 and 2022 appeared sooner
in August, with emissions of 729.37 and 357.57 Tg, respectively. The 2021 ascent of BONA emissions might be linked to
altered land-use guidelines or increased farming activities (Zerriffi et al., 2023) and the many wildfires that occurred (Hoffman
et al., 2022), while California's heightened investment in fire mitigation programs (Umunnakwe et al., 2022) and the U.S.
Forest Service's implementation of a decade-long strategy (Confronting the Wildfire Crisis, 2023) in 2022 have effectively
curbed wildfire incidents in the TENA region. This shift in the perception of forest fire management has been instrumental in
mitigating wildfire risk in the area. Nevertheless, it is important to acknowledge that the occurrence of wildfires varies over
time (Bowman et al., 2017).

Figure 5 shows the notable temporal fluctuations in global wildfire carbon emissions for different fire types throughout the
study period from 2020 to 2022. Global combustion exhibited the highest carbon emissions in August and September. In
September 2020, single-month emissions peaked at 500.62 Tg C. However, in 2021 and 2022, the zenith of carbon emissions
from fires occurred in August, registering at 729.37 and 357.57 Tg respectively. The smaller peaks observed in March should

not be overlooked. Interestingly, although the timing of these emission peaks varied, their main contributing factors remained similar. In September, the daily carbon emission peaks from savanna grasslands, woody savanna/shrubs, and tropical forest regions were 7.54 (38%), 7.12 (37%), and 3.36 (31%) Tg C/day, respectively. These sources constituted the primary contributors to the global biomass combustion carbon emissions from July to October.

Spatial and temporal variations in global OBB emissions are pronounced because of the differences in ecosystems, climatic conditions, and human activities across different regions (Moritz et al., 2012; Ward et al., 2018). For instance, areas with expansive tropical grasslands, such as Sub-Saharan Africa and Australia, typically experience high levels of OBB emissions because of the prevalence of both natural and anthropogenic fire activities (Williams et al., 2019; Zheng et al., 2021a). Moreover, many regions undergo cyclical OBB emission patterns, coinciding with the onset of the dry and wet seasons (Dury et al., 2011; Gautam et al., 2013). The dry season, characterized by an increase in dry biomass and conducive weather conditions, often witnesses a surge in fire activity, resulting in elevated emission levels (Zhang et al., 2023b). These considerable spatial and temporal fluctuations in global OBB emissions mirror the diversity of ecosystems and climatic conditions across various geographic locations (Fagre et al., 2003), which are further influenced by human endeavours and natural fire regimes (Jones et al., 2022).

In 2020 and 2021, significant wildfire events, such as the California wildfires and Australian forest fires, led to an escalation in carbon emissions from fires (Collins et al., 2021, 2022; Gallagher et al., 2021; Keeley and Syphard, 2021; Safford et al., 2022). However, a dual phenomenon was observed in 2022. The implementation of robust wildfire control measures contributed to a reduction in emissions (Wollstein et al., 2022); however, an overall augmentation in annual precipitation led to a reduction in the degree of drought (Thackeray et al., 2022; Zhang et al., 2023a). Consequently, the annual OBB carbon emissions in 2022 were lower than those in the preceding years.

Specifically, carbon emissions resulting from fire events were analysed in 14 global subregions from 2020 to 2022 (Figure 6). This analysis revealed the primary sources of carbon emissions from fires worldwide and provided insights into the main constituents of combustion in different regions. Emission patterns across different global regions vary both temporally and spatially. The top three major emitting regions were SHAF, SHSA, and NHAF, which were closely associated with global emission trends, representing the main source of the emission peak in August and the emission during the winter months. During 2020 to 2022, the OBB conditions in the SHAF, SHSA, and NHAF regions have been relatively stable, with daily peak values of 12.04 Tg, 9.81 Tg and 4.38 Tg respectively. For the SHAF and SHSA, burning activities were predominantly observed from July to September, which can be attributed to a combination of dry weather, strong winds, and specific meteorological conditions (Eames et al., 2023; Li et al., 2023). These factors collectively enhanced the combustibility of the biomass during this period, leading to an increased likelihood of burning. In the SHAF, emissions were primarily influenced by savanna grasslands (49%) and woody savanna/shrubs (47%). Similarly, in the SHSA, emissions were mainly affected by savanna grasslands (34%) and tropical forests (38%). While burning in the NHAF region is concentrated between November and January, primarily in January, this pattern is significantly influenced by the practice of slash-and-burn agriculture (Serrani et al., 2022), with savanna grasslands accounting for 77% of the contributing factors.

CEAM and SEAS exhibited similar wildfire patterns, primarily occurring in March, and a noticeable decrease in burning activity emissions from 2020 to 2022. The predominant fire type in the CEAM region was woody savanna/shrubs (50%), whereas in the SEAS region, it was mainly influenced by woody savanna/shrubs (50%) and tropical forest (25%). Overall, owing to similarities in factors, such as biomass fuel load and climate, the wildfire types in the CEAM and SEAS were quite alike.

The BONA, TENA, EURO, MIDE, BOAS, and AUST share a common characteristic: OBB carbon emissions exhibit a high degree of randomness, indicating their primary influence on natural wildfire events. For instance, British Columbia, Canada, experienced a series of wildfires in July 2021 (Copes-Gerbitz et al., 2022), leading to peak carbon emissions for BONA in 2021 (4.46 Tg). TENA, affected by a series of wildfires in the western United States in 2020 (Safford et al., 2022) and the ongoing wildfires in California in 2021 (Varga et al., 2022), showed elevated emissions in both years (2020, 6.12 Tg; 2021, 3.76 Tg), with woody savanna/shrubs being the main fire event type. For the EURO, the apex of wildfires in 2021 was distinctly shaped by wildfires in Southern and Southeastern Europe (Tedim et al., 2022). The emissions were predominantly associated with fire type savanna grassland (48%). Moreover, in the BOAS region, wildfires were influenced by forest fires in Siberia (Ponomarev et al., 2022), where the principal fire type was woody savanna/shrubs (31%). Regarding AUST, in January 2020, a significant forest fire event occurred (Storey et al., 2023), resulting in peak emission of 4.48 Tg. The primary fire types were temperate forest (24%) and savanna grassland (18%).

The situation of OBB in CEAS is intricate. In March, substantial OBB emissions resulted from agricultural practices, such as slash and burn cultivation and the burning of crop residues (Liu and Shi, 2023), with crops being the predominant fire event type (30%). In contrast, from August to November, OBB was mainly attributed to scorching weather and monsoon conditions (Shi et al., 2018), with savanna grasslands being the dominant type (28%). Recently, owing to improvements in agricultural management practices, there has been a noticeable decrease in OBB events of crop types.

### 3.3 Cross-verification in different database

In this study, we juxtaposed the global distribution of OBB carbon emissions as estimated in GEIOBB with data published in the GFAS, GFED, and FEER datasets for 2020–2022 (Figure 7). Overall, our assessments corresponded well with the GFAS, GFED, and FEER, although there was an overestimation in high-latitude regions, the overall differences across large regions were minimal. For instance, we estimated the total carbon emissions in the BONA region to be 72.71 Tg, while the values from GFAS, GFED, and FEER were 61.21, 125.05, and 35.83 Tg, respectively. This variance can be attributed to the different resolutions (1 km×1 km, 0.1°×0.1°, 0.25°×0.25°, and 0.1°×0.1°) and different estimation methodologies employed. Both our study and the GFED adopted an estimation approach based on the burned area, whereas the GFAS and FEER formulated their inventories based on fire radiative energy. Consequently, our inventory yielded accurate assessment results and captured the spatial variation and heterogeneity of minor OBB emissions effectively, which could have been overlooked in coarse-scale analyses. Additionally, the GFED utilizes MODIS satellite data to calculate the available biomass fuel, whereas we leverage the higher precision and small fire quantification capability of FY-3D GFR data. Disparities between different satellite data

and variations in parameter definitions during inventory formulation contribute to these differences. Moreover, we adopted published local measurement-based emission factors and improved correlation coefficients for estimating OBB carbon emissions, which are more reliable and significantly enhance the local emission estimation accuracy.

Specifically, in high emission regions (Figure 8), such as NHAF, NHSA, and CEAS, our estimation of OBB carbon emissions (multi-year average 394.25, 42.93, and 27.93 Tg; monthly peak average 102.52, 11.86, and 6.24 Tg) aligned closely with those of GFED (multi-year average 342.31, 29.10, and 38.16 Tg; monthly peak average 97.58, 9.86, and 10.91 Tg) and GFAS (multi-year average 288.81, 35.80, and 43.51 Tg; monthly peak average 70.65, 9.64, and 9.82 Tg). However, discrepancies were observed between MIDE and EQAS, with FINN notably overestimating carbon emissions from fires. This overestimation by FINN is attributed to its methodology (Wiedinmyer et al., 2011), which relies on a combination of emission factors, conversion rates, and fire radiative energy values to estimate the emissions from agricultural residue burning. This contrasts with our approach, which bases estimates on the burned area and thus can accurately quantify carbon emissions from large fires and reduce uncertainty in fire data (Shi et al., 2020). Additionally, emission estimates during the periods by FINN, GFED, and GFAS were generated using data from the Terra and Aqua satellites, which captured data at 10:30 and 13:30 LT. However, the use of FY-3D, which captures data at 14:00, proved highly effective in capturing such events. Furthermore, fire incidents tend to peak in the afternoon (Mehmood et al., 2022b), with agricultural waste and crop residue burning more frequently occurring during this period due to higher temperatures that enhance burning efficiency (Jurdao et al., 2012). While, the average annual estimated OBB emissions exceed those reported by GFED by 617.14 Tg C/year. These discrepancies are probably related to small-scale fire events. For instance, the largest difference is observed in the SHAF region, exceeding by 248.01 Tg C/year, followed by SHSA (190.28 Tg C/year) and SEAS (103.92 Tg C/year). In the SHAF region, compared to MODIS active fire, FY-3D GFR detects more small fire points (Figure S2, Figure S3 (a), Figure S3 (b)), which are isolated within 5-kilometer resolution pixels. However, in this area, the majority of fire events are large-scale incidents, which means that although small fires are more numerous, they contribute minimally to the total emissions. Furthermore, fire events in SHSA (Figure S3 (c), Figure S3 (d)) and SEAS (Figure S3 (e), Figure S3 (f)) are primarily triggered by human activities, consisting of small-scale incidents that are significantly linked to the overall emissions. In contrast, areas frequently affected by large-scale fire events show relatively smaller discrepancies, such as TENA (99.05 Tg C/year), NHAF (51.94 Tg C/year), and other regions including NHSA, AUST, CEAM, MIDE, EURO, and EQAS (all under 15.00 Tg C/year).

The AGB values used in this study were directly derived from a dataset generated by combining field and satellite observations (Avitabile et al., 2016). GFED, calculates this value through simulations using the biogeochemical CASA model. While GFED has adjusted turnover rates for herbaceous leaves and surface litter at the ecosystem level to match the observed AGB used in this study, the significant differences in the estimated AGB between biogeochemical model simulations and field measurements are noteworthy (van der Werf et al., 2017). Furthermore, a high-resolution emissions inventory of $1 \times 1$ km was developed. This inventory allows for the capture and description of spatial variations and heterogeneity in small-scale OBB emissions, providing detailed information on spatial discrepancies that may be missed by large and coarse grid pixels (Shi et al., 2019).

We compared and validated the accuracy of monthly OBB carbon emission estimates in 14 global subregions using three global OBB fire products: GFAS, GFED, and FEER (Figure 9). The Taylor diagram illustrates a high degree of consistency between these estimates and other inventories in terms of the standard deviation, correlation coefficient, and amplitude ratio. Overall, the results of this study were closer to the GFED and GFAS inventories, with the best agreement observed with the GFAS inventory. Our results show a correlation coefficient >0.70 (p < 0.01) in over 80% of the regions with the other three inventories, indicating a strong positive correlation and consistency in data trends between our study and the other three lists in most regions. Furthermore, in the top three emission source regions, SHAF, SHSA, and NHAF, our correlation coefficients with the other three emission inventories were all >0.90, standard deviation ratios were <2.00, and normalized centered root mean square errors were <0.50. For example, compared with the other three inventories in the NHAF region, the correlation coefficients were all 0.97, with standard deviations of 0.93 (GFED), 0.66 (GFAS), and 1.24 (FEER). However, when compared with the FEER inventory, there were still disparities in the estimated results between the FEER inventory and this study. For instance, in low-emission regions, such as EQAS, NHSA, CEAM, and MIDE, the correlation coefficients ranged from 0.60 to 0.95, with standard deviation exceeding 1.00. This was attributed to FEER's use of the FRE-based approach and overestimation in quantifying small fire points (Ye et al., 2023).

In summary, we demonstrated that the GEIOBB was a dataset with relatively high-quality estimates of global OBB emissions and performed well across all time periods and regions. Overall, a comparison with multiple inventories indicated that our GEIOBB model could effectively capture the spatial and temporal distribution characteristics of OBB at large scales.

## 3.4 Advantages

To create a more accurate and effective biomass combustion carbon emission inventory, our research introduced three significant improvements compared to other inventory products. (1) The input global fire spot monitoring data from FY-3D showed a higher accuracy than MODIS in monitoring active fires (Xian et al., 2021). The OBB emissions exhibited significant consistency with the satellite fire detection results. Existing OBB emission estimation inventories differ mainly in the optimization of relevant parameters and estimation methods; however, they all rely on MODIS fire detection results as their primary data source. Our experiment utilized data from FY-3D GFR, which provides higher precision and the capability to quantify small-scale fire points more accurately (Yin et al., 2020) . Consequently, the accuracy of the OBB carbon emissions assessment significantly improved. (2) Satellite and observational AGB resulted in less uncertainty than land cover based available biomass. Previous studies have used fixed values for AGB with regional and land cover-based partitioning. Our research employed AGB inventory data, which, in contrast to the traditional method of regional sub-surface value assignment, better represents spatial variation trends. Additionally, by incorporating dynamic adjustment methods, we mitigated the temporal distribution shortcomings inherent in AGB data. This approach significantly enhances the portrayal of global biomass distribution across both time and space dimensions; (3) Spatially and temporally variable CF scaled by several vegetation indices can reflect a more accurate fraction of burned biomass than the allocated constants based on fire types. We optimized the previous single fixed value or simple formula-based definitions of CF by incorporating numerous parameters to better

represent vegetation combustion conditions. To address the varying fire conditions, we performed a detailed subdivision based on different fire types. This advancement over conventional methods of fixed-value assignment or unified fixed-value methods without substrate distinction, enables a more effective computation of burn factors for different types of fires, which can significantly enhance the delineation and understanding of burn factors in the biomass combustion process, paving the way for a more accurate carbon emission inventory. Through these notable improvements, our biomass combustion carbon emissions inventory is a robust tool that provides precise and insightful analyses instrumental for advancement in the field of biomass combustion carbon emissions assessment.

## 3.5 Uncertainties

There were relatively high uncertainties in the estimation of OBB emissions for the seven types; the uncertainties were associated with the burned area, F, CF, and EF. Although the FY-3D GFR dataset is reliable for most OBB events, its resolution of 1 km results in poor detection performance for small fire points (Zheng et al., 2023). The detected active fires were also underestimated due to cloud cover/thick smoke and omitted between satellite overpass, with an omission error of approximately from 10%–30% (Giglio et al., 2006; Roberts et al., 2009; Schroeder et al., 2008). Furthermore, the diurnal cycle cannot be sufficiently represented using observations from polar orbiting satellites, as these satellites have limited temporal coverage and may not capture the full range of fire activity throughout the day (Huang et al., 2024; Zheng et al., 2021b). Additionally, the uncertainties in the AGB calculations developed by Spawn and Gibbs (2020) ranged from 20% to 80%. Specifically, for approximately 80% of the area, the AGB uncertainties were <30%, whereas in regions, such as Africa and South America, high uncertainties of 60%–70% were observed. The estimated CF shows uncertainties of approximately 20–30% based on empirical formulas (Zhang et al., 2008). The typical uncertainties for trace gas and aerosol emission factors for each land type, as compiled by Shi et al. (2015), ranged from 20% to 50%. Owing to the inherent uncertainties in all input parameters, after estimating the OBB emission inventories, we quantitatively assessed the estimation uncertainties of all emission species using 20,000 Monte Carlo simulations to calculate emission ranges with a 90% confidence interval. Based on this, the emission ranges for different species are as follows: 1,168.02–4,120.83 Tg C, 2.31–5.48 Tg BC, 7.73–25.26 Tg $CH_4$, 193.11–505.66 Tg CO, 2,994.71–14,153.75 Tg $CO_2$, 3.31–8.49 Tg of $NH_3$, 7.92–26.08 Tg $NO_2$, 12.70–26.87 Tg $NO_X$, 8.37–29.35 Tg OC, 37.66–84.17 Tg $PM_{10}$, 19.85–61.62 Tg $PM_{2.5}$, and 1.67–6.69 Tg $SO_2$.

## 4 Conclusion

We developed a high-spatial-resolution (1 km×1 km grid) and daily inventory of global OBB emissions. Our inventory used the updated satellite-based burned area product (FY-3D GFR), observational and satellite-based AGB, and vegetation index-based spatiotemporally variable combustion efficiency data to estimate global OBB carbon emissions. The average annual estimated OBB emissions for 2020–2022 were 2,586.88 Tg C, 8841.45 Tg $CO_2$, 382.96 Tg CO, 15.83 Tg $CH_4$, 18.42 Tg $NO_X$, 4.07 Tg $SO_2$, 18.68 Tg OC, 3.77 Tg BC, 5.24 Tg $NH_3$, 15.85 Tg $NO_2$, 42.46 Tg $PM_{2.5}$ and 56.03 Tg $PM_{10}$.

Taking carbon emission as an example, the average annual estimated OBB emissions were 72.71 Tg of BONA, 165.72 Tg of TENA, 34.11 Tg of CEAM, 42.93 Tg of NHSA, 520.54 Tg of SHSA, 13.02 Tg of EURO, 8.37 Tg of MIDE, 394.32 Tg of NHAF, 847.03 Tg of SHAF, 167.35 Tg of BOAS, 27.93 Tg of CEAS, 197.29 Tg of SEAS, 13.20 Tg of EQAS, and 82.37 Tg of AUST. NHAF, as the primary contributor in January, accounted for 50.74%, 81.16%, and 67.66% in the three respective years. During the first peak of the years, March was mainly influenced by increased SEAS emissions (2020: 50.82%, 2021: 57.78%, and 2022: 40.67%). In 2020, the annual peak occurred in September at 500.62 Tg, while in 2021 and 2022, it shifted to August, reaching 729.37 and 357.57 Tg, respectively. Peaks from savanna grasslands, woody savanna/shrubs, and tropical forest regions were 7.54 (38.37%), 7.12 (37.42%), and 3.36 Tg (31.01%), respectively.

We demonstrated that savanna grassland contributed the largest portion (46.74%) of total emissions, followed by woody savanna/shrubs (33.04%) and tropical forest (12.11%). Total OBB carbon emissions were the highest from SHAF, followed by SHSA, and NHAF. The fire types where fires occurred were predominantly savanna grasslands, woody savanna/shrubs, and tropical forest in the SHAF, SHSA, and NHAF, and woody savanna/shrubs in SEAS. Furthermore, our data indicate a pronounced seasonal trend in carbon emissions. Regions, such as the SHAF, SHSA, and TENA, played pivotal roles, accounting for the surge in global carbon emissions observed in August.

Our high-spatial-resolution multi-species emission inventory and spatiotemporal characteristics analysis will provide scientific and reliable evidence for formulating carbon emission policies and assessing temporal emission variation. Effective control of the savanna grasslands fire in the SHAF, SHSA, and NHAF as well as tropical forest fires in the SHSA and woody savanna/shrubs fires in the SHAF can greatly reduce carbon emissions. Moreover, this carbon emissions inventory can be used for regional biogeochemical circulation, atmospheric chemical simulations, and environmental health impacts. The accuracy and depth of our findings further underscore the potential for combining our bottom-up approach with top-down satellite observational methods, paving the way for refinement in future studies.

**Code/Data availability.** The GEIOBB dataset can be downloaded at http://figshare.com with the following identifier DOI: https://doi.org/10.6084/m9.figshare.24793623 (Liu et al., 2023).

**Author contributions.** YL and YS produced GEIOBB. JC, WZ, GW, and TS conducted the data processing. YL, YS and JC conceived the manuscript. YL and YS conducted data analysis and produced figures. YL and YS wrote the draft. YL, YS, TS and JC reviewed and revised the manuscript.

**Competing interests.** The contact author has declared that neither they nor their co-authors have any competing interests.

**Acknowledgements.** Thanks to the anonymous reviewers for the valuable comments. This research is supported by the State Key Laboratory of Remote Sensing Science of Aerospace Information Research Institute and Innovation Center for FengYun Meteorological Satellite of National Satellite Meteorological Center.

**Financial support.** This research is supported by the National Key Research and Development Program of China (2023YFB3907404), FY-3 Lot 03 Meteorological Satellite Engineering Ground Application System Ecological Monitoring and Assessment Application Project Phase I (ZQC-R22227) and National Natural Science Foundation of China (42071398).

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

**Table 1. Emission factor (g/kg) of different species.**

| Species | Grasslands and Savannas | Woody Savannas or Shrubs | Tropical Forests | Temperate Forests | Boreal Forests | Temperate Evergreen Forests | Crop | | | |
|---|---|---|---|---|---|---|---|---|---|---|
| | | | | | | | Maize | Sugar | Rice | Wheat |
| C | 488.31 | 489.41 | 491.77 | 468.31 | 478.88 | 493.18 | 687.09 | 323.35 | 368.04 | 429.17 |
| $CO_2$ | 1,686[a] | 1,681[a] | 1,643[a] | 1,510[a] | 1,565[b] | 1,623[a] | 2,327[c] | 1,130[c] | 1,177[c] | 1,470[e] |
| CO | 63.00[a] | 67.00[a] | 93.00[a] | 122.00[a] | 111.00[b] | 112.00[a] | 114.70[c] | 34.70[c] | 93.00[c] | 60.00[e] |
| $CH_4$ | 2.00[a] | 3.00[a] | 5.10[a] | 5.61[a] | 6.00[b] | 3.40[a] | 4.40[c] | 0.40[c] | 9.59[c] | 3.40[e] |
| $NO_X$ | 3.90[a] | 3.65[a] | 2.60[a] | 1.04[a] | 0.95[b] | 1.96[a] | 4.30[c] | 2.60[c] | 2.28[c] | 3.30[e] |
| $SO_2$ | 0.90[a] | 0.68[a] | 0.40[a] | 1.10[a] | 1.00[b] | 1.10[a] | 0.44[c] | 0.22[c] | 0.18[c] | 0.85[e] |
| OC | 2.60[a] | 3.70[a] | 4.70[a] | 7.60[a] | 7.80[b] | 7.60[a] | 2.25[c] | 3.30[c] | 2.99[c] | 3.90[d] |
| BC | 0.37[a] | 1.31[a] | 0.52[a] | 0.56[a] | 0.20[b] | 0.56[a] | 0.78[d] | 0.82[d] | 0.52[d] | 0.52[d] |
| $NH_3$ | 0.56[a] | 1.20[a] | 1.30[a] | 2.47[a] | 1.80[b] | 1.17[a] | 0.68[c] | 1.00[c] | 4.10[c] | 0.37[e] |
| $NO_2$ | 3.22[a] | 2.58[a] | 3.60[a] | 2.34[a] | 0.63[b] | 2.34[a] | | 2.99[f] | | |
| $PM_{2.5}$ | 7.17[a] | 7.10[a] | 9.90[a] | 15.00[a] | 18.40[b] | 17.90[a] | | 6.43[f] | | |
| $PM_{10}$ | 7.20[a] | 11.4[a] | 18.50[a] | 16.97[a] | 18.40[b] | 18.40[a] | | 7.02[f] | | |

All the value of C were Calculated by $CO_2$, CO, and $CH_4$.

[a] is average value from (Akagi et al., 2011).

[b] is average from (Akagi et al., 2011) and (Urbanski, 2014).

[c] is average from (Akagi et al., 2011; Fang et al., 2017; Liu et al., 2016; Santiago-De La Rosa et al., 2018; Stockwell et al., 2015).

[d] is from (Kanabkaew and Kim Oanh, 2011).

[e] is from (Cao et al., 2008).

[f] is from (Li et al., 2007).

**Table 2. Global OBB annual emissions and region-specific average annual emissions during 2020–2022 (Tg Species/year).**

| | C | BC | $CH_4$ | CO | $CO_2$ | $NH_3$ | $NO_2$ | $NO_X$ | OC | $PM_{10}$ | $PM_{2.5}$ | $SO_2$ |
|---|---|---|---|---|---|---|---|---|---|---|---|---|
| 2020 | 2,861.05 | 4.09 | 17.39 | 423.12 | 9,777.79 | 5.76 | 17.58 | 20.37 | 20.64 | 61.59 | 47.18 | 4.54 |
| 2021 | 2,991.16 | 4.52 | 18.22 | 439.67 | 10,226.55 | 6.11 | 18.17 | 21.36 | 21.64 | 64.76 | 48.89 | 4.70 |
| 2022 | 1,908.42 | 2.69 | 11.87 | 283.09 | 6,520.04 | 3.87 | 11.82 | 13.53 | 13.74 | 41.76 | 31.31 | 2.97 |
| average | 2,586.88 | 3.77 | 15.83 | 381.96 | 8,841.46 | 5.24 | 15.85 | 18.42 | 18.68 | 56.03 | 42.46 | 4.07 |
| BONA | 72.71 | 0.16 | 0.49 | 10.92 | 248.08 | 0.18 | 0.36 | 0.49 | 0.63 | 1.80 | 1.29 | 0.11 |
| TENA | 165.73 | 0.30 | 1.02 | 26.14 | 563.78 | 0.38 | 0.92 | 1.11 | 1.45 | 3.98 | 3.18 | 0.28 |
| CEAM | 34.11 | 0.06 | 0.23 | 5.21 | 116.26 | 0.08 | 0.20 | 0.23 | 0.27 | 0.81 | 0.56 | 0.05 |
| NHSA | 42.93 | 0.06 | 0.28 | 6.42 | 146.58 | 0.08 | 0.28 | 0.30 | 0.31 | 1.01 | 0.70 | 0.06 |
| SHSA | 520.55 | 0.61 | 3.74 | 83.09 | 1,767.83 | 1.12 | 3.42 | 3.45 | 4.01 | 13.00 | 9.08 | 0.74 |
| EURO | 13.02 | 0.02 | 0.09 | 2.02 | 44.33 | 0.03 | 0.08 | 0.09 | 0.09 | 0.26 | 0.22 | 0.02 |
| MIDE | 8.37 | 0.01 | 0.06 | 1.28 | 28.54 | 0.02 | 0.05 | 0.06 | 0.05 | 0.15 | 0.13 | 0.01 |
| NHAF | 394.25 | 0.41 | 2.05 | 54.58 | 1,354.19 | 0.62 | 2.56 | 2.99 | 2.39 | 7.01 | 6.01 | 0.66 |
| SHAF | 847.03 | 1.28 | 4.52 | 116.23 | 2,910.72 | 1.52 | 5.17 | 6.40 | 5.55 | 16.48 | 12.82 | 1.38 |
| BOAS | 167.35 | 0.31 | 0.98 | 23.57 | 573.90 | 0.35 | 0.93 | 1.22 | 1.22 | 3.53 | 2.68 | 0.27 |
| CEAS | 27.93 | 0.04 | 0.21 | 4.55 | 94.68 | 0.08 | 0.17 | 0.19 | 0.20 | 0.56 | 0.47 | 0.04 |
| SEAS | 197.29 | 0.37 | 1.54 | 32.49 | 668.10 | 0.55 | 1.16 | 1.26 | 1.71 | 5.24 | 3.50 | 0.28 |
| EQAS | 13.20 | 0.03 | 0.10 | 2.04 | 44.94 | 0.03 | 0.08 | 0.09 | 0.11 | 0.36 | 0.22 | 0.02 |
| AUST | 82.38 | 0.11 | 0.52 | 13.41 | 279.54 | 0.19 | 0.48 | 0.54 | 0.70 | 1.83 | 1.59 | 0.15 |


**Table 3. Annual carbon emissions from global OBB in different regions during 2020–2022 (Unit: Tg/year).**

| Different Region | Savanna Grasslands | Woody Savanna/Shrubs | Tropical Forest | Temperate Forest | Boreal Forest | Temperate Evergreen Forest | Crop | Total |
|---|---|---|---|---|---|---|---|---|
| BONA | 4.43 | 57.55 | 0.00 | 0.36 | 7.58 | 2.15 | 0.63 | 72.70 |
| TENA | 41.20 | 83.89 | 0.00 | 5.71 | 0.00 | 30.85 | 4.07 | 165.72 |
| CEAM | 8.62 | 17.47 | 4.57 | 2.33 | 0.00 | 0.02 | 1.11 | 34.12 |
| NHSA | 19.12 | 11.08 | 12.23 | 0.28 | 0.00 | 0.00 | 0.22 | 42.93 |
| SHSA | 225.86 | 76.69 | 177.17 | 27.49 | 0.00 | 0.37 | 12.98 | 520.56 |
| EURO | 5.21 | 4.60 | 0.00 | 0.71 | 0.19 | 0.40 | 1.92 | 13.03 |
| MIDE | 4.95 | 1.17 | 0.00 | 0.15 | 0.00 | 0.33 | 1.78 | 8.38 |
| NHAF | 300.21 | 47.03 | 30.31 | 3.93 | 0.00 | 0.00 | 12.78 | 394.26 |
| SHAF | 465.86 | 324.09 | 41.17 | 12.70 | 0.00 | 0.00 | 3.22 | 847.04 |
| BOAS | 59.51 | 95.97 | 0.00 | 1.29 | 9.01 | 0.07 | 1.50 | 167.35 |
| CEAS | 10.31 | 7.71 | 0.68 | 1.86 | 0.59 | 0.33 | 6.45 | 27.93 |
| SEAS | 21.46 | 101.57 | 42.39 | 22.26 | 0.00 | 0.26 | 9.36 | 197.30 |
| EQAS | 1.43 | 7.23 | 4.45 | 0.02 | 0.00 | 0.00 | 0.08 | 13.21 |
| AUST | 40.95 | 18.66 | 0.35 | 13.57 | 0.00 | 6.86 | 1.97 | 82.36 |

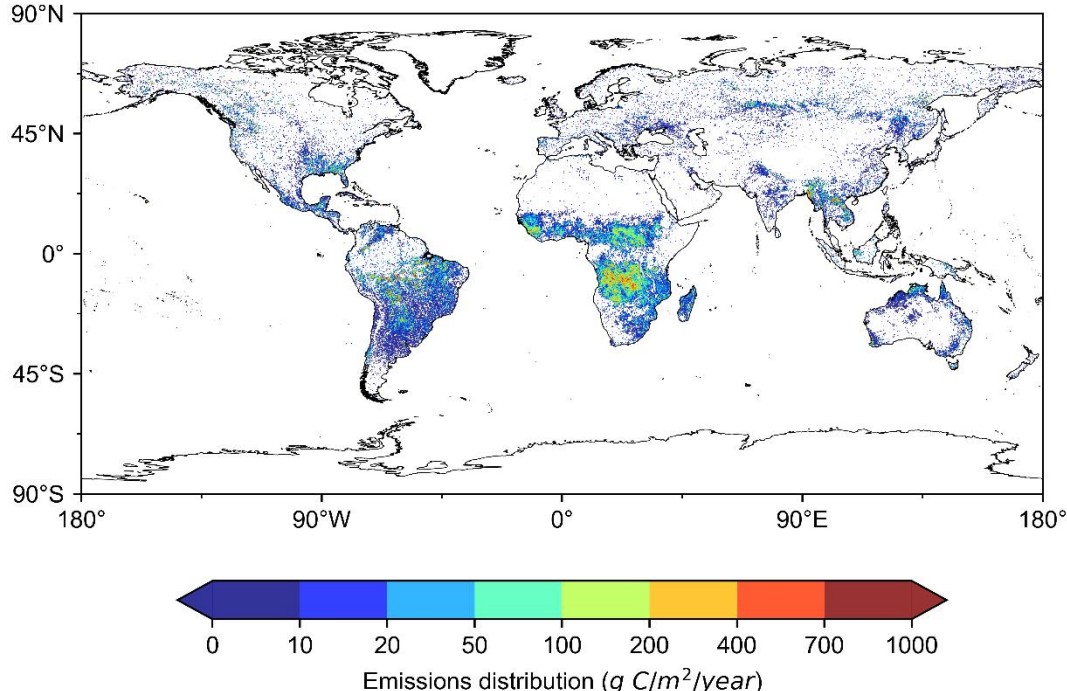

**Figure 1: Spatial distribution of annual average OBB carbon emissions (1 × 1 km) during 2020–2022.**

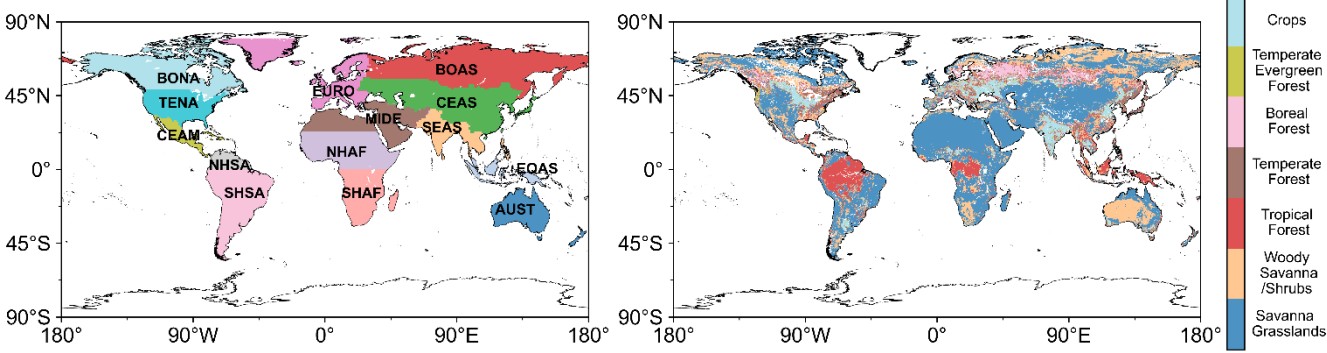

**Figure 2 (a) Global geographic regions and its abbreviations. The acronyms on the figure represent the following: BONA: Boreal North America; TENA: Temperate North America; CEAM: Central America; NHSA: Northern Hemisphere South America; SHSA: Southern Hemisphere South America; EURO: Europe; MIDE: Middle East; NHAF: Northern Hemisphere Africa; SHAF: Southern Hemisphere Africa; BOAS: Boreal Asia; CEAS: Central Asia; SEAS: Southeast Asia; EQAS: Equatorial Asia; AUST: Australia and New Zealand;(b) Global land cover type reclassification.**

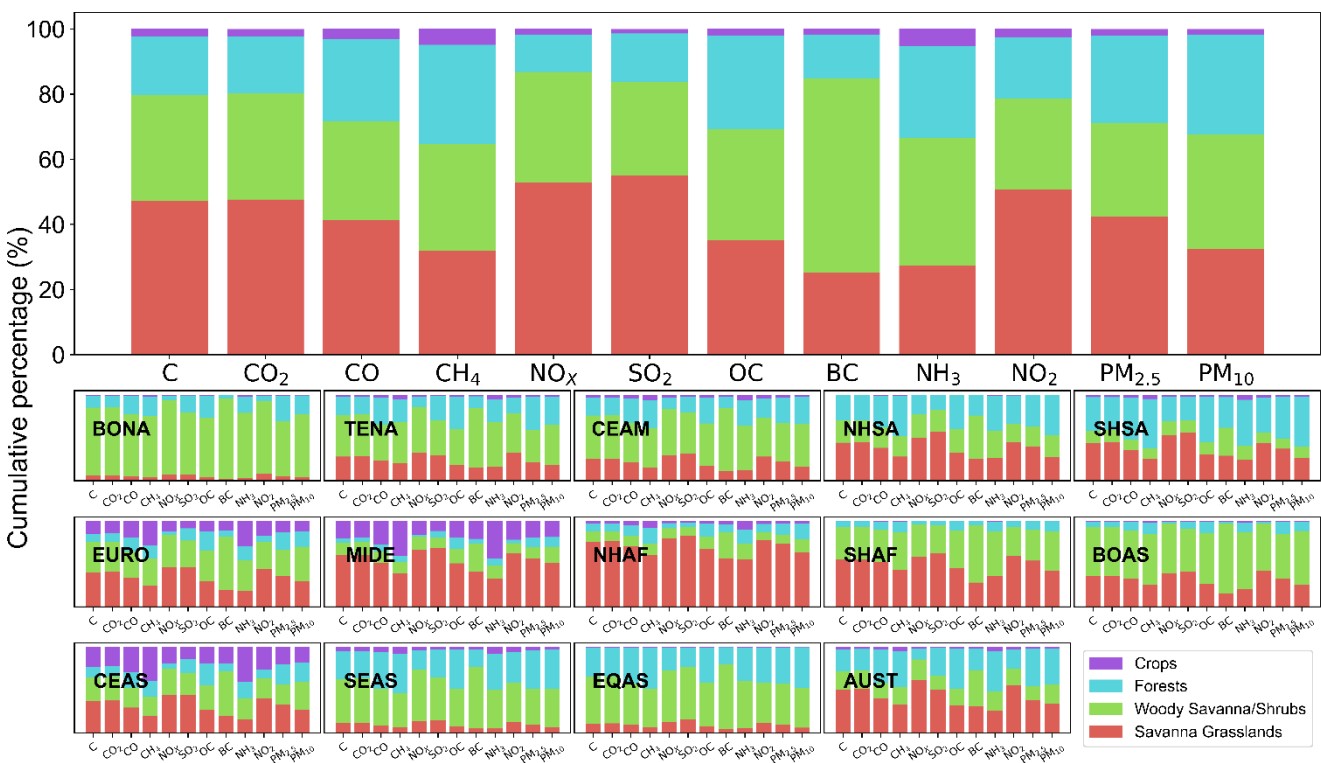

Figure 3: Cumulative percentage of annual OBB emissions for each land type in each region during 2020–2022.

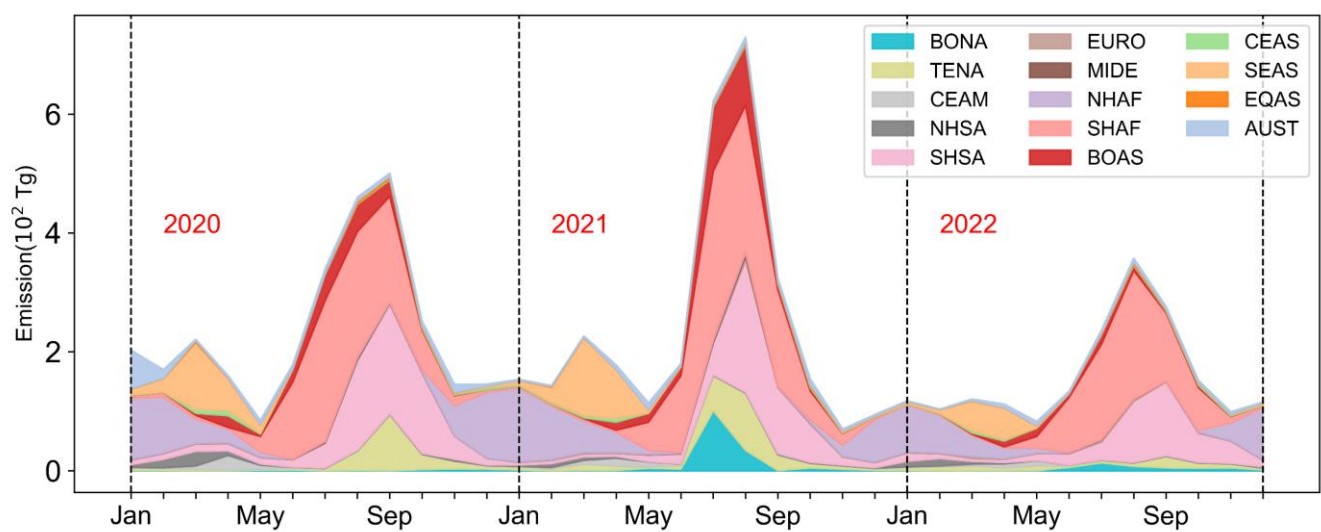

**Figure 4: Global OBB carbon emissions in different regions during 2020–2022.**

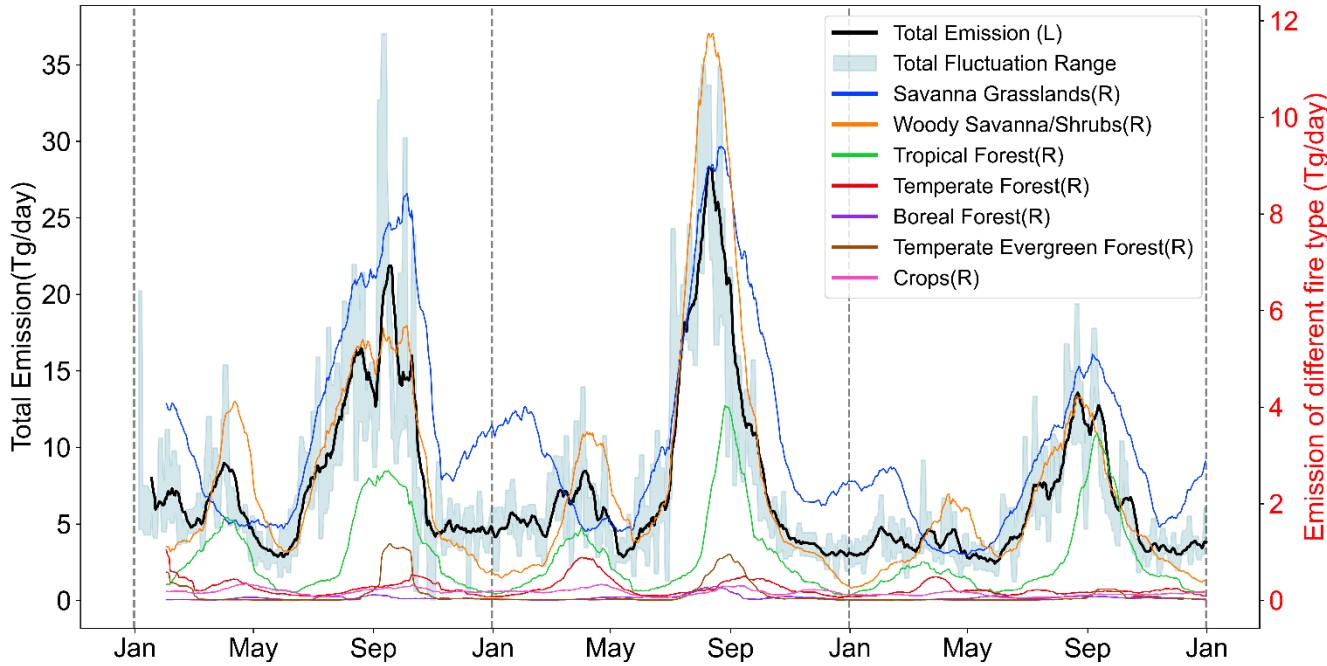


**Figure 5: Variations in total global OBB carbon emissions and carbon emissions in different fire types across various regions from 2020 to 2022.**

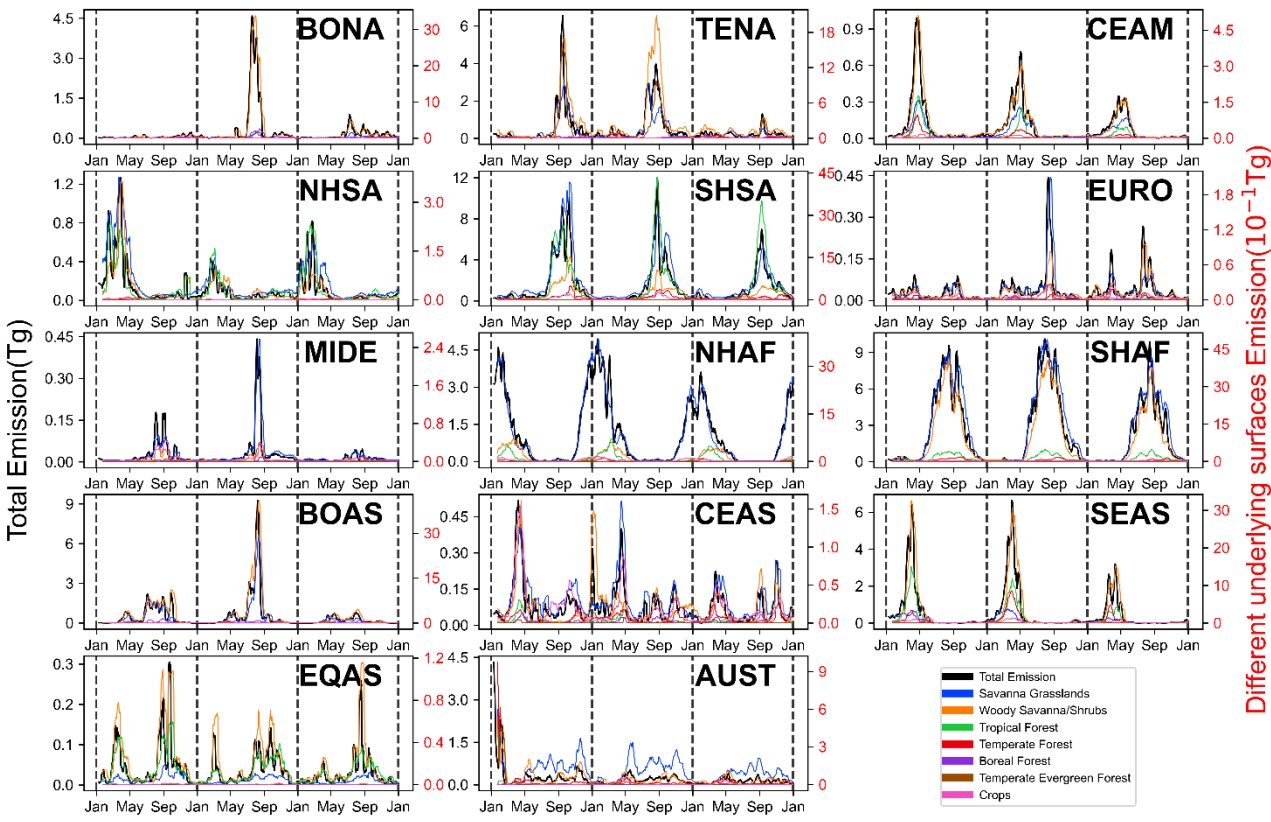

**Figure 6: Global OBB emissions for different fire types in different regions (averaged over a 15-day window) from 2020 to 2022.**

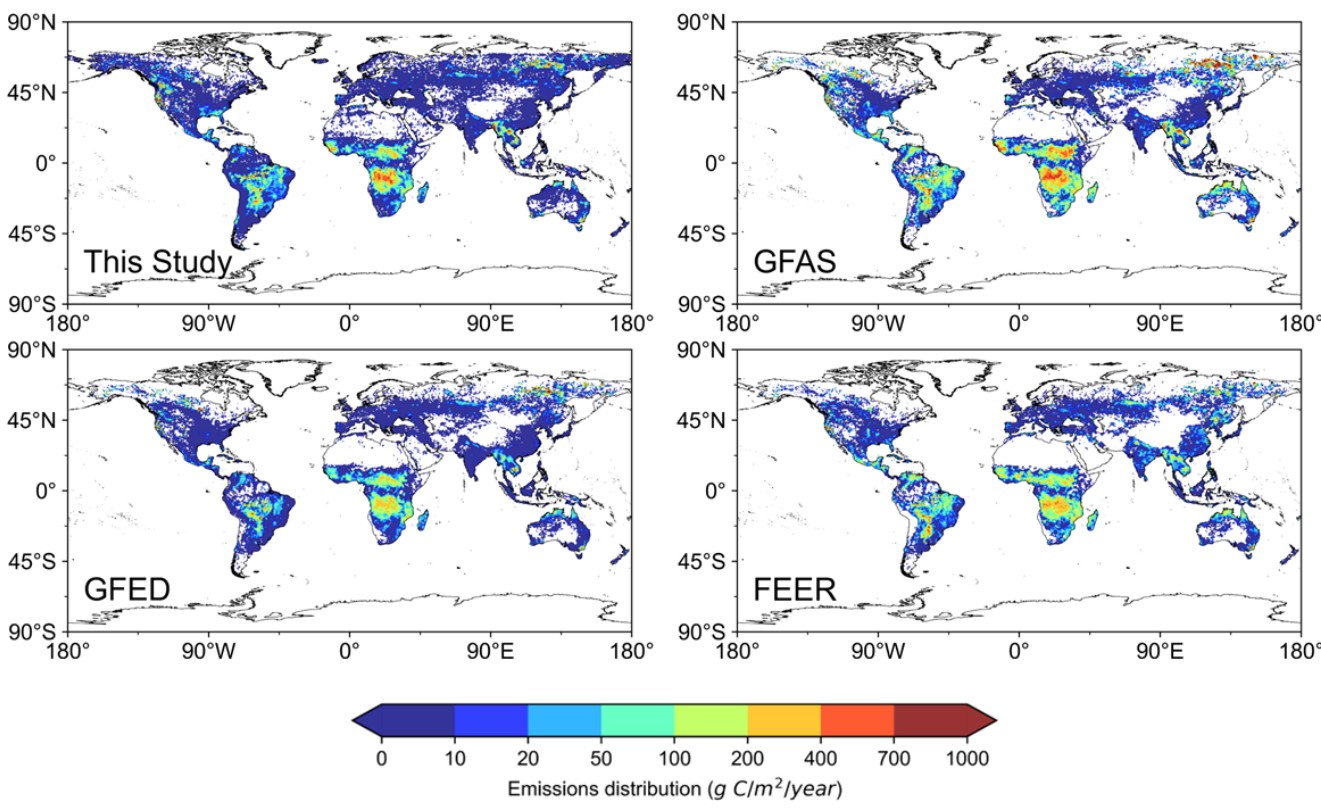

**Figure 7: Comparison between this study and other emission inventories during 2020–2022 average emissions at 0.5° resolution.**

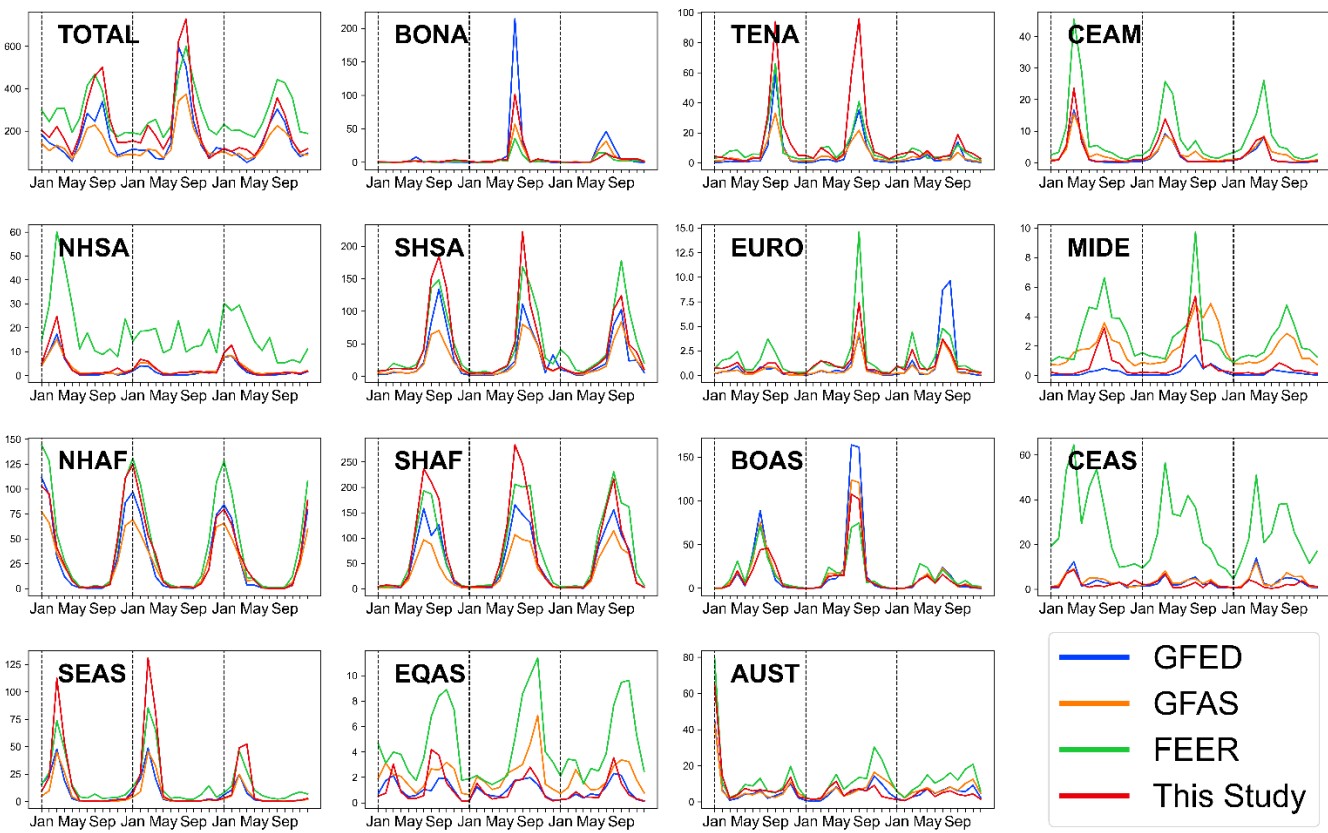


**Figure 8: Comparison of monthly emissions in different regions of this and other emission inventories.**

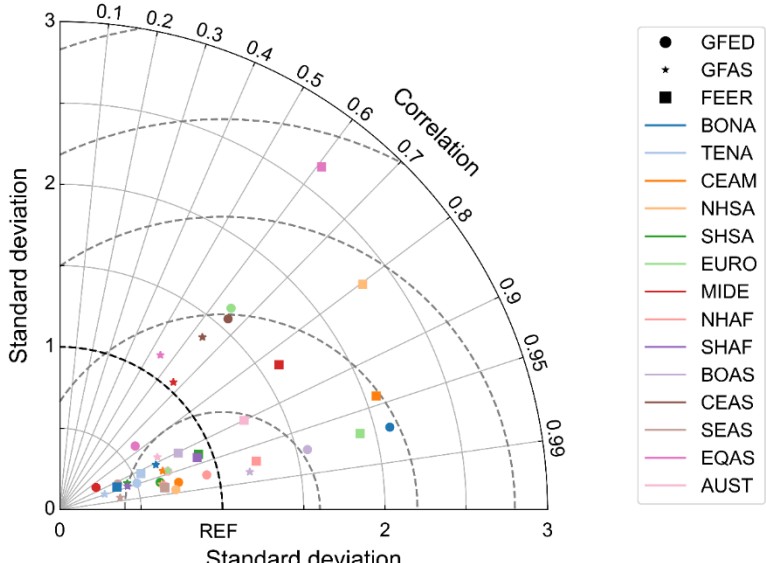

**Figure 9: Normalized Taylor diagram plot of the comparison between GFED, GFAS, and FEER and this study with monthly OBB carbon emission.**