# Peer review of "Global Emissions Inventory from Open Biomass Burning (GEIOBB): Utilizing Fengyun-3D global fire spot monitoring data"

_Earth System Science Data, 2023_

## Author Response (AR1)

This study developed a global daily emission inventory of OBB with 1km×1km based on global fire point monitoring data from the Chinese Fengyun-3D satellite, fuel loading, combustion factor and emission factor. Considering the scientific impact of each study, several analysis is needed to substantiate the conclusions in your manuscript. Firstly, the manuscript emphasizes that the compared with MODIS, significant advantage of using the FY-3D fire detection product is the ability to enhance the detection of small fires, but the analysis of the results does not show how much the use of the FY-3D detection product has increased the emission estimates of small fires? Secondly, in the section about verification, the manuscript emphasizes the consistency with other datasets, but does not quantify the advantages of this study. Thirdly, the advance of activity data selected in this study needs to be verified, such as the quality and the resolution of the data. The manuscript can be considered for publication if the issues mentioned above and following specific comment could be addressed.

We thank the Reviewer for the constructive comments and suggestions. We shall revise the manuscript accordingly, and we address the comments as follows.

**Specific comments:**

*P1 line23-25: The full name are not given for some regions (e.g., BONA), and them are given for some regions (e.g., SHSA).*

We added full name for other regions.

"72.71 (Boreal North America; BONA), 165.7 (Temperate North America, TENA), 34.1 (Central America; CEAM), 42.9 (Northern Hemisphere South America; NHSA), 520.5 (Southern Hemisphere South America; SHSA), 13 (Europe; EURO), 8.4 (Middle East; MIDE), 394.3 (Northern Hemisphere Africa; NHAF), 847 (Southern Hemisphere Africa; SHAF), 167.4 (Boreal Asia; BOAS), 27.9 (Central Asia; CEAS), 197.3 (Southeast Asia; SEAS), 13.2 (Equatorial Asia; EQAS), and 82.4 (Australia and New Zealand; AUST) Tg".

*P2 line64-65: The detection accuracy of MODIS and other related indexes should be clearly given to facilitate readers to compare directly. As well as the comparison with MODIS, other commonly used polar-orbiting satellite sensors (SNPP-VIIRS, Landsat-8, etc.) can be considered for comparison to highlight the advantages of FY-3D.*

We revised according to the reviewer's comment. We added comparisons with other product. The changes are as follows.

"Furthermore, the Global Fire Monitoring (GFR) product with FY−3D employs optimized automatic identification algorithms for fire spots (Tianchan and Wei, 2022), leading to an improved accuracy of fire point detection. This resulted in an impressive overall accuracy rate of 79.43% and an exclusion omission error accuracy of 88.50%, surpassing the capabilities of MODIS satellite products (Chen et al., 2022; Xian et al., 2021), based on field−collected references throughout 2020 in China. The cross−verification between MODIS and FY−3D shows the highest consistency results (over 80%) in Africa and Asia, while America, Europe, and Oceania demonstrate consistency exceeding 70% (Chen et al., 2022). In July, August, and September, the number of fire spots was higher, with a mean consistency of over 85% between MODIS and FY−3D fire products (Chen et al., 2022). Although Landsat Fire and Thermal Anomaly (LFTA) product has finer spatial resolution, its lower temporal

resolution typically allows global coverage only every 16 days, which does not allow for frequent detection of biomass burning activity. Therefore, employing the FY–3D GFR product and allocation approaches for small fires is expected to yield reliable estimates of OBB emissions."

Furthermore, we added the comparison of parameters related to MERSI–II, MODIS, and VIIRS. Please refer to Table 1.

*P4 line128: It is suggested that formulas could be transferred to the manuscript from SI, with the supplement of corresponding unit of the variable.*

We revised according to the reviewer's comment. We transferred the formulas to the manuscript from SI, with the supplement of corresponding unit of the variable. Please refer to Line 149.

*P5 line148: Source of the constant 0.013 in the formula? Empirical values should give literature. The fitted values should depict the fitting process and significance test results.*

We revised according to the reviewer's comment. We added the reference to the constant 0.013 in the formula.

*P5 line152: There are other products (MODIS) of NDVI with a time interval of 8d. Why the products of 16d was selected in this study?*

The 8d product is based on mod09 calculations and because it is measured in the absence of atmospheric scattering or absorption, the product contains lower data, as well as cloud cover. The 16-day product was derived from the 8-day product, offering advantages such as an enhanced signal-to-noise ratio and reduced cloud contamination. This is achieved through a longer temporal composite period, leading to more dependable and precise assessments of vegetation health and productivity.

In the process of assessing vegetation conditions, the accuracy and completeness of the data are prioritized over the requirement for event resolution. Therefore, we opt to utilize the 16-day product.

*P6 Table 1: The EF for specific biomass (e.g., crop) is fixed value for different regions with various crop distribution characteristic. Regional or crop differences should be reflected in EF values.*

We made a distinction in the CF for different types of fire event emissions in different regions, and the EF was only used as a factor between the pollutant emitted and the type of fire.

**Table 2. Emission factor (g/kg) of different species.**

| Species | Grasslands and Savannas | Woody Savanna or Shrubs | Tropical Forest | Temperate Forest | Boreal Forest | Temperate Evergreen Forest | Crop | | | |
|---|---|---|---|---|---|---|---|---|---|---|
| | | | | | | | Maize | Sugar | Sugar | Wheat |
| C | 488.31 | 489.41 | 491.77 | 468.31 | 478.88 | 493.18 | 687.09 | 323.35 | 368.04 | 429.17 |
| $CO_2$ | 1,686[a] | 1,681[a] | 1,643[a] | 1,510[a] | 1,565[b] | 1,623[a] | 2,327[c] | 1,130[c] | 1,177[c] | 1,470[e] |
| CO | 63.00[a] | 67.00[a] | 93.00[a] | 122.00[a] | 111.00[b] | 112.00[a] | 114.70[c] | 34.70[c] | 93.00[c] | 60.00[e] |

| | | | | | | | | | | |
|---|---|---|---|---|---|---|---|---|---|---|
| $CH_4$ | 2.00[a] | 3.00[a] | 5.10[a] | 5.61[a] | 6.00[b] | 3.40[a] | 4.40[c] | 0.40[c] | 9.59[c] | 3.40[e] |
| $NO_X$ | 3.90[a] | 3.65[a] | 2.60[a] | 1.04[a] | 0.95[b] | 1.96[a] | 4.30[c] | 2.60[c] | 2.28[c] | 3.30[e] |
| $SO_2$ | 0.90[a] | 0.68[a] | 0.40[a] | 1.10[a] | 1.00[b] | 1.10[a] | 0.44[c] | 0.22[c] | 0.18[c] | 0.85[e] |
| OC | 2.60[a] | 3.70[a] | 4.70[a] | 7.60[a] | 7.80[b] | 7.60[a] | 2.25[c] | 3.30[c] | 2.99[c] | 3.90[d] |
| BC | 0.37[a] | 1.31[a] | 0.52[a] | 0.56[a] | 0.20[b] | 0.56[a] | 0.78[d] | 0.82[d] | 0.52[d] | 0.52[d] |
| $NH_3$ | 0.56[a] | 1.20[a] | 1.30[a] | 2.47[a] | 1.80[b] | 1.17[a] | 0.68[c] | 1.00[c] | 4.10[c] | 0.37[e] |
| $NO_2$ | 3.22[a] | 2.58[a] | 3.60[a] | 2.34[a] | 0.63[b] | 2.34[a] | | 2.99[f] | | |
| $PM_{2.5}$ | 7.17[a] | 7.10[a] | 9.90[a] | 15.00[a] | 18.40[b] | 17.90[a] | | 6.43[f] | | |
| $PM_{10}$ | 7.20[a] | 11.4[a] | 18.50[a] | 16.97[a] | 18.40[b] | 18.40[a] | | 7.02[f] | | |

All the value of C were Calculated by $CO_2$, CO, and $CH_4$.

[a] is average value from (Akagi et al., 2011).

[b] is average from (Akagi et al., 2011) and (Urbanski, 2014).

[c] is average from (Akagi et al., 2011; Fang et al., 2017; Liu et al., 2016; Santiago-De La Rosa et al., 2018; Stockwell et al., 2015).

[d] is from (Kanabkaew and Kim Oanh, 2011).

[e] is from (Cao et al., 2008).

[f] is from (Li et al., 2007).

*P11 line269: What does "intensify both the frequency and frequency of fires in the area" mean?*
We changed "intensify both the frequency and frequency of fires in the area" to "intensify frequency of fires in the area" make it clear to understand.

*P15 line352: Why the dataset is not include FINN (e.g., FINNv2.5)? The resolution of it is the same with the dataset developed in this study (1km, 1d).*
The analytical discussion in this study is based on the example of Carbon, for which no estimation exists in FINN (FINNv2.5, 0.1 degree), so there is no FINN product. In SI, we added the comparison results with FINN for CO2 analysis in different regions.

*P16 line377-379: There is a lack of clarity in the explanation of how FY-3D can capture small fires more effectively compared to MODIS, and how the difference in transit times between the two satellites affects the detection of agricultural small fires. More data analysis is needed to support this question.*
*P18 line410: The article should add a comparative analysis of how much the addition of FY-3D improves emission estimates for small fires, which is a key factor in determining the innovativeness of the study.*
We added Comparison of parameters related to MERSI–II, MODIS, and VIIRS to explain the advantages of the FY-3D for fire point detection at the hardware level. Please refer to Table 1. Furthermore, we added more details about FY-3D GFR product to support FY-3D can capture small fires more effectively.

**Table 1. Comparison of parameters related to MERSI–II, MODIS, and VIIRS.**

| | MERSI–II (FY–3D) | MODIS (AQUA) | VIIRS ((NOAA–20)) |
|---|---|---|---|
| Orbit altitude (km) | 836 | 705 | 824 |
| Equator Crossing time | 14:00 LT | 13:30 LT | 14:20 LT |
| Swath (km) | 2900 | 2330 | 3060 |
| Pixel resolution at nadir (km) | 1 | 1 | 0.75/0.375 |
| Pixel resolution at the edge (km) | >6 | 4 | 1.5/0.75 |
| ID MIR Band (s) | 21 | 21/22 | M–13/I–4 |
| Spectral range (µm) | 3.973–4.128 | 3.929–3.989 3.940–4.001 | 3.973–4.128 3.550–3.930 |
| TMAX (SNR–NEΔT on orbit) | 380 K (0.25) | 500 K (0.183) 331 K (0.019) | 634 K (0.04) |
| ID TIR Band (s) | 24 | 31 | M–15/I–5 |
| Spectral range (µm) | 10.300–11.300 | 10.780–11.280 | 10.263–11.263 10.500–12.400 |
| TMAX (SNR–NEΔT on orbit) | 330 K (0.4) | 400 K (0.017) | 343 K (0.03) |

"…Compared to MODIS, FY–3D fire products have been optimized in terms of auxiliary parameters, fire identification and re–identification. Firstly, FY–3D introduces the adaptive threshold and eliminates the limitations by fixed thresholds of MODIS and VIIRS algorithms by automatic identification algorithms for fire spot detection (Chen et al., 2022). Secondly, FY–3D uses a re–identification index reflecting varying geographical latitude and underlying surfaces types, together with the effect by cloud, water, and bare land (Zheng et al., 2020). The integration of multiple influencing factors increases the accuracy of fire detection. For example, the influences of factory thermal anomalies and high reflectance of photovoltaic power plants are greatly removed. Finally, the far–infrared channel employed in FY–3D has a high resolution of 250 m, higher than MODIS with 1 km, resulting in higher accuracy in big fire detection (Zheng and Chen, 2020). Overall, the FY–3D GFR product achieves an accuracy of 94.0% globally, with accuracies of 94.6%, 94.1%, 90.6%, 91.8%, and 92.7% in south–central Africa, east central south America, Siberia, Australia and Indochinese Peninsula (Chen et al., 2022), respectively. Specifically, due to the removal of underlying surface interference in China, the FY-3D achieves accuracies of 79.43% and 88.50% for accuracy and accuracy without omission, respectively, both of which are higher than the accuracies of 74.23% and 79.69% achieved by MODIS (Chen et al., 2022)."

*P19 line435: What is the difference between the 1° spatial resolution of FY-3D mentioned here and the 1km mentioned previously (line 106)?*

We changed "1 degree" to "1 km".

The authors claim that "the GFR product, which was integrated with the MERSI–2 instrument, exhibited superior judgment accuracy" (Line 102-103), "Consequently, our inventory yielded accurate assessment results and captured the spatial variation and heterogeneity of minor OBB emissions effectively" (Line 358-359), and "the accuracy of the OBB carbon emissions assessment significantly improved" (Line 417-418). In my opinion, these claims are not sufficiently justified in the manuscript. First, the superiority of MERSI-2 over MODIS in detecting active fires is not well explained. MERSI-2 on FY-3D has higher spatial resolution than MODIS in the visible and NIR bands. However, the active fire algorithm mainly uses the mid-infrared band, where both MERSI-2 and MODIS have a spatial resolution of 1km. The authors cited a number of previous papers (such as Dong et al. 2022 and Chen et al., 2022) to show better fire detection accuracy from MERSI-2 than from MODIS. However, these studies were mostly based on comparisons with limited data samples from manual inspection, and are not very convincing to me. Second, there are many limitations in the algorithm that are not mentioned in the manuscript. For example, this study used AGB as the fuel load, completely ignoring the emissions from soil organic matter burning. The omission error of active fires due to cloud cover/thick smoke is also not quantified. Third, this emissions dataset was derived from FY-3D active fires, but many MODIS products are still needed to generate GEIOBB. The use of MODIS products, which include MOD44B, MODIS NDVI, and MODIS land cover type data, may hinder the effect of quantifying global fire emissions after Terra and Aqua are gone. This potential problem should also be addressed in this paper.

We thank the Reviewer for the constructive comments and suggestions. We shall revise the manuscript accordingly, and we address the comments as follows.

1. We added more details about FY-3D to support FY-3D can capture small fires more effectively. Please refer to line 136 and Table 1.

2. Soil organic matter is also a contributor to the total OBB. However, a very small proportion of emissions can be found in most forest and grassland except in peatland burning in Indonesia. After calculation, we found the carbon emissions in Indonesia was very low since the drought and human induced fires detected were very low during 2020-2022.

   Additionally, we analyzed the omission error of active fires due to cloud cover/thick smoke.

   "The detected active fires were also underestimated due to cloud cover/thick smoke, with an omission error of approximately from 10%–30% (Schroeder et al., 2008; Roberts et al., 2009; Giglio et al., 2006)."

3. In this study, MODIS provide globally commonly used available products including the tree cover, land cover, NDVI to generate OBB emissions worldwide. Other available high-resolution and multi-year products will be used in our study to produce more reliable emission inventory.

*Many statements are incorrect or lacking scientific support.*

*The estimation method and the use of fuel loading (F) are not clearly described. While the authors mention in the manuscript that three data sources, NDVI, TC, and AGB are used for fuel loading, the approach for combining different data streams and forming the fuel loading is embedded in the supporting text only. This formula was*

*presented without any scientific justification or explanation (there are also some errors in the description of this formula, e.g., 2020 should be corrected to 2010).*

We revised according to the reviewer's comment. We added a description of fuel compliance with fuel loading in the main text and added references related to the formula. We corrected the misdescription in the formula. Please refer to line 149.

*The use of emission factors (EF) is also ambiguously described in the manuscript. In section 2.4, the authors simply listed a table of EFs without indicating the specific data sources. Although references to various studies and some locally measured data are cited, the specific methodology employed to construct Table 1 remains undisclosed.*

We add data sources for EF in Table 1. Please refer to Table 2.

**Table 2. Emission factor (g/kg) of different species.**

| Species | Grasslands and Savannas | Woody Savanna or Shrubs | Tropical Forest | Temperate Forest | Boreal Forest | Temperate Evergreen Forest | Crop Maize | Crop Sugar | Crop Sugar | Crop Wheat |
|---|---|---|---|---|---|---|---|---|---|---|
| C | 488.31 | 489.41 | 491.77 | 468.31 | 478.88 | 493.18 | 687.09 | 323.35 | 368.04 | 429.17 |
| $CO_2$ | 1,686[a] | 1,681[a] | 1,643[a] | 1,510[a] | 1,565[b] | 1,623[a] | 2,327[c] | 1,130[c] | 1,177[c] | 1,470[e] |
| CO | 63.00[a] | 67.00[a] | 93.00[a] | 122.00[a] | 111.00[b] | 112.00[a] | 114.70[c] | 34.70[c] | 93.00[c] | 60.00[e] |
| $CH_4$ | 2.00[a] | 3.00[a] | 5.10[a] | 5.61[a] | 6.00[b] | 3.40[a] | 4.40[c] | 0.40[c] | 9.59[c] | 3.40[e] |
| $NO_X$ | 3.90[a] | 3.65[a] | 2.60[a] | 1.04[a] | 0.95[b] | 1.96[a] | 4.30[c] | 2.60[c] | 2.28[c] | 3.30[e] |
| $SO_2$ | 0.90[a] | 0.68[a] | 0.40[a] | 1.10[a] | 1.00[b] | 1.10[a] | 0.44[c] | 0.22[c] | 0.18[c] | 0.85[e] |
| OC | 2.60[a] | 3.70[a] | 4.70[a] | 7.60[a] | 7.80[b] | 7.60[a] | 2.25[c] | 3.30[c] | 2.99[c] | 3.90[d] |
| BC | 0.37[a] | 1.31[a] | 0.52[a] | 0.56[a] | 0.20[b] | 0.56[a] | 0.78[d] | 0.82[d] | 0.52[d] | 0.52[d] |
| $NH_3$ | 0.56[a] | 1.20[a] | 1.30[a] | 2.47[a] | 1.80[b] | 1.17[a] | 0.68[c] | 1.00[c] | 4.10[c] | 0.37[e] |
| $NO_2$ | 3.22[a] | 2.58[a] | 3.60[a] | 2.34[a] | 0.63[b] | 2.34[a] |  | 2.99[f] |  |  |
| $PM_{2.5}$ | 7.17[a] | 7.10[a] | 9.90[a] | 15.00[a] | 18.40[b] | 17.90[a] |  | 6.43[f] |  |  |
| $PM_{10}$ | 7.20[a] | 11.4[a] | 18.50[a] | 16.97[a] | 18.40[b] | 18.40[a] |  | 7.02[f] |  |  |

All the value of C were Calculated by $CO_2$, CO, and $CH_4$.

[a] is average value from (Akagi et al., 2011).

[b] is average from (Akagi et al., 2011) and (Urbanski, 2014).

[c] is average from (Akagi et al., 2011; Fang et al., 2017; Liu et al., 2016; Santiago-De La Rosa et al., 2018; Stockwell et al., 2015).

[d] is from (Kanabkaew and Kim Oanh, 2011).

[e] is from (Cao et al., 2008).

[f] is from (Li et al., 2007).

*Line 27-29: "Moreover, notable seasonal variability characterizes the OBB carbon emissions, with marked increases observed in July and August. This surge in carbon emissions is chiefly attributed to fires in the savanna grasslands, woody savanna/shrubs, and tropical forests of SHAF, SHSA, and NHAF." The peak burning month for NHAF is in boreal winter months. How can the burning in this region contribute to the surge in carbon emissions in July and August?*

We changed the content.

"Moreover, notable seasonal variability characterizes the OBB carbon emissions, with marked increases observed in August and September, and lower emissions in winter. These carbon emissions are chiefly attributed to fires in the savanna grasslands, woody savanna/shrubs, and tropical forests of SHAF, SHSA, and NHAF."

*Line 166: "EF denotes the amount of pollutants released during burning." This seems not the correct definition or description of the emission factor (EF).*

We changed the descriptions of EF.

"EF denotes the amount of pollutants released per unit of fuel burned during burning."

*Line 181-182: "significant spatial variations in the OBB carbon emissions were observed across Africa, and certain regions in the Americas and Asia.". How do you define 'significant'? Based on Figure 1, I think the spatial variations in all continents are big.*

We changed the content.

"…obvious spatial variations in the OBB carbon emissions were observed across Africa, and certain regions in the Americas and Asia."

*Line 215: "According to GFED". Which version of GFED data are you using? Please be more specific.*

We revised according to the reviewer's comment. We added version of GFED about GFED data. We changed "GFED" to "GFED4.1s".

*Line 230: "This suggests relative homogeneity in the NHAF's biomass–burning emission sources". I don't understand how did you get this conclusion based on the previous results "In the NHAF, the predominant source of OBB was savanna grasslands (Roberts et al., 2009), contributing 76.14% to the region's total biomass–burning carbon emissions, averaging 300.21 Tg/year."*

We deleted this ambiguous expression to make it clearer to understand.

*Line 233: "...leading to increased OBB and carbon emissions in this region". In fact in this region (NHAF), the emissions from biomass burning have been decreasing during the past 2 decades.*

We deleted this ambiguous expression to make it clearer to understand.

*"…are the major factors in this region."*

*Line 257-258: "emissions from SHSA decreased at a rate of 105.22 Tg per year from 2020 to 2022, with peak monthly emissions over the 3 years reaching 184.63, 222.12, and 123.98, respectively, consistent with Griffin et al. (2023)". Griffin et al. (2023) explored the wildfire CO emissions. But it's unclear to me which part of your results is "consistent with" with that paper.*

We changed the content to make it clearer to understand.

*"…emissions from SHSA decreased at a rate of 105.22 Tg per year from 2020 to 2022, with peak monthly emissions over the 3 years reaching 184.63, 222.12, and 123.98, respectively, size and status of emissions consistent with Griffin et al. (2023)"*

*Line 259: "NHAF also exhibited a decreasing trend in annual emissions, … over the 3 years". 3 years are too short for deriving meaningful trends in annual emissions.*

We changed "NHAF also exhibited a decreasing trend in annual emissions, … over the 3 years" to "annual C emissions in NHAF also declined, …over the 3 years".

*Line 316-317: "The top three major emitting regions were SHAF, SHSA, and NHAF, which exhibited emission patterns that aligned closely with global emission trends over time". The comparison between Figure 5 and Figure 6 does not seem to support this conclusion. NHAF emissions have a very different seasonal cycle than SHAF and SHSA. The interannual variability of emissions in these regions is also different.*

We changed "The top three major emitting regions were SHAF, SHSA, and NHAF, which exhibited emission patterns that aligned closely with global emission trends over time" to "The top three major emitting regions were SHAF, SHSA, and NHAF, which were closely associated with global emission trends, representing the main source of the emission peak in August and the emission during the winter months."

*Line 379: "However, the use of FY–3D, which captures data at 14:00, was highly effective in capturing such events."*
*This is also a statement without supporting evidence. Similar to Terra and Aqua, FY-3D also records data twice a day for a given location and cannot detect short-lived fires. The local time difference between FY-3D and Aqua is only 30 minutes (13:30 vs 14:00), which won't make much difference in the ability to detect agricultural fires.*
We changed the description of FY-3D product. Please refer to Line 136.

There are many citations in this manuscript that do not support the text before the citation. It seems that the authors didn't really read and try to understand these references, but just made the citation based on some related keywords. Below is a partial list of inappropriate citations I have found. Please carefully double check the citations throughout the manuscript.

Line 39: (Hussain and Reza, 2023) is not a good citation here; it studied the detrimental impact on global health by general environmental damages, not specifically from open biomass burning. There are many studies in literature about

this topic which can be used for citation here.

Response:

We changed the references.

"and have profound impacts on the global carbon cycle, climate change, and air quality, thus exerting a significant influence on the global environment and human health (Wu et al., 2022)."

Line 40-41: (Estrellan and Iino, 2010) reviewed toxic emissions from open burning. It did not provide evidence for "major fire types worldwide". So it is also not a good citation.

Response:

We changed the references.

"Forest clearing, accidental fires, firewood burning, agricultural residue burning, peatland burning and straw burning are among the major fire types worldwide (van der Werf et al., 2017)."

Line 42: (Manisalidis et al., 2020) is a review of environmental and health impacts of air pollution. It did not talk about the specific impacts from "open burning activities".

Response:

We changed the references.

"These open burning activities severely impact air quality and ecosystems and exacerbate climate change and air pollution issues (Anon, 2017)."

Line 44: (Ma et al., 2022) studied wildfires in Amazon during 2019 only. The paper does not support the claim "regions worldwide are experiencing a notable increase in fire incidents".

Response:

We changed the references.

"…the Amazon rainforest fires (Pivello, 2011) …"

Line 45: (You and Xu, 2023) investigated how delayed wildfires in 2020 promote snowpack melting in the western US. Same as above, this paper does not support the 'increase in fire accidents'.

We changed the references.

"…wildfires in the United States (Burke et al., 2021) …"

Line 56: (Lv et al., 2020) studies CO2 mixing ratio using satellite observations. They used the GFED dataset for CO2 emissions from biomass burning. This study does not support the previous sentence "Alternatively, a method based on the fire radiative power can effectively enhance the assessment of small fire events, thereby addressing this issue to a certain extent."

We deleted it. Please refer to Line

"... For example, similar approaches have been employed in Fire Emissions and Energy Research (FEER) and the

Global Fire Assimilation System (GFAS)"

*Line 128: (Spawn and Gibbs, sssss2020). Remove the sssss here.*
We Remove the sssss here.

Line 255: (Russell-Smith et al., 2021) focus on opportunities and challenges for savanna burning emissions abatement. It did not provide sufficient evidence to support the conclusion "In August, specific meteorological conditions, such as high temperatures and low humidity facilitated the increased combustibility of biomass, resulting in a peak in carbon emissions".
We changed the references to support.
"In August, specific meteorological conditions, such as high temperatures and low humidity facilitated the increased combustibility of biomass, resulting in a peak in carbon emissions (Shea et al., 1996)."

Line 297: (Wiggins et al., 2020) presented estimates of fire emissions in the USA using data from the FIREX-AQ mission. It has little connection to the text preceding the citation.
We changed the references to support.
"typically experience high levels of OBB emissions because of the prevalence of both natural and anthropogenic fire activities (Williams et al., 2019; Zheng et al., 2021)."

Line 308: (Thackeray et al., 2022) did study the precipitation change under global warming, but the main topic of this paper was precipitation extremes. It does not support the statement in this manuscript "an overall augmentation in annual precipitation played a key role".
We changed it.
"… however, an overall augmentation in annual precipitation led to a reduction in the degree of drought (Thackeray et al., 2022; Zhang et al., 2023a)."

*There are also many cases where the presentation is poorly structured, vague, or inconsistent.*
*Line 23-26: The presentations of region names within the parentheses are inconsistent; the full name is shown for some regions, but not shown for other regions.*
We added full name for other regions.
"72.71 (Boreal North America; BONA), 165.7 (Temperate North America, TENA), 34.1 (Central America; CEAM), 42.9 (Northern Hemisphere South America; NHSA), 520.5 (Southern Hemisphere South America; SHSA), 13 (Europe; EURO), 8.4 (Middle East; MIDE), 394.3 (Northern Hemisphere Africa; NHAF), 847 (Southern Hemisphere Africa; SHAF), 167.4 (Boreal Asia; BOAS), 27.9 (Central Asia; CEAS), 197.3 (Southeast Asia; SEAS), 13.2 (Equatorial Asia; EQAS), and 82.4 (Australia and New Zealand; AUST) Tg".

*Line 27-28: "...notable seasonal variability characterizes the OBB carbon emissions, with marked increases observed*

*in July and August." Although I understand the meaning of this sentence, it is not well organized. For example, what is the object of comparison when you say 'marked increase'?*

We changed "...notable seasonal variability characterizes the OBB carbon emissions, with marked increases observed in July and August." to "... notable seasonal variability characterizes the OBB carbon emissions, with marked increases observed in August and September (annual average 441.32 Tg C) compared to other months (annual average 170.42 Tg C)."

*Line 41-42: "These open burning activities severely impact air quality and ecosystems and exacerbate climate change and air pollution issues." In this sentence "severely impact air quality" and "exacerbate…air pollution" are basically referring to the same thing.*

We changed it.

"These open burning activities severely impact air quality and ecosystems ..."

*Line 46-47: "These fires release substantial amounts of harmful particulate matter and organic pollutants, posing serious threats to air quality and potentially causing health problems". I don't understand why this sentence is here. Does it represent the same meaning as the first sentence in this paragraph?*

We deleted this ambiguous expression to make it clearer to understand.

*Line 51: "The burned area method…". I believe most readers don't know what the 'burned area method' is. A short definition or introduction to this method needs to be presented here.*

We added a short introduction to this method.

"The burned area method demonstrated good accuracy in quantifying larger fire events, which is based on the burned area, the available biomass fuels burned in fields, the fuel-related combustion efficiency, and emission factors."

*Line 52-53: "Shi et al. (2020) estimated OBB emissions in tropical continents from 2001 to 2017 using widely used inventory data, such as the Global Fire Emissions Database (GFED) and the Fire INventory from NCAR (FINN)". I don't think Shi et al. (2020) estimated OBB emissions using GFED and FINN, since GFED and FINN are themselves global emissions datasets.*

We changed this ambiguous expression to make it clearer to understand.

"Shi et al. (2020) estimated OBB emissions in tropical continents from 2001 to 2017. As well as other open-access databases, such as the Global Fire Emissions Database (GFED) and the Fire INventory from NCAR (FINN)"

*Line 103: "... exhibited superior judgment accuracy". What is 'judgment accuracy' referring to?*

We changed the description of FY-3D product. Please refer to Line 136.

*Line 117-118: "In contrast, satellite data cover the entire globe and provide surface parameters, thereby enabling biomass estimation." This is a potentially confusing sentence; Ground observations can also "provide surface*

*parameters and enables biomass estimation".*

We changed it.

"In contrast, satellite data cover the entire globe and provide surface parameters worldwide, thereby enabling biomass estimation."

*Line 126-128: "Global AGB for other years was generated based on the global aboveground and belowground biomass carbon density maps for the 2010 product". While I now understand the method by reading the SI, the sentence is not very clear in its current form. It's better to day that in 2010 the Spawn and Gibbs product was used and then say that in other years the AGB was estimated using a scalar based on TC and NDVI. BTW, AGB stands for "above ground biomass"; how did you derive the 'below ground' biomass?*

We changed it.

"we combined the "global aboveground and belowground biomass carbon density maps for the 2010" product provided by Spawn and Gibbs(2020), annual TC, and NDVI data, and obtained by linear stretching the fuel loading for other years"

The below ground biomass is also provided by "global aboveground and belowground biomass carbon density maps for the 2010" product.

*Line 136: "the subsurface condition" should mean the below ground condition, but I suspect that you are referring to 'surface condition' here.*

We changed "the subsurface condition" to "surface condition".

*Line 171-172: "the EF for the following seven land types were updated". It's not clear to me what original EF data were used and what data were used to replace (update) them.*

We have changed the previous expression.

"…the EF for the following seven land types of other database were updated:"

We also added the reference of the EF data in Table 2.

*Line 178-181: Please combine/simplify these three sentences.*

We combined these sentences.

"Taking carbon as an example, the annual carbon emissions from OBB were estimated for the period of 2020–2022 (Figure 1) and the total OBB carbon emissions reached 7760.63 Tg C."

*Line 261: "Cumulatively, these territories represent…". What are "these territories". Based on the previous paragraph, they should probably include SHAF and NHAF. But these should be explicitly stated.*

We changed "these territories" to "SHAF, SHSA, and NHAF ...".

*Line 317: "Over the past 3 years". The 'past 3 years' can change depending on the reference year. This kind of*

*description should be more specific.*

We changed "Over the past 3 years" to "During 2020 to 2022".

*Line 427: What are "substrate types"?*

We changed incorrect description.

"To address the varying fire conditions, we performed a detailed subdivision based on different fire types."

*There are other minor issues, including potential errors or typos*

*Line 60: "MEIRSI–2" should be "MERSI-2"*

We corrected it.

*Figure 2: If these geographical regions are the same to that in GFED, you probably need to acknowledge/cite the GFED group/paper.*

We've added citation information.

*Line 269: "intensify both the frequency and frequency of fires in the area". One 'frequency' should be removed or changed to other words.*

We removed it.

*Line 280: "in the Tropical Eastern North America (TENA) region". As shown in Figure 2, TENA should be 'Temperate North America'.*

We changed "in the Tropical Eastern North America (TENA) region" to "in the Temperate North America (TENA) region".

*Line 435-436: "Although the FY–3D GFR dataset is reliable for most OBB events, its resolution of 1 degree…"*
*Shouldn't the resolution of FY-3D GFR dataset 1 km?*

We changed "1 degree" to "1 km".

**References**

Abbasi, B., Qin, Z., Du, W., Fan, J., Zhao, C., Hang, Q., Zhao, S., and Li, S.: An Algorithm to Retrieve Total Precipitable Water Vapor in the Atmosphere from FengYun 3D Medium Resolution Spectral Imager 2 (FY-3D MERSI-2) Data, Remote Sensing, 12, 3469, https://doi.org/10.3390/rs12213469, 2020.

Akagi, S. K., Yokelson, R. J., Wiedinmyer, C., Alvarado, M. J., Reid, J. S., Karl, T., Crounse, J. D., and Wennberg, P. O.: Emission factors for open and domestic biomass burning for use in atmospheric models, Atmospheric Chemistry and Physics, 11, 4039–4072, https://doi.org/10.5194/acp-11-4039-2011, 2011.

Archibald, S., Roy, D. P., Van WILGEN, B. W., and Scholes, R. J.: What limits fire? An examination of drivers of burnt area in Southern Africa, Global Change Biology, 15, 613–630, https://doi.org/10.1111/j.1365-2486.2008.01754.x, 2009.

Cao, G., Zhang, X., Wang, Y., and Zheng, F.: Estimation of emissions from field burning of crop straw in China, Chin. Sci. Bull., 53, 784–790, https://doi.org/10.1007/s11434-008-0145-4, 2008.

Chen, J., Yao, Q., Chen, Z., Li, M., Hao, Z., Liu, C., Zheng, W., Xu, M., Chen, X., Yang, J., Lv, Q., and Gao, B.: The Fengyun-3D (FY-3D) global active fire product: principle, methodology and validation, Earth System Science Data, 14, 3489–3508, https://doi.org/10.5194/essd-14-3489-2022, 2022.

Dong, Z., Yu, J., An, S., Zhang, J., Li, J., and Xu, D.: Forest Fire Detection of FY-3D Using Genetic Algorithm and Brightness Temperature Change, Forests, 13, 963, https://doi.org/10.3390/f13060963, 2022.

Fang, Z., Deng, W., Zhang, Y., Ding, X., Tang, M., Liu, T., Hu, Q., Zhu, M., Wang, Z., Yang, W., Huang, Z., Song, W., Bi, X., Chen, J., Sun, Y., George, C., and Wang, X.: Open burning of rice, corn and wheat straws: primary emissions, photochemical aging, and secondary organic aerosol formation, Atmospheric Chemistry and Physics, 17, 14821–14839, https://doi.org/10.5194/acp-17-14821-2017, 2017.

Freitas, S. R., Longo, K. M., Silva Dias, M. A. F., Silva Dias, P. L., Chatfield, R., Prins, E., Artaxo, P., Grell, G. A., and Recuero, F. S.: Monitoring the transport of biomass burning emissions in South America, Environ Fluid Mech, 5, 135–167, https://doi.org/10.1007/s10652-005-0243-7, 2005.

Giglio, L.: Characterization of the tropical diurnal fire cycle using VIRS and MODIS observations, Remote Sensing of Environment, 108, 407–421, https://doi.org/10.1016/j.rse.2006.11.018, 2007.

Giglio, L., Randerson, J. T., van der Werf, G. R., Kasibhatla, P. S., Collatz, G. J., Morton, D. C., and DeFries, R. S.: Assessing variability and long-term trends in burned area by merging multiple satellite fire products, Biogeosciences, 7, 1171–1186, https://doi.org/10.5194/bg-7-1171-2010, 2010.

Kanabkaew, T. and Kim Oanh, N. T.: Development of Spatial and Temporal Emission Inventory for Crop Residue Field

Burning, Environ Model Assess, 16, 453–464, https://doi.org/10.1007/s10666-010-9244-0, 2011.

Le Page, Y., Pereira, J. M. C., Trigo, R., da Camara, C., Oom, D., and Mota, B.: Global fire activity patterns (1996–2006) and climatic influence: an analysis using the World Fire Atlas, Atmospheric Chemistry and Physics, 8, 1911–1924, https://doi.org/10.5194/acp-8-1911-2008, 2008.

Li, S. and Banerjee, T.: Spatial and temporal pattern of wildfires in California from 2000 to 2019, Sci Rep, 11, 8779, https://doi.org/10.1038/s41598-021-88131-9, 2021.

Li, X., Wang, S., Duan, L., Hao, J., Li, C., Chen, Y., and Yang, L.: Particulate and Trace Gas Emissions from Open Burning of Wheat Straw and Corn Stover in China, Environ. Sci. Technol., 41, 6052–6058, https://doi.org/10.1021/es0705137, 2007.

Liu, X., Zhang, Y., Huey, L. G., Yokelson, R. J., Wang, Y., Jimenez, J. L., Campuzano-Jost, P., Beyersdorf, A. J., Blake, D. R., Choi, Y., St. Clair, J. M., Crounse, J. D., Day, D. A., Diskin, G. S., Fried, A., Hall, S. R., Hanisco, T. F., King, L. E., Meinardi, S., Mikoviny, T., Palm, B. B., Peischl, J., Perring, A. E., Pollack, I. B., Ryerson, T. B., Sachse, G., Schwarz, J. P., Simpson, I. J., Tanner, D. J., Thornhill, K. L., Ullmann, K., Weber, R. J., Wennberg, P. O., Wisthaler, A., Wolfe, G. M., and Ziemba, L. D.: Agricultural fires in the southeastern U.S. during SEAC4RS: Emissions of trace gases and particles and evolution of ozone, reactive nitrogen, and organic aerosol, Journal of Geophysical Research: Atmospheres, 121, 7383–7414, https://doi.org/10.1002/2016JD025040, 2016.

Santiago-De La Rosa, N., González-Cardoso, G., Figueroa-Lara, J. de J., Gutiérrez-Arzaluz, M., Octaviano-Villasana, C., Ramírez-Hernández, I. F., and Mugica-Álvarez, V.: Emission factors of atmospheric and climatic pollutants from crop residues burning, Journal of the Air & Waste Management Association, 68, 849–865, https://doi.org/10.1080/10962247.2018.1459326, 2018.

Shi, Y. and Yamaguchi, Y.: A high-resolution and multi-year emissions inventory for biomass burning in Southeast Asia during 2001–2010, Atmospheric Environment, 98, 8–16, https://doi.org/10.1016/j.atmosenv.2014.08.050, 2014.

Stockwell, C. E., Veres, P. R., Williams, J., and Yokelson, R. J.: Characterization of biomass burning emissions from cooking fires, peat, crop residue, and other fuels with high-resolution proton-transfer-reaction time-of-flight mass spectrometry, Atmospheric Chemistry and Physics, 15, 845–865, https://doi.org/10.5194/acp-15-845-2015, 2015.

Urbanski, S.: Wildland fire emissions, carbon, and climate: Emission factors, Forest Ecology and Management. 317: 51-60., 51–60, https://doi.org/10.1016/j.foreco.2013.05.045, 2014.

---

## Author Response (AR4)

This study developed a global daily emission inventory of OBB with 1km×1km based on global fire point monitoring data from the Chinese Fengyun-3D satellite, fuel loading, combustion factor and emission factor. Considering the scientific impact of each study, several analysis is needed to substantiate the conclusions in your manuscript. Firstly, the manuscript emphasizes that the compared with MODIS, significant advantage of using the FY–3D fire detection product is the ability to enhance the detection of small fires, but the analysis of the results does not show how much the use of the FY–3D detection product has increased the emission estimates of small fires? Secondly, in the section about verification, the manuscript emphasizes the consistency with other datasets, but does not quantify the advantages of this study. Thirdly, the advance of activity data selected in this study needs to be verified, such as the quality and the resolution of the data. The manuscript can be considered for publication if the issues mentioned above and following specific comment could be addressed.

We thank the Reviewer for the constructive comments and suggestions. We shall revise the manuscript accordingly, and we address the comments as follows.

**Specific comments:**

*P1 line23–25: The full name are not given for some regions (e.g., BONA), and them are given for some regions (e.g., SHSA).*

Response: We added full name for other regions.

"72.71 (Boreal North America; BONA), 165.7 (Temperate North America, TENA), 34.1 (Central America; CEAM), 42.9 (Northern Hemisphere South America; NHSA), 520.5 (Southern Hemisphere South America; SHSA), 13 (Europe; EURO), 8.4 (Middle East; MIDE), 394.3 (Northern Hemisphere Africa; NHAF), 847 (Southern Hemisphere Africa; SHAF), 167.4 (Boreal Asia; BOAS), 27.9 (Central Asia; CEAS), 197.3 (Southeast Asia; SEAS), 13.2 (Equatorial Asia; EQAS), and 82.4 (Australia and New Zealand; AUST) Tg.".

*P2 line64–65: The detection accuracy of MODIS and other related indexes should be clearly given to facilitate readers to compare directly. As well as the comparison with MODIS, other commonly used polar–orbiting satellite sensors (SNPP–VIIRS, Landsat–8, etc.) can be considered for comparison to highlight the advantages of FY–3D.*

Response: We revised according to the reviewer's comment. We added comparisons with other product. The changes are as follows.

"Furthermore, the Global Fire Monitoring (GFR) product with FY–3D employs optimized automatic identification algorithms for fire spots (Shan and Zheng, 2022), leading to an improved accuracy of fire point detection. This resulted in an impressive overall accuracy rate of 79.43% and an exclusion omission error accuracy of 88.50%, surpassing the capabilities of MODIS satellite products (Chen et al., 2022; Xian et al., 2021), based on field–collected references throughout 2020 in China. The cross–verification between MODIS and FY–3D shows the highest consistency results (over 80%) in Africa and Asia, while America, Europe, and Oceania demonstrate consistency exceeding 70% (Chen et al., 2022). In July, August, and September,

the number of fire spots was higher, with a mean consistency of over 85% between MODIS and FY–3D fire products (Chen et al., 2022). Although Landsat Fire and Thermal Anomaly (LFTA) product has finer spatial resolution, its lower temporal resolution typically allows global coverage only every 16 days, which does not allow for frequent detection of biomass burning activity. Therefore, employing the FY–3D GFR product and allocation approaches for small fires is expected to yield reliable estimates of OBB emissions.".

Furthermore, we added the comparison of parameters related to MERSI–II, MODIS, and VIIRS. Please refer to Table 1.

*P4 line128: It is suggested that formulas could be transferred to the manuscript from SI, with the supplement of corresponding unit of the variable.*

Response: We revised according to the reviewer's comment. We transferred the formulas to the manuscript from SI, with the supplement of corresponding unit of the variable. Please refer to Line 145.

*P5 line148: Source of the constant 0.013 in the formula? Empirical values should give literature. The fitted values should depict the fitting process and significance test results.*

Response: We revised according to the reviewer's comment. We added the reference to the constant 0.013 in the formula.

*P5 line152: There are other products (MODIS) of NDVI with a time interval of 8d. Why the products of 16d was selected in this study?*

Response: The 8d product is based on mod09 calculations and because it is measured in the absence of atmospheric scattering or absorption, the product contains lower data, as well as cloud cover. The 16–day product was derived from the 8–day product, offering advantages such as an enhanced signal–to–noise ratio and reduced cloud contamination. This is achieved through a longer temporal composite period, leading to more dependable and precise assessments of vegetation health and productivity.

In the process of assessing vegetation conditions, the accuracy and completeness of the data are prioritized over the requirement for event resolution. Therefore, we opt to utilize the 16–day product.

*P6 Table 1: The EF for specific biomass (e.g., crop) is fixed value for different regions with various crop distribution characteristic. Regional or crop differences should be reflected in EF values.*

Response: We made a distinction in the CF for different types of fire event emissions in different regions, and the EF was only used as a factor between the pollutant emitted and the type of fire.

**Table 2. Emission factor (g/kg) of different species.**

| Species | Grasslands and Savannas | Woody Savanna or Shrubs | Tropical Forest | Temperate Forest | Boreal Forest | Temperate Evergreen Forest | Crop | | | |
|---|---|---|---|---|---|---|---|---|---|---|
| | | | | | | | Maize | Sugar | Sugar | Wheat |
| C | 488.31 | 489.41 | 491.77 | 468.31 | 478.88 | 493.18 | 687.09 | 323.35 | 368.04 | 429.17 |

| | | | | | | | | | | |
|---|---|---|---|---|---|---|---|---|---|---|
| $CO_2$ | 1,686[a] | 1,681[a] | 1,643[a] | 1,510[a] | 1,565[b] | 1,623[a] | 2,327[c] | 1,130[c] | 1,177[c] | 1,470[e] |
| CO | 63.00[a] | 67.00[a] | 93.00[a] | 122.00[a] | 111.00[b] | 112.00[a] | 114.70[c] | 34.70[e] | 93.00[c] | 60.00[e] |
| $CH_4$ | 2.00[a] | 3.00[a] | 5.10[a] | 5.61[a] | 6.00[b] | 3.40[a] | 4.40[c] | 0.40[c] | 9.59[c] | 3.40[e] |
| $NO_X$ | 3.90[a] | 3.65[a] | 2.60[a] | 1.04[a] | 0.95[b] | 1.96[a] | 4.30[c] | 2.60[c] | 2.28[c] | 3.30[e] |
| $SO_2$ | 0.90[a] | 0.68[a] | 0.40[a] | 1.10[a] | 1.00[b] | 1.10[a] | 0.44[c] | 0.22[c] | 0.18[c] | 0.85[e] |
| OC | 2.60[a] | 3.70[a] | 4.70[a] | 7.60[a] | 7.80[b] | 7.60[a] | 2.25[c] | 3.30[c] | 2.99[c] | 3.90[d] |
| BC | 0.37[a] | 1.31[a] | 0.52[a] | 0.56[a] | 0.20[b] | 0.56[a] | 0.78[d] | 0.82[d] | 0.52[d] | 0.52[d] |
| $NH_3$ | 0.56[a] | 1.20[a] | 1.30[a] | 2.47[a] | 1.80[b] | 1.17[a] | 0.68[c] | 1.00[c] | 4.10[c] | 0.37[e] |
| $NO_2$ | 3.22[a] | 2.58[a] | 3.60[a] | 2.34[a] | 0.63[b] | 2.34[a] | | 2.99[f] | | |
| $PM_{2.5}$ | 7.17[a] | 7.10[a] | 9.90[a] | 15.00[a] | 18.40[b] | 17.90[a] | | 6.43[f] | | |
| $PM_{10}$ | 7.20[a] | 11.4[a] | 18.50[a] | 16.97[a] | 18.40[b] | 18.40[a] | | 7.02[f] | | |

All the value of C were Calculated by $CO_2$, CO, and $CH_4$.

[a] is average value from (Akagi et al., 2011).

[b] is average from (Akagi et al., 2011) and (Urbanski, 2014).

[c] is average from (Akagi et al., 2011; Fang et al., 2017; Liu et al., 2016; Santiago-De La Rosa et al., 2018; Stockwell et al., 2015).

[d] is from (Kanabkaew and Kim Oanh, 2011).

[e] is from (Cao et al., 2008).

[f] is from (Li et al., 2007).

*P11 line269: What does "intensify both the frequency and frequency of fires in the area" mean?*

Response: We changed "intensify both the frequency and frequency of fires in the area" to "intensify frequency of fires in the area" make it clear to understand.

*P15 line352: Why the dataset is not include FINN (e.g., FINNv2.5)? The resolution of it is the same with the dataset developed in this study (1km, 1d).*

Response: The analytical discussion in this study is based on the example of Carbon, for which no estimation exists in FINN (FINNv2.5, 0.1 degree), so there is no FINN product. In SI, we added the comparison results with FINN for $CO_2$ analysis in different regions.

*P16 line377–379: There is a lack of clarity in the explanation of how FY–3D can capture small fires more effectively compared to MODIS, and how the difference in transit times between the two satellites affects the detection of agricultural small fires. More data analysis is needed to support this question.*

*P18 line410: The article should add a comparative analysis of how much the addition of FY–3D improves emission estimates for small fires, which is a key factor in determining the innovativeness of the study.*

Response: We added Comparison of parameters related to MERSI–II, MODIS, and VIIRS to explain the advantages of the

FY–3D for fire point detection at the hardware level. Please refer to Table 1. Furthermore, we added more details about FY–3D GFR product to support FY–3D can capture small fires more effectively.

**Table 1. Comparison of parameters related to MERSI–II, MODIS, and VIIRS.**

| | MERSI–II (FY–3D) | MODIS (AQUA) | VIIRS ((NOAA–20)) |
|---|---|---|---|
| Orbit altitude (km) | 836 | 705 | 824 |
| Equator Crossing time | 14:00 LT | 13:30 LT | 14:20 LT |
| Swath (km) | 2900 | 2330 | 3060 |
| Pixel resolution at nadir (km) | 1 | 1 | 0.75/0.375 |
| Pixel resolution at the edge (km) | >6 | 4 | 1.5/0.75 |
| ID MIR Band (s) | 21 | 21/22 | M–13/I–4 |
| Spectral range (μm) | 3.973–4.128 | 3.929–3.989 3.940–4.001 | 3.973–4.128 3.550–3.930 |
| TMAX (SNR–NEΔT on orbit) | 380 K (0.25) | 500 K (0.183) 331 K (0.019) | 634 K (0.04) |
| ID TIR Band (s) | 24 | 31 | M–15/I–5 |
| Spectral range (μm) | 10.300–11.300 | 10.780–11.280 | 10.263–11.263 10.500–12.400 |
| TMAX (SNR–NEΔT on orbit) | 330 K (0.4) | 400 K (0.017) | 343 K (0.03) |

"…Compared to MODIS, FY–3D fire products have been optimized in terms of auxiliary parameters, fire identification and re–identification. Firstly, FY–3D introduces the adaptive threshold and eliminates the limitations by fixed thresholds of MODIS and VIIRS algorithms by automatic identification algorithms for fire spot detection (Chen et al., 2022). Secondly, FY–3D uses a re–identification index reflecting varying geographical latitude and underlying surfaces types, together with the effect by cloud, water, and bare land (Zheng et al., 2020). The integration of multiple influencing factors increases the accuracy of fire detection. For example, the influences of factory thermal anomalies and high reflectance of photovoltaic power plants are greatly removed. Finally, the far–infrared channel employed in FY–3D has a high resolution of 250 m, higher than MODIS with 1 km, resulting in higher accuracy in big fire detection (Zheng and Chen, 2020). Overall, the FY–3D GFR product achieves an accuracy of 94.0% globally, with accuracies of 94.6%, 94.1%, 90.6%, 91.8%, and 92.7% in South–central Africa, East central South America, Siberia, Australia and Indochinese Peninsula (Chen et al., 2022), respectively. Specifically, due to the removal of underlying surface interference in China, the FY–3D achieves accuracies of 79.43% and 88.50% for accuracy and accuracy without omission, respectively, both of which are higher than the accuracies of 74.23% and 79.69% achieved by MODIS (Chen et al., 2022).".

*P19 line435: What is the difference between the 1° spatial resolution of FY–3D mentioned here and the 1km mentioned previously (line 106)?*

Response: We changed "1 degree" to "1 km".

**Anonymous Referee #2 of Revised submission**

The authors claim that "the GFR product, which was integrated with the MERSI–2 instrument, exhibited superior judgment accuracy" (Line 102–103), "Consequently, our inventory yielded accurate assessment results and captured the spatial variation and heterogeneity of minor OBB emissions effectively" (Line 358–359), and "the accuracy of the OBB carbon emissions assessment significantly improved" (Line 417–418). In my opinion, these claims are not sufficiently justified in the manuscript. First, the superiority of MERSI–2 over MODIS in detecting active fires is not well explained. MERSI–2 on FY–3D has higher spatial resolution than MODIS in the visible and NIR bands. However, the active fire algorithm mainly uses the mid–infrared band, where both MERSI–2 and MODIS have a spatial resolution of 1km. The authors cited a number of previous papers (such as Dong et al. 2022 and Chen et al., 2022) to show better fire detection accuracy from MERSI–2 than from MODIS. However, these studies were mostly based on comparisons with limited data samples from manual inspection, and are not very convincing to me. Second, there are many limitations in the algorithm that are not mentioned in the manuscript. For example, this study used AGB as the fuel load, completely ignoring the emissions from soil organic matter burning. The omission error of active fires due to cloud cover/thick smoke is also not quantified. Third, this emissions dataset was derived from FY–3D active fires, but many MODIS products are still needed to generate GEIOBB. The use of MODIS products, which include MOD44B, MODIS NDVI, and MODIS land cover type data, may hinder the effect of quantifying global fire emissions after Terra and Aqua are gone. This potential problem should also be addressed in this paper.

We thank the Reviewer for the constructive comments and suggestions. We shall revise the manuscript accordingly, and we address the comments as follows.

1. We added more details about FY–3D to support FY–3D can capture small fires more effectively. Please refer to line 109 and Table 1.

2. Soil organic matter is also a contributor to the total OBB. However, a very small proportion of emissions can be found in most forest and grassland except in peatland burning in Indonesia. After calculation, we found the carbon emissions in Indonesia was very low since the drought and human induced fires detected were very low during 2020–2022.

   Additionally, we analyzed the omission error of active fires due to cloud cover/thick smoke.

   "The detected active fires were also underestimated due to cloud cover/thick smoke, with an omission error of approximately from 10%–30% (Schroeder et al., 2008; Roberts et al., 2009; Giglio et al., 2006)."

3. In this study, MODIS provide globally commonly used available products including the tree cover, land cover, NDVI to generate OBB emissions worldwide. Other available high–resolution and multi–year products will be used in our study to produce more reliable emission inventory.

**Specific comments:**

*Many statements are incorrect or lacking scientific support.*

*The estimation method and the use of fuel loading (F) are not clearly described. While the authors mention in the*

*manuscript that three data sources, NDVI, TC, and AGB are used for fuel loading, the approach for combining different data streams and forming the fuel loading is embedded in the supporting text only. This formula was presented without any scientific justification or explanation (there are also some errors in the description of this formula, e.g., 2020 should be corrected to 2010).*

Response: We revised according to the reviewer's comment. We added a description of fuel compliance with fuel loading in the main text and added references related to the formula. We corrected the misdescription in the formula. Please refer to line 145.

"AGB shows a large linear correlation with TC and NDVI (Xingcheng et al., 2017), so we combined the global aboveground and belowground biomass carbon density maps for the 2010 product (https://daac.ornl.gov/cgi-bin/dsviewer.pl?ds_id=1763) provided by Spawn and Gibbs(2020), annual TC, and NDVI data, and obtained by linear stretching the fuel loading for other years.

$$F(x,t) = \left(\frac{NDVI_{now}}{NDVI_{2010}} + \frac{TC_{now}}{TC_{2010}}\right) * AGB \qquad (2)$$

Where $NDVI_{now}$ is the mean value of the month before a single fire event, $NDVI_{2010}$ is the mean value of NDVI in 2020, $TC_{now}$ is the tree cover in the year of the fire incident, $TC_{2010}$ is the tree cover in 2020, and AGB is the Above Ground Biomass data in 2010.".

*The use of emission factors (EF) is also ambiguously described in the manuscript. In section 2.4, the authors simply listed a table of EFs without indicating the specific data sources. Although references to various studies and some locally measured data are cited, the specific methodology employed to construct Table 1 remains undisclosed.*

Response: We add data sources for EF in Table 1. Please refer to Table 2.

**Table 2. Emission factor (g/kg) of different species.**

| Species | Grasslands and Savannas | Woody Savanna or Shrubs | Tropical Forest | Temperate Forest | Boreal Forest | Temperate Evergreen Forest | Crop | | | |
|---|---|---|---|---|---|---|---|---|---|---|
| | | | | | | | Maize | Sugar | Sugar | Wheat |
| C | 488.31 | 489.41 | 491.77 | 468.31 | 478.88 | 493.18 | 687.09 | 323.35 | 368.04 | 429.17 |
| CO$_2$ | 1,686[a] | 1,681[a] | 1,643[a] | 1,510[a] | 1,565[b] | 1,623[a] | 2,327[c] | 1,130[c] | 1,177[c] | 1,470[e] |
| CO | 63.00[a] | 67.00[a] | 93.00[a] | 122.00[a] | 111.00[b] | 112.00[a] | 114.70[c] | 34.70[c] | 93.00[c] | 60.00[e] |
| CH$_4$ | 2.00[a] | 3.00[a] | 5.10[a] | 5.61[a] | 6.00[b] | 3.40[a] | 4.40[c] | 0.40[c] | 9.59[c] | 3.40[e] |
| NO$_X$ | 3.90[a] | 3.65[a] | 2.60[a] | 1.04[a] | 0.95[b] | 1.96[a] | 4.30[c] | 2.60[c] | 2.28[c] | 3.30[e] |
| SO$_2$ | 0.90[a] | 0.68[a] | 0.40[a] | 1.10[a] | 1.00[b] | 1.10[a] | 0.44[c] | 0.22[c] | 0.18[c] | 0.85[e] |
| OC | 2.60[a] | 3.70[a] | 4.70[a] | 7.60[a] | 7.80[b] | 7.60[a] | 2.25[c] | 3.30[c] | 2.99[c] | 3.90[d] |
| BC | 0.37[a] | 1.31[a] | 0.52[a] | 0.56[a] | 0.20[b] | 0.56[a] | 0.78[d] | 0.82[d] | 0.52[d] | 0.52[d] |
| NH$_3$ | 0.56[a] | 1.20[a] | 1.30[a] | 2.47[a] | 1.80[b] | 1.17[a] | 0.68[c] | 1.00[c] | 4.10[c] | 0.37[e] |

| | | | | | | | |
|---|---|---|---|---|---|---|---|
| NO$_2$ | 3.22[a] | 2.58[a] | 3.60[a] | 2.34[a] | 0.63[b] | 2.34[a] | 2.99[f] |
| PM$_{2.5}$ | 7.17[a] | 7.10[a] | 9.90[a] | 15.00[a] | 18.40[b] | 17.90[a] | 6.43[f] |
| PM$_{10}$ | 7.20[a] | 11.4[a] | 18.50[a] | 16.97[a] | 18.40[b] | 18.40[a] | 7.02[f] |

All the value of C were Calculated by $CO_2$, CO, and $CH_4$.

[a] is average value from (Akagi et al., 2011).

[b] is average from (Akagi et al., 2011) and (Urbanski, 2014).

[c] is average from (Akagi et al., 2011; Fang et al., 2017; Liu et al., 2016; Santiago-De La Rosa et al., 2018; Stockwell et al., 2015).

[d] is from (Kanabkaew and Kim Oanh, 2011).

[e] is from (Cao et al., 2008).

[f] is from (Li et al., 2007).

*Line 27–29: "Moreover, notable seasonal variability characterizes the OBB carbon emissions, with marked increases observed in July and August. This surge in carbon emissions is chiefly attributed to fires in the savanna grasslands, woody savanna/shrubs, and tropical forests of SHAF, SHSA, and NHAF." The peak burning month for NHAF is in boreal winter months. How can the burning in this region contribute to the surge in carbon emissions in July and August?*

Response: We changed the content.

"Moreover, notable seasonal variability characterizes the OBB carbon emissions, with marked increases observed in August and September, and lower emissions in winter. These carbon emissions are chiefly attributed to fires in the savanna grasslands, woody savanna/shrubs, and tropical forests of SHAF, SHSA, and NHAF.".

*Line 166: "EF denotes the amount of pollutants released during burning." This seems not the correct definition or description of the emission factor (EF).*

Response: We changed the descriptions of EF.

"EF denotes the amount of pollutants released per unit of fuel burned during burning.".

*Line 181–182: "significant spatial variations in the OBB carbon emissions were observed across Africa, and certain regions in the Americas and Asia.". How do you define 'significant'? Based on Figure 1, I think the spatial variations in all continents are big.*

Response: We changed the content.

"…obvious spatial variations in the OBB carbon emissions were observed across Africa, and certain regions in the Americas and Asia.".

*Line 215: "According to GFED". Which version of GFED data are you using? Please be more specific.*

Response: We revised according to the reviewer's comment. We changed "GFED" to "GFED4.1s".

*Line 230: "This suggests relative homogeneity in the NHAF's biomass–burning emission sources". I don't understand how did you get this conclusion based on the previous results "In the NHAF, the predominant source of OBB was savanna grasslands (Roberts et al., 2009), contributing 76.14% to the region's total biomass–burning carbon emissions, averaging 300.21 Tg/year."*
Response: We deleted this ambiguous expression to make it clearer to understand.

*Line 233: "...leading to increased OBB and carbon emissions in this region". In fact in this region (NHAF), the emissions from biomass burning have been decreasing during the past 2 decades.*
Response: We deleted this ambiguous expression to make it clearer to understand.
"…are the major factors in this region.".

*Line 257–258: "emissions from SHSA decreased at a rate of 105.22 Tg per year from 2020 to 2022, with peak monthly emissions over the 3 years reaching 184.63, 222.12, and 123.98, respectively, consistent with Griffin et al. (2023)". Griffin et al. (2023) explored the wildfire CO emissions. But it's unclear to me which part of your results is "consistent with" with that paper.*
Response: We changed the content to make it clearer to understand.
"…emissions from SHSA decreased at a rate of 105.22 Tg per year from 2020 to 2022, with peak monthly emissions over the 3 years reaching 184.63, 222.12, and 123.98, respectively, size and status of emissions consistent with Griffin et al. (Griffin et al., 2023)"

*Line 259: "NHAF also exhibited a decreasing trend in annual emissions, ... over the 3 years". 3 years are too short for deriving meaningful trends in annual emissions.*
Response: We changed "NHAF also exhibited a decreasing trend in annual emissions, … over the 3 years" to "annual C emissions in NHAF also declined, …over the 3 years".

*Line 316–317: "The top three major emitting regions were SHAF, SHSA, and NHAF, which exhibited emission patterns that aligned closely with global emission trends over time". The comparison between Figure 5 and Figure 6 does not seem to support this conclusion. NHAF emissions have a very different seasonal cycle than SHAF and SHSA. The interannual variability of emissions in these regions is also different.*
Response: We changed "The top three major emitting regions were SHAF, SHSA, and NHAF, which exhibited emission patterns that aligned closely with global emission trends over time" to "The top three major emitting regions were SHAF, SHSA, and NHAF, which were closely associated with global emission trends, representing the main source of the emission peak in August and the emission during the winter months.".

*Line 379: "However, the use of FY–3D, which captures data at 14:00, was highly effective in capturing such events." This is also a statement without supporting evidence. Similar to Terra and Aqua, FY–3D also records data twice a day*

*for a given location and cannot detect short–lived fires. The local time difference between FY–3D and Aqua is only 30 minutes (13:30 vs 14:00), which won't make much difference in the ability to detect agricultural fires.*

Response: We changed the description of FY–3D product.

"Compared to MODIS, FY–3D fire products have been optimized in terms of auxiliary parameters, fire identification and re–identification. Firstly, FY–3D introduces the adaptive threshold and eliminates the limitations by fixed thresholds of MODIS and VIIRS algorithms by automatic identification algorithms for fire spot detection (Chen et al., 2022). Secondly, FY–3D uses a re–identification index reflecting varying geographical latitude and underlying surfaces types, together with the effect by cloud, water, and bare land (Zheng et al., 2020). The integration of multiple influencing factors increases the accuracy of fire detection. For example, the influences of factory thermal anomalies and high reflectance of photovoltaic power plants are greatly removed. Finally, the far–infrared channel employed in FY–3D has a high resolution of 250 m, higher than MODIS with 1 km, resulting in higher accuracy in big fire detection (Zheng and Chen, 2020). Overall, the FY–3D GFR product achieves an accuracy of 94.0% globally, with accuracies of 94.6%, 94.1%, 90.6%, 91.8%, and 92.7% in South–central Africa, East central South America, Siberia, Australia and Indochinese Peninsula (Chen et al., 2022), respectively. Specifically, due to the removal of underlying surface interference in China, the FY-3D achieves accuracies of 79.43% and 88.50% for accuracy and accuracy without omission, respectively, both of which are higher than the accuracies of 74.23% and 79.69% achieved by MODIS (Chen et al., 2022).".

And, we Compared the parameters related to MERSI–II, MODIS, and VIIRS in Table 1.

**Table 1. Comparison of parameters related to MERSI–II, MODIS, and VIIRS.**

| | MERSI–II (FY–3D) | MODIS (AQUA) | VIIRS ((NOAA–20)) |
|---|---|---|---|
| Orbit altitude (km) | 836 | 705 | 824 |
| Equator Crossing time | 14:00 LT | 13:30 LT | 14:20 LT |
| Swath (km) | 2900 | 2330 | 3060 |
| Pixel resolution at nadir (km) | 1 | 1 | 0.75/0.375 |
| Pixel resolution at the edge (km) | >6 | 4 | 1.5/0.75 |
| ID MIR Band (s) | 21 | 21/22 | M–13/I–4 |
| Spectral range (μm) | 3.973–4.128 | 3.929–3.989 3.940–4.001 | 3.973–4.128 3.550–3.930 |
| TMAX (SNR–NEΔT on orbit) | 380 K (0.25) | 500 K (0.183) 331 K (0.019) | 634 K (0.04) |
| ID TIR Band (s) | 24 | 31 | M–15/I–5 |
| Spectral range (μm) | 10.300–11.300 | 10.780–11.280 | 10.263–11.263 10.500–12.400 |
| TMAX (SNR–NEΔT on orbit) | 330 K (0.4) | 400 K (0.017) | 343 K (0.03) |

There are many citations in this manuscript that do not support the text before the citation. It seems that the authors didn't really read and try to understand these references, but just made the citation based on some related keywords. Below is a partial list of inappropriate citations I have found. Please carefully double check the citations throughout the manuscript.

Line 39: (Hussain and Reza, 2023) is not a good citation here; it studied the detrimental impact on global health by general environmental damages, not specifically from open biomass burning. There are many studies in literature about this topic which can be used for citation here.

Response:

Response: We changed the references.

"and have profound impacts on the global carbon cycle, climate change, and air quality, thus exerting a significant influence on the global environment and human health (Wu et al., 2022).".

Line 40–41: (Estrellan and Iino, 2010) reviewed toxic emissions from open burning. It did not provide evidence for "major fire types worldwide". So it is also not a good citation.

Response:

Response: We changed the references.

"Forest clearing, accidental fires, firewood burning, agricultural residue burning, peatland burning and straw burning are among the major fire types worldwide (van der Werf et al., 2017, p.1997–2016).".

Line 42: (Manisalidis et al., 2020) is a review of environmental and health impacts of air pollution. It did not talk about the specific impacts from "open burning activities".

Response:

Response: We changed the references.

"These open burning activities severely impact air quality and ecosystems and exacerbate climate change and air pollution issues (Anon, 2017).".

Line 44: (Ma et al., 2022) studied wildfires in Amazon during 2019 only. The paper does not support the claim "regions worldwide are experiencing a notable increase in fire incidents".

Response:

Response: We changed the references.

"…the Amazon rainforest fires (Pivello, 2011) …".

Line 45: (You and Xu, 2023) investigated how delayed wildfires in 2020 promote snowpack melting in the western US. Same as above, this paper does not support the 'increase in fire accidents'.

Response: We changed the references.

"…wildfires in the United States (Burke et al., 2021) …".

Line 56: (Lv et al., 2020) studies CO2 mixing ratio using satellite observations. They used the GFED dataset for CO2 emissions from biomass burning. This study does not support the previous sentence "Alternatively, a method based on the fire radiative power can effectively enhance the assessment of small fire events, thereby addressing this issue to a certain extent."

Response: We deleted it.

"... For example, similar approaches have been employed in Fire Emissions and Energy Research (FEER) and the Global Fire Assimilation System (GFAS)".

*Line 128: (Spawn and Gibbs, sssss2020). Remove the sssss here.*

Response: We Remove the "sssss" here.

Line 255: (Russell-Smith et al., 2021) focus on opportunities and challenges for savanna burning emissions abatement. It did not provide sufficient evidence to support the conclusion "In August, specific meteorological conditions, such as high temperatures and low humidity facilitated the increased combustibility of biomass, resulting in a peak in carbon emissions".

Response: We changed the references to support.

"In August, specific meteorological conditions, such as high temperatures and low humidity facilitated the increased combustibility of biomass, resulting in a peak in carbon emissions (Shea et al., 1996).".

Line 297: (Wiggins et al., 2020) presented estimates of fire emissions in the USA using data from the FIREX–AQ mission. It has little connection to the text preceding the citation.

Response: We changed the references to support.

"typically experience high levels of OBB emissions because of the prevalence of both natural and anthropogenic fire activities (Williams et al., 2019; Zheng et al., 2021).".

Line 308: (Thackeray et al., 2022) did study the precipitation change under global warming, but the main topic of this paper was precipitation extremes. It does not support the statement in this manuscript "an overall augmentation in annual precipitation played a key role".

Response: We changed it.

"… however, an overall augmentation in annual precipitation led to a reduction in the degree of drought (Thackeray et al., 2022; Zhang et al., 2023a).".

*There are also many cases where the presentation is poorly structured, vague, or inconsistent.*
*Line 23–26: The presentations of region names within the parentheses are inconsistent; the full name is shown for some regions, but not shown for other regions.*

Response: We added full name for other regions.

"72.71 (Boreal North America; BONA), 165.7 (Temperate North America, TENA), 34.1 (Central America; CEAM), 42.9 (Northern Hemisphere South America; NHSA), 520.5 (Southern Hemisphere South America; SHSA), 13 (Europe; EURO), 8.4 (Middle East; MIDE), 394.3 (Northern Hemisphere Africa; NHAF), 847 (Southern Hemisphere Africa; SHAF), 167.4 (Boreal Asia; BOAS), 27.9 (Central Asia; CEAS), 197.3 (Southeast Asia; SEAS), 13.2 (Equatorial Asia; EQAS), and 82.4 (Australia and New Zealand; AUST) Tg.".

Line 27–28: "...notable seasonal variability characterizes the OBB carbon emissions, with marked increases observed in July and August." Although I understand the meaning of this sentence, it is not well organized. For example, what is the object of comparison when you say 'marked increase'?

Response: We changed "...notable seasonal variability characterizes the OBB carbon emissions, with marked increases observed in July and August." to "... notable seasonal variability characterizes the OBB carbon emissions, with marked increases observed in August and September (annual average 441.32 Tg C) compared to other months (annual average 170.42 Tg C).".

Line 41–42: "These open burning activities severely impact air quality and ecosystems and exacerbate climate change and air pollution issues." In this sentence "severely impact air quality" and "exacerbate...air pollution" are basically referring to the same thing.

Response: We changed it.

"These open burning activities severely impact air quality and ecosystems ...".

Line 46–47: "These fires release substantial amounts of harmful particulate matter and organic pollutants, posing serious threats to air quality and potentially causing health problems". I don't understand why this sentence is here. Does it represent the same meaning as the first sentence in this paragraph?

Response: We deleted this ambiguous expression to make it clearer to understand.

Line 51: "The burned area method…". I believe most readers don't know what the 'burned area method' is. A short definition or introduction to this method needs to be presented here.

Response: We added a short introduction to this method.

"The burned area method demonstrated good accuracy in quantifying larger fire events, which is based on the burned area, the available biomass fuels burned in fields, the fuel–related combustion efficiency, and emission factors.".

Line 52–53: "Shi et al. (2020) estimated OBB emissions in tropical continents from 2001 to 2017 using widely used inventory data, such as the Global Fire Emissions Database (GFED) and the Fire INventory from NCAR (FINN)". I don't think Shi et al. (2020) estimated OBB emissions using GFED and FINN, since GFED and FINN are themselves global emissions datasets.

Response: We changed this ambiguous expression to make it clearer to understand.

"Shi et al. (2020) estimated OBB emissions in tropical continents from 2001 to 2017. As well as other open–access databases, such as the Global Fire Emissions Database (GFED) and the Fire INventory from NCAR (FINN)".

*Line 103: "... exhibited superior judgment accuracy". What is 'judgment accuracy' referring to?*
Response: We changed the description of FY–3D product. Please refer to Line 109.

*Line 117–118: "In contrast, satellite data cover the entire globe and provide surface parameters, thereby enabling biomass estimation." This is a potentially confusing sentence; Ground observations can also "provide surface parameters and enables biomass estimation".*
Response: We changed it.
"In contrast, satellite data cover the entire globe and provide surface parameters worldwide, thereby enabling biomass estimation."

*Line 126–128: "Global AGB for other years was generated based on the global aboveground and belowground biomass carbon density maps for the 2010 product". While I now understand the method by reading the SI, the sentence is not very clear in its current form. It's better to day that in 2010 the Spawn and Gibbs product was used and then say that in other years the AGB was estimated using a scalar based on TC and NDVI. BTW, AGB stands for "above ground biomass"; how did you derive the 'below ground' biomass?*
Response: We changed it.
"we combined the "global aboveground and belowground biomass carbon density maps for the 2010" product provided by Spawn and Gibbs(2020), annual TC, and NDVI data, and obtained by linear stretching the fuel loading for other years".
The below ground biomass is also provided by "global aboveground and belowground biomass carbon density maps for the 2010" product.

*Line 136: "the subsurface condition" should mean the below ground condition, but I suspect that you are referring to 'surface condition' here.*
Response: We changed "the subsurface condition" to "surface condition".

*Line 171–172: "the EF for the following seven land types were updated". It's not clear to me what original EF data were used and what data were used to replace (update) them.*
Response: We have changed the previous expression.
"…the EF for the following seven land types of other database were updated:".
We also added the reference of the EF data in Table 2.

*Line 178–181: Please combine/simplify these three sentences.*
Response: We combined these sentences.

"Taking carbon as an example, the annual carbon emissions from OBB were estimated for the period of 2020–2022 (Figure 1) and the total OBB carbon emissions reached 7760.63 Tg C.".

*Line 261: "Cumulatively, these territories represent…". What are "these territories". Based on the previous paragraph, they should probably include SHAF and NHAF. But these should be explicitly stated.*
Response: We changed "these territories" to "SHAF, SHSA, and NHAF ...".

*Line 317: "Over the past 3 years". The 'past 3 years' can change depending on the reference year. This kind of description should be more specific.*
Response: We changed "Over the past 3 years" to "During 2020 to 2022".

*Line 427: What are "substrate types"?*
Response: We changed incorrect description.
"To address the varying fire conditions, we performed a detailed subdivision based on different fire types.".

*There are other minor issues, including potential errors or typos*
*Line 60: "MEIRSI–2" should be "MERSI–2"*
Response: We corrected it.

*Figure 2: If these geographical regions are the same to that in GFED, you probably need to acknowledge/cite the GFED group/paper.*
Response: We've added citation information.

*Line 269: "intensify both the frequency and frequency of fires in the area". One 'frequency' should be removed or changed to other words.*
Response: We removed it.

*Line 280: "in the Tropical Eastern North America (TENA) region". As shown in Figure 2, TENA should be 'Temperate North America'.*
Response: We changed "in the Tropical Eastern North America (TENA) region" to "in the Temperate North America (TENA) region".

*Line 435–436: "Although the FY–3D GFR dataset is reliable for most OBB events, its resolution of 1 degree…" Shouldn't the resolution of FY–3D GFR dataset 1 km?*
Response: We changed "1 degree" to "1 km".

We thank the Reviewer for the constructive comments and suggestions. We shall revise the manuscript accordingly, and we address the comments as follows.

**Specific comments:**

*1. In the reply P16 line377–379, there is no clear explanation about how the difference in transmission time between the two satellites affects the detection of agricultural small fires. The authors only highlights that the algorithm of FY3D's adaptive threshold can effectively capture small fires and has high spatial resolution in the far infrared band, and high accuracy for large fire detection. The authors should further explain why FY3D is more capable of detecting small fires compared to MODIS from the algorithm principle.*

Response: We added more details to explain the capacity of FY-3D in detecting small fire. Please refer to the Line 135, 146, and 151.

Revision:

1. "First, FY-3D introduces an adaptive threshold using automatic identification algorithms for fire spot detection, which calculates the background temperature as the mean temperature of all the background pixels within each 3×3 window. If fewer than 20% of the pixels are identified as cloudless, the window size is expanded to 5×5, continuing up to 51×51 in order to accommodate more data (Chen et al., 2022). This approach eliminates the limitations posed by fixed thresholds in the MODIS and VIIRS algorithms, which set T4 to greater than 360 K (320 K at night) and fixed the moving window size at 21×21 (Giglio et al., 2016).".

2. "Finally, FY-3D employs a far-infrared band with a high resolution of 250 m, and channels 24 and 25, which has a higher resolution than MODIS (1 km) (Zheng et al., 2023). The far-infrared band has a higher sensitivity to large fires or high-brightness fire events and can distinguish differences against background brightness temperatures (Zheng and Chen, 2020).".

3. "Overall, the FY-3D GFR product has an accuracy of 94.01% globally, as calculated using fire detection after eliminating errors based on visual checks conducted using SMART (Visual Check) in 2019. It has accuracies of 94.61, 94.12, 90.63, 91.76, and 92.69% for Southern Central Africa, Eastern Central South America, Siberia, Australia, and the Indo-Chinese Peninsula, respectively (Chen et al., 2022). Specifically, owing to the removal of the underlying surface interference in China, FY-3D has accuracies of 79.43% and 88.50% for accuracy and accuracy without omission (Chen et al., 2022). These accuracies were determined by comparing the results of a large-scale field experiment conducted jointly by the State Grid Corporation of China and China Meteorological Administration with the GFR product, thereby calculating the accuracy, including and excluding mis-judgments. This comprehensive assessment took place throughout 2020 across five provinces in China—Guangdong, Guangxi, Yunnan, Guizhou, and Hainan—utilizing a combination of real-time satellite data and ground-truth validation to evaluate the suitability of these fire detection products. These accuracies are significantly higher than those achieved by MODIS, which are 74.23 and 79.69%, respectively (Chen et al., 2022).".

*2. According to the statement from the authors, FY-3D was able to enhance the detection of small fires compared to MODIS, but the analysis of the results did not indicate how much the estimated emissions increased due to the FY-3D detection product for the small fires.*

Response: We added analysis of the results to explain the analysis of the small fire by FY-3D detection product. Please refer to the Line 501. We added the comparisons of small fire detects in FY-3D and MODIS in supplements. Please refer to the Figure S2 and S3.

Revision: "While, the average annual estimated OBB emissions exceed those reported by GFED by 617.14 Tg C/year. These discrepancies are probably related to small-scale fire events. For instance, the largest difference is observed in the SHAF region, exceeding by 248.01 Tg C/year, followed by SHSA (190.28 Tg C/year) and SEAS (103.92 Tg C/year). In the SHAF region, compared to MODIS active fire, FY-3D GFR detects more small fire points (Figure S2, Figure S3 (a), Figure S3 (b)), which are isolated within 5-kilometer resolution pixels. However, in this area, the majority of fire events are large-scale incidents, which means that although small fires are more numerous, they contribute minimally to the total emissions. Furthermore, fire events in SHSA (Figure S3 (c), Figure S3 (d)) and SEAS (Figure S3 (e), Figure S3 (f)) are primarily triggered by human activities, consisting of small-scale incidents that are significantly linked to the overall emissions. In contrast, areas frequently affected by large-scale fire events show relatively smaller discrepancies, such as TENA (99.05 Tg C/year), NHAF (51.94 Tg C/year), and other regions including NHSA, AUST, CEAM, MIDE, EURO, and EQAS (all under 15.00 Tg C/year).".

[Figure]

[Figure]

Figure S2. Small fire count in 2020-2022 between MODIS active fire and FY-3D GFR

Figure S3. Small fire counts in 2020-2022. (a) FY-3D GFR in Africa. (b) MODIS GFR in Africa. (c) FY-3D GFR in South America. (d) MODIS active fire in South America. (e) FY-3D GFR in Southeast Asia. (f) MODIS active fire in Southeast Asia. (g) FY-3D GFR in Global. (h) MODIS active fire in Global.

The quality of the revised manuscript remains below the standard expected for publication in the ESSD journal. Numerous inconsistencies, spelling and grammatical errors, scientific inaccuracies, unclear references, and unrelated citations persist throughout the manuscript. Specifically, the 'Introduction' and 'Materials and Methods' sections suffer from poor presentation, characterized by lengthy and convoluted sentences, as well as disjointed transitions.

During the previous round of revision, it appears that the authors addressed only the errors pointed out by the reviewers without proactively seeking to identify and rectify similar issues elsewhere in the manuscript.

While the 'Methods', 'Results', and 'Discussion' sections present scientific findings, they fail to significantly advance the field beyond the utilization of a different satellite active fire product. Many of the methodologies employed in this study rely on outdated knowledge and overlook recent developments.

Based on these observations, I do not recommend publication of the manuscript in its current form. A substantial overhaul of both the textual presentation and scientific discussion is imperative for future consideration of this manuscript for publication.

Below, I outline some of the issues observed in the 'Introduction' and 'Materials and Methods'. Please note that this list is not exhaustive, and it is essential for the authors to enhance the overall clarity and coherence of the manuscript.

We thank the Reviewer for the constructive comments and suggestions. We have revised the manuscript accordingly, and we address the comments as follows.

**Specific comments:**

*P1 L16-17: "Global high–resolution satellites can detect active fires, enabling a more accurate estimation of these emissions."*

*Regarding the term 'more accurate', it's important to specify what the comparison is being made against. Without a clear comparison, the statement lacks clarity.*

*The phrase 'these emissions' is ambiguous as it doesn't specify which emissions are being referred to. It would be helpful to clarify that these emissions pertain to those from Open Biomass Burning (OBB) to enhance the reader's understanding.*

Response: We clarified and rephrased the sentence for better understanding.

Revision: "A global emission inventory based on high-resolution satellite fire detection enables an accurate estimation of OBB emissions.". Please refer to Line 17.

*P1 L19: "satellite and observational biomass data"*

*While "satellite biomass data" is a type of observational data, the phrase "satellite and observational" is redundant. Consider using "satellite-derived biomass data" or "observational biomass data" to avoid redundancy.*

Response: We changed "satellite and observational" into "satellite-derived biomass data". Please refer to Line 21.

*P1 L19: "vegetation index–derived spatiotemporal variable combustion efficiencies"*

*Consider revising "spatiotemporal" to "spatiotemporally" for grammatical correctness.*

Response: We changed "spatiotemporal" into "spatiotemporally". Please refer to Line 21.

*P1 L20: "The average annual OBB emissions for 2020–2022 were…"*

*Specify that these values are estimates derived from the study rather than presented as absolute facts. For example, consider revising to "The average annual estimated OBB emissions for 2020–2022 were..."*

Response: We changed "The average annual OBB emissions for 2020–2022 were…" into "The average annual estimated OBB emissions for 2020–2022 were ...". Please refer to Line 22.

*P1 L23-27:*

*The average annual OBB emissions for different regions have different precisions, ranging from 0 (e.g., 13 for EURO), 1 (e.g., 165.7 for TENA), to 2 (e.g., 72.71 for BONA). These inconsistencies in precision across different regions negatively impact the scientific credibility of the study.*

Response: We standardized precision to eliminate inconsistencies. Please refer to Line 25.

Revision: "the average annual estimated OBB for 2020–2022 were 72.71 (Boreal North America; BONA), 165.73 (Temperate North America, TENA), 34.11 (Central America; CEAM), 42.93 (Northern Hemisphere South America; NHSA), 520.55 (Southern Hemisphere South America; SHSA), 13.02 (Europe; EURO), 8.37 (Middle East; MIDE), 394.25 (Northern Hemisphere Africa; NHAF), 847.03 (Southern Hemisphere Africa; SHAF), 167.35 (Boreal Asia; BOAS), 27.93 (Central Asia; CEAS), 197.29 (Southeast Asia; SEAS), 13.20 (Equatorial Asia; EQAS), and 82.38 (Australia and New Zealand; AUST) Tg C/year.".

*P1 L28: "the lion's share of total emissions"*

*While "lion's share" is a colorful and idiomatic expression, it may not be the most appropriate choice for a scientific journal. Scientific writing typically aims for clarity and precision, avoiding figurative language or colloquialisms that might distract from the data or findings being presented.*

Response: We combined the question "P1 L28", "P1 L29-30", "P1 L31", and "P1 L32-34" as follows.

We newly added and modified the following contents. Please refer to Line 30.

Revision: "Overall, savanna grassland burning contributed the largest proportion of the annual total carbon emissions (1,209.12 Tg C/year; 46.74%), followed by woody savanna/shrubs (33.04%) and tropical forests (12.11%). SHAF was found to produce the most carbon emissions globally (847.04 Tg C/year), followed by SHSA (525.56 Tg C/year), NHAF (394.26 Tg C/year), and SEAS (197.30 Tg C/year). More specifically, savanna grassland burning was predominant in SHAF (55.00%, 465.86 Tg C/year), SHSA (43.39%, 225.86 Tg C/year), and NHAF (76.14%, 300.21 Tg C/year), while woody savanna/shrub fires were dominant in SEAS (51.48%, 101.57 Tg C/year). Furthermore, carbon emissions exhibited significant seasonal variability, peaking in September 2020, and August of 2021 and 2022, with an average of 441.32 Tg C/month, which was higher than the monthly average of 215.57 Tg C/month.".

*P1 L29-30: "with marked increases observed in August and September (annual average 441.32 Tg C) compared to other months"*

*According to Figure 4, OBB emissions in September did not have 'marked increases' from that in July (in fact, 2021 July emissions were higher than that in 2021 September). Avoid subjective terms like "marked increases" unless they are supported by the data.*

Response: We clarified and rephrased the sentence for better understanding. We combined the question "P1 L28", "P1 L29-30", "P1 L31", and "P1 L32-34" as follows. Please refer to the Response to Specific Comment P1 L28 above and Line 30.

*P1 L31: "This surge in carbon emissions is…"*

*Choose a more neutral term than "surge" to describe the increase in carbon emissions. Perhaps "significant rise" or "notable increase" would be more appropriate.*

Response: We changed "surge" into a more neutral term to describe the increase in carbon emissions. We combined the question "P1 L28", "P1 L29-30", "P1 L31", and "P1 L32-34" as follows. Please refer to the Response to Specific Comment P1 L28 above and Line 30.

*P1 L32-34: "Fires in savanna grasslands were predominant in the NHAF, contributing to 77% of emissions during January–April, whereas in the SEAS, woody savanna/shrubs (52%) and tropical forests (23%) were the primary sources."*
*This sentence only presents detailed results from this study without any scientific explanation and discussion. I think it should belong to the results section, not here in the abstract.*

Response: We clarified and rephrased the sentence for better understanding. We combined the question "P1 L28", "P1 L29-30", "P1 L31", and "P1 L32-34" as follows. Please refer to the Response to Specific Comment P1 L28 above and Line 30.

*P2 L40-41: "Forest clearing, accidental fires, firewood burning, agricultural residue burning, peatland burning and straw burning are among the major fire types worldwide"*
*The fire types listed here are not well organized. 'Straw burning' is a type of 'agricultural residue burning'. It's redundant to list both items. 'Accidental fires' represent one of the fire sources (not types). Also, you forgot to list the burning type that consumes most of the global burned area: savanna/grassland/shrub fires. Please organize the fire types more efficiently and eliminate redundancy.*

Response: We reorganized the fire types, removed the redundancy and ensured all major types are covered. Please refer to Line 51.

Revision: "The burning of forests, shrublands, grasslands, crop residues, and peatland constitutes the major types of fires worldwide"

*P2 L44: "However, some regions worldwide are…"*

*Since there is no clear contrast in the preceding sentences, avoid using "however" here.*

Response: We changed "However" into "In addition,". Please refer to Line 55.

*P2 L44-46: "...experiencing a notable increase in fire incidents (Richardson et al., 2022), such as, the Amazon rainforest fires (Pivello, 2011), Australian bushfires (Jegasothy et al., 2023), and wildfires in the United States (You and Xu, 2023), which are large–scale fire incidents that occur multiple times annually."*

*Remove the comma after "such as". Also please clarify what does "...which are large–scale fire incidents that occur multiple times annually" refer to.*

Response: We removed the comma after "such as" and clarified regarding the referenced incidents. Please refer to Line 56.

Revision: "… such as the Amazon rainforest (Pivello, 2011), Australian bush (Jegasothy et al., 2023), and the United States (You and Xu, 2023), where large-scale fire incidents occur periodically and frequently (Kolden et al., 2024).".

*P2 L50-52: "The burned area method demonstrated good accuracy in quantifying larger fire events, which is based on the burned area, the available biomass fuels burned in fields, the fuel-related combustion efficiency, and emission factors."*

*Consider changing "the burned area method" to 'the burned area based fire emissions method". Also, please clarify what the relative pronoun "which" refers to.*

Response: We changed "the burned area method" to "the burned-area-based fire emission estimation method", and clarified regarding the referenced incidents. Please refer to Line 62.

Revision: "The burned-area-based fire emission estimation method, which is based on the burned area, available biomass fuels burned in the fields, fuel-related combustion efficiency, and emission factors, has demonstrated good accuracy in quantifying larger fire events.".

*P2 L52-54: "As well as other open-access databases, such as the Global Fire Emissions Database (GFED) and the Fire INventory from NCAR (FINN) (Jiang et al., 2012; van Wees et al., 2022)"*

*This is not even a complete sentence!*

*Jiang et al 2012 is not a good reference for FINN and van Wees et al 2022 is not a good reference for GFED.*

Response: We completed the sentence and changed the reference for FINN and GFED. Please refer to Line 66.

Revision: "This method has been widely used in databases such as the Global Fire Emissions Database (GFED) (van der Werf et al., 2017) and the Fire INventory from NCAR (FINN) (Wiedinmyer et al., 2023).".

*P2 L57: "the Global Fire Assimilation System (GFAS) (Di Giuseppe et al., 2017)."*

*Replace Di Giuseppe et al., 2017 with a better reference for GFAS.*

Response: We changed a better reference for GFAS. Please refer to Line 76.

Revision: "the Global Fire Assimilation System (GFAS) (Kaiser et al., 2012)."

*P2 L57-58: "this approach has a drawback in that it tends to overestimate emissions during localized fire events."*

*Specify what is meant by 'localized fire events' and why the FRP approach tends to overestimate emissions in such cases. Provide more clarity and context.*

Response: We added a definition of "localized fire events" and provided more information and background to explain this. Please refer to Line 69.

Revision: "A method based on fire radiative power (FRP) can enhance the detection and quantification of small fire events by measuring the energy released during combustion (Filizzola et al., 2023). However, these approaches can overestimate emissions from localized fire events, which are intense, small-scale fires that may not reflect wider fire activity (Nguyen et al., 2023). For example, Fire Emissions and Energy Research (FEER), based on FRP, reported that the global total particulate matter emissions were approximately 55% higher than those estimated by the GFED (Ichoku and Ellison, 2014). Similarly, the Global Fire Assimilation System (GFAS) using FRP estimated global and regional combustion values exceeding those of the GFED by approximately 126 Tg C/year during 2003-2008 (Kaiser et al., 2012).".

*P2 L60: "the Fengyun–3D (FY–3D) satellite offers spatial resolutions of 250 and 1000m…, which, when compared to MODIS, significantly enhances its capacity…"*
*For clarity, it might be better to say a satellite 'has spatial resolutions' instead of 'offers spatial resolutions'. The structure of this sentence is also not optimal and needs revision.*

Response: We changed the "offers spatial resolutions" into "has spatial resolution", and changed the structure of this sentence. Please refer to Line 79.

Revision: "Similar to the MERSI-2 instrument, the Fengyun-3D (FY-3D) satellite has spatial resolutions of 250 and 1000 m at the nadir (Yin et al., 2020), which is more advantageous in detecting and monitoring various active fire events compared with MODIS (Zheng et al., 2023).".

*P2 L64: "This resulted in an impressive overall accuracy rate"*
*In scientific writing, it's generally advisable to maintain a tone of objectivity and avoid overly subjective or emphatic language such as 'impressive' in this sentence.*

Response: We deleted "impressive" in the sentence.

*P3 L72-73: "Therefore, employing the FY–3D GFR product and allocation approaches for small fires is expected to yield reliable estimates of OBB emissions."*
*Strengthen the causal relationship between the preceding sentence and this one (since you are using 'therefore' here)*

Response: We accepted the suggestion for strengthening the causal relationship. Please refer to Line 90.

Revision: "Although the Landsat Fire and Thermal Anomaly (LFTA) product has a finer spatial resolution, its lower temporal resolution limits its global coverage to only 16 days; thus, large numbers of fires with short durations are missed. Given these limitations in the monitoring frequency with the LFTA product, employing the FY-3D GFR product and allocation approaches for short fires are expected to yield reliable estimates of OBB emissions.".

*P3 L73-74: "Many studies treat F as a constant based on regional land cover types, neglecting the actual spatial and temporal variability (Wiedinmyer et al., 2011)"*

*Many recent studies have already explicitly addressed the spatial and temporal resolution of fuel loads.*

*Methods that utilize fuel load parameterization based on regional land cover types do not necessarily neglect spatial and temporal variability; they simplify the spatial and temporal representation for computational efficiency.*

Response: We clarified the representation of spatial and temporal variability. Here we want to stress the variabilities of F within each land type, which means that the F of each land type change from pixel to pixel. We revised the expression as follows. Please refer to Line 96.

Revision: "Many studies have adopted a static approach to F (Chang and Song, 2010; Zhou et al., 2017; Puliafito et al., 2020; Shi et al., 2020), assigning constant values based on regional land-cover types. This methodology overlooks the inherent spatial and temporal variability of F within each land type, which changes continuously and dynamically (Wiedinmyer et al., 2011).".

*P3 L74-75: "the combustion factor (CF), which represents the proportion of small biomass burned in a fire event, is typically assumed to be constant without considering the fuel status and humidity conditions (Pfeiffer et al., 2013)"*

*This definition of CF seems different to what I know. Please review the definition of CF to ensure accuracy and clarity. Pfeiffer et al., 2013, focuses on a fire model for global biomass burning, which may not be the most appropriate reference for the CF approach used in observation-based global emissions calculations.*

Response: We changed the definition of combustion factor and replaced the reference. Please refer to Line 100.

Revision: "The combustion factor (CF), which denotes the ratio of consumed fuel to total available fuels, is typically a linear variable within a specific range when considering the fuel status and humidity conditions (van der Werf et al., 2006; Wiedinmyer et al., 2011).".

*P4 L98-99: "where Ei (g/m2) represents type i emissions at location x, which is equal to the product of the burning area B (m2) at time t and location x, biomass F (g C/m2) at location x, CF (expressed as a fraction), and the emission factor EF (g/kg) for type i pollutants."*

*The units of the variables in Equation 1 do not align: multiplying the units of B, F, CF, and EF does not result in the unit of E.*

Response: We changed the units in Equation 1. Please refer to Line 124.

Revision: "where $E_i$ (g) represents pollutant type $i$ emissions at location $x$, which is equal to the product of burning area $B$ ($m^2$) at time $t$ and location $x$, biomass $F$ ($kg/m^2$) at location $x$, $CF$ (expressed as a *fraction*), and the emission factor $EF$ (g/kg) for pollutant type $i$ .".

*P4 L100: "FY–3D global fire spot monitoring data based burned area (B)"*

*Please acknowledge that the active fire spot data is used as a proxy for burned areas in this section. Clarify the difference between these two datasets and the uncertainties resulting from this approximation to provide a more comprehensive understanding.*

Response: We clarified the difference between burned area we used and active spot fire. Please refer to Line 165.

Revision: "The location, timing and burned area of the fire events used in the GEIOBB were determined globally using the FY-3D GFR product (Chen et al., 2022). Processed fire event detection data Fengyun Satellite Remote Sensing Data Service Network of National Satellite Meteorological Centre (http://satellite.nsmc.org.cn/PortalSite/Default.aspx), which estimated the actual area of fire spots based on radiation in different infrared channels. When the mid-infrared channel was not saturated, it was used to estimate the sub-pixel fire spot area and temperature. Otherwise, a far-infrared channel was employed for the estimation (Zheng and Chen, 2020). These data offer daily fire detection at a 1-km resolution, including the location, time, burned area, and confidence level (Liu and Shi, 2023). Furthermore, multiple counts of the same fire may have been recorded on a single day, leading to data duplication. To address this issue, we performed a global identification and removed multiple daily detections of the same fire pixels and data with confidence levels below 20%. Specifically, we removed single daily fire detections within a 1-km radius of another fire detection. Thus, only one fire per 1 km2 of a hotspot could be counted per day and was reset on the next day (Wiedinmyer et al., 2023).".

*P4 L101: " Chines polar–orbiting meteorological satellites."*
*Correct the typo "Chines" to "Chinese"*
Response: We changed "Chines" to "Chinese". Please refer to Line 129.

*P4 L102-103: "It (FY-3D) is at an altitude of 836 km and was launched on November 15, 2017 and published on may, 2020."*
*What do you mean by saying a satellite (FY-3D) was "published"?*
*'may' should be changed to 'May'.*
Response: We changed "may" into "May" and changed "published" into "became accessible". Please refer to Line 131.
Revision: "It was launched on November 15, 2017, at an altitude of 836 km, and the data became accessible in May 2020 (Li et al., 2017)."

*P4 L106-107: "FY–3D introduces the adaptive threshold and eliminates the limitations by fixed thresholds of MODIS and VIIRS algorithms"*
*The 'fixed thresholds' are not clearly described. There are multiple thresholds used in the spectral and contextual test in the MODIS/VIIRS algorithm. Specify which specific fixed thresholds are being referred to. Provide more detail to enhance understanding.*
Response: We added more detail about the "fixed thresholds" and the differences between algorithms. Please refer to Line 135.
Revision: "First, FY-3D introduces an adaptive threshold using automatic identification algorithms for fire spot detection, which calculates the background temperature as the mean temperature of all the background pixels within each 3×3 window. If fewer than 20% of the pixels are identified as cloudless, the window size is expanded to 5×5, continuing up to 51×51 in order to accommodate more data (Chen et al., 2022). This approach eliminates the limitations posed by fixed

thresholds in the MODIS and VIIRS algorithms, which set T4 to greater than 360 K (320 K at night) and fixed the moving window size at 21×21 (Giglio et al., 2016).".

*P4 L111-112: "the far–infrared channel employed in FY–3D has a high resolution of 250 m, higher than MODIS with 1 km, resulting in higher accuracy in big fire detection"*
*Explain in more detail how the higher resolution of the far-infrared channel in FY–3D may lead to increased accuracy in detecting big fires.*
Response: We added more detail about far-infrared channel in in detecting big fires. Please refer to Line 146.
Revision: "Finally, FY-3D employs a far-infrared band with a high resolution of 250 m, and channels 24 and 25, which has a higher resolution than MODIS (1 km) (Zheng et al., 2023). The far-infrared band has a higher sensitivity to large fires or high-brightness fire events and can distinguish differences against background brightness temperatures (Zheng and Chen, 2020).".

*P4 L113: "the FY–3D GFR product achieves an accuracy of 94.0% globally"*
*Define the metric used to determine accuracy and provide information on the reference datasets. Specify the variables compared and the methodology used to derive the accuracy percentage.*
Response: We added the method used to determine accuracy. Please refer to Line 151.
Revision: "Overall, the FY-3D GFR product has an accuracy of 94.01% globally, as calculated using fire detection after eliminating errors based on visual checks conducted using SMART (Visual Check) in 2019. It has accuracies of 94.61, 94.12, 90.63, 91.76, and 92.69% for Southern Central Africa, Eastern Central South America, Siberia, Australia, and the Indo-Chinese Peninsula, respectively (Chen et al., 2022).".

*P4 L116: "for accuracy and accuracy without omission"*
*Define "accuracy without omission" to ensure clarity and understanding.*
Response: We added more details about "accuracy without omission" to ensure clarity and understanding. Please refer to Line 155.
Revision: "Specifically, owing to the removal of the underlying surface interference in China, FY-3D has accuracies of 79.43% and 88.50% for accuracy and accuracy without omission (Chen et al., 2022). These accuracies were determined by comparing the results of a large-scale field experiment conducted jointly by the State Grid Corporation of China and China Meteorological Administration with the GFR product, thereby calculating the accuracy, including and excluding mis-judgments. This comprehensive assessment took place throughout 2020 across five provinces in China—Guangdong, Guangxi, Yunnan, Guizhou, and Hainan—utilizing a combination of real-time satellite data and ground-truth validation to evaluate the suitability of these fire detection products. These accuracies are significantly higher than those achieved by MODIS, which are 74.23 and 79.69%, respectively (Chen et al., 2022).".

*P4 L126: Table 1*
*'((NOAA-20))' should be '(NOAA-20)';*

*Explain the values within the parentheses (for TMAX)?*

*Give a definition for TMAX, SNR, and NEdT*

Response: We changed "((NOAA-20))" into "(NOAA-20)" and added the definition for TMAX, SNR, and NEdT under Table 1. Please refer to Supplements Information (SI) Table S1.

Revision: "TMAX means maximum temperature, SNR means signal-to-noise ratio, and NE$\Delta$T means noise equivalent differential temperature.".

*P5 L129-130: "Previous studies on emission inventories based on wildfire areas were mostly used to assess F by defining different fire types in different areas"*

*Clarify the meaning of the sentence (or rephrase it) for better understanding.*

Response: We clarified and rephrased the sentence for better understanding. Please refer to Line 181.

Revision: "Previous studies based on burned areas have distinguished F by categorizing it according to regions of different fire types (Wiedinmyer et al., 2011).".

*P5 L136-139: "This fusion method combines the high accuracy of ground observation data with wide coverage of satellite data to produce reliable and precise global biomass products. Using this method, it is possible to overcome the limitations of a single data source, thereby enhancing the accuracy and reliability of biomass estimation."*

*Tone down the language describing the fusion method (such as 'effective', 'reliable', 'precise') to avoid overconfidence. While the approach of combining different data streams can help mitigate limitations of using a single data source, it may introduce additional uncertainties.*

Response: We toned down the language describing the fusion method. Please refer to Line 189.

Revision: "This fusion method combines the high accuracy of ground observation data with the wide coverage of satellite data to generate global biomass products. Using this method, it is possible to overcome the limitations of using a single data source, thereby enhancing the accuracy of biomass estimations.".

Actually, the produced biomass map of by using this fusion method had a root mean square error (RMSE) 15–21% lower than those reported in Saatchi et al. (2011) and Baccini et al. (2012). We newly added the following content to clarify it. Please refer to Line 203.

"A combination of 2118 other ground measurements and Lidar data to validate observations, and showed that the fused map had a root mean-square error (RMSE) that was 15–21% lower than those reported by Saatchi et al. (2011) and Baccini et al. (2012)."

*P5 L139: "This study used multi–source data, including NDVI, tree cover (TC), and satellite and observational AGB, to ..."*

*NDVI is never defined in the manuscript.*

*Again, satellite AGB is a type of observational AGB. So it's redundant to say 'satellite and observational AGB'.*

Response: We have defined NDVI in previous manuscript. Please refer to the existing material in Line 122.

We changed "satellite and observational AGB" into "AGB". Please refer to Line 193.

*P5 L143: "GEE platform"*

*GEE is not defined.*

Response: We changed "GEE platform" to "google earth engine platform". Please refer to Line 195.

Revision: "…Google Earth Engine platform".

*P5 L144: "so we combined the global aboveground and belowground biomass carbon density maps"*

*Provide a more detailed description of the global aboveground and belowground biomass carbon density maps to enhance understanding, as they are core datasets for fuel loading used in the study.*

Response: We clarified and rephrased the sentence for better understanding. "Global Aboveground and Belowground Biomass Carbon Density Maps for the Year 2010" is the product name for the AGB data. Please refer to Line 198.

Revision: "AGB data were obtained from the Global Aboveground and Belowground Biomass Carbon Density Maps for the Year 2010 product (https://daac.ornl.gov/cgi-bin/dsviewer.pl?ds_id=1763) provided by Spawn and Gibbs (2020). This dataset uses thousands of satellite data points and ground measurements to produce a biomass map with a 1-km resolution (Spawn and Gibbs, 2020). A combination of 2118 other ground measurements and Lidar data to validate observations, and showed that the fused map had a root mean-square error (RMSE) that was 15–21% lower than those reported by Saatchi et al. (2011) and Baccini et al. (2012). We used the AGB for 2010, annual TC, and NDVI data, and linearly stretched the fuel loading for other years.".

*P6 L147: Equation 2*

*Verify the accuracy of Equation 2, as it appears to lead to a discrepancy. For year 2010, this equation lead to 2\*AGB, while AGB is defined as AGB data in 2010. I guess there should be an additional coefficient of ½ in the equation.*

Response: We corrected the equation 2. Please refer to Line 213.

Revision:

$$F(x,t) = \left( \frac{NDVI_{now} + TC_{now}}{NDVI_{2010} + TC_{2010}} \right) * AGB \qquad (2)$$

*P6 L149: "TC2010 is the tree cover in 2020"*

*Confirm the correct year for TC2010, as it appears to be a typo.*

Response: We changed "$TC_{2010}$ is the tree cover in 2020" into "$NDVI_{2010}$ is the mean value of NDVI in 2010". Please refer to Line 215.

*P6 L154: "Typically, CF is set as a constant"*

*Revise to reflect the evolving understanding in recent studies, which often employ parameterizations to account for spatial and temporal variations, rather than assuming a constant CF.*

Response: We changed the wrong description. Please refer to Line 220.

Revision: "Typically, the CF is set as a linear variable within a specific range, which may lead to biases in emission

estimations and generate significant uncertainties. Although some studies used TC to quantify CF and explain its spatial and temporal variations (Wiedinmyer et al., 2006; Qiu et al., 2016; Bray et al., 2018; Wu et al., 2018), previous research has mainly focused on areas with herbaceous vegetation cover, where the TC ranges from 40% to 60%.".

*P6 L159: "A major influence on fire discharge in the framework is the surface condition…"*
*Clarify the terms 'fire discharge' and 'framework'*
Response: We clarified and rephrased the sentence for better understanding.
Revision: "The fire type at the location of the fire event has a major influence on OBB."

*P6 L159-160: "Different landtypes exhibit different biological qualities and correlations."*
*Clarify the term "biological quantities" for better understanding.*
Response: We removed this incorrect sentence for better understanding.

*P6 L162-163: "reclassified the original 17 classifications, and reclassified the results to reorganize the subsurface types into seven categories"*
*Revise for clarity and organization. Consider splitting the sentence for improved readability.*
Response: We clarified and rephrased the sentence for better understanding.
Revision: "We reclassified the original 17 classifications into 7 categories to better differentiate fire types; grasslands and savannas (V1), woody savannas or shrubs (V2), tropical forests (V3), temperate forests (V4), boreal forests (V5), temperate evergreen forests (V6), and crops (V7).".

*P6 L167-168: "we amalgamated the reclassification outcomes of V3, V4, V5, and V6 into a forest type category, designated V1 and V2 as woodlands, and assigned V7 to crops"*
*Clarify the two-step process of aggregation for better understanding.*
*Explain the relationship between grassland and woodland and how the original classifications were aggregated to represent grassland.*
*Provide rationale for not directly aggregating the MCD12Q1 classifications to the final categories of forest, woodland, and cropland.*
Response: In the calculation of CFs, we classified fire types into 4 categories, which can reflect the extent of burning of each land type. Currently, this is the most concrete and detailed classifications. However, EFs of each species of different fire types vary, we classified them into 7 classes to accurately estimate the OBB emissions. Throughout the whole text, our analysis were based on 7 classes of different fire types.

*P6 L173: "We incorporated the VCI to ascertain fuel moisture conditions"*
*Define 'ascertain' for clarity*
Response: We changed the "ascertain" into "assess".

*P6 L176: Equation 4*

*Capitalize the variable 'vci'*

Response: We changed "*vci*" into "*VCI*" in Equation 4. Please refer to Line 250.

*P7 L177: Equation 5*

*Justify the changes made to Equation 5 compared to Equation 10 in Ito and Penner, 2004, to provide clarity on the modifications and their impact on the model.*

Response: We provided the justification for changes in Equation 5 compared to previous equations. Please refer to Line 241.

Revision: "For grassland fires, a change in the NDVI is usually associated with the occurrence of fires, especially in dry seasons or in areas prone to wildfires. Generally, a decrease in NDVI may indicate deteriorating vegetation health, which increases the risk of fires because dry or withered vegetation is more prone to burning. We introduced the vegetation condition index (VCI) to determine the fuel moisture conditions, which were used to measure the vegetation drought conditions by calculating contemporaneous changes in NDVI as a metric for assessing the contemporaneous conditions of vegetation. We supplemented our research based on Ito and Penner (2004) by replacing the percentage of green grass from the total grass with the VCI, which was computed using the NDVI with a time interval of 16 d at a spatial resolution of 1 km for the period of 2020–2022. In addition, we introduced a compensatory term to mitigate the impact of tree cover on grassland fires."

*P7 L178-179: "NDVImax the maximum value of NDVI in the same period in the previous 3 years"*

*Explain the rationale for using the previous 3 years as the reference period for NDVImax to provide context for the choice of this time frame.*

Response: We clarified and rephrased the sentence for better understanding. Please refer to Line 252.

Revision: "where $NDVI_{now}$ is the mean value of the month before a single fire event, $NDVI_{max}$ is the maximum value of $NDVI$ for the same period in the previous three years of the fire event, and $NDVI_{min}$ is the minimum value of $NDVI$ for the same period in the previous three years of the fire event."

*P7 L180-181: "conducted an analysis based on the partitioning provided"*

*Clarify what is meant by "partitioning provided" to ensure understanding.*

Response: We added the clarification of "partitioning provided". Please refer to Line 255.

Revision: "For forest fires, we used moisture category factors (MCF) to measure forest moisture and conducted an analysis based on the partitioning of MCF values (very dry: 0.33, dry: 0.5, moderate: 1, moist: 2, wet: 2, and very wet: 5) provided by Anderson et al. (2004)."

*P7 L182: " function fitting was executed"*

*Whether the 'function fitting' is referring to the 'power function fitting'?*

*Provide details on the data used for the fitting. I understand VCI was calculated from MODIS NDVI, but where is the*

*MCF data coming from?*

Response: We changed "function fitting" to "power function fitting". We added the details on the data used for the fitting. Please refer to Line 257.

Revision: "We used the VCI as a criterion for assessing wetness and dryness and discovered that it approximately conformed to the power function distribution characteristics of VCI."

*P7 L182-183: "For grasslands, the VCI could be directly calculated and utilized."*

*Clarify the relevance of including information about grasslands in a paragraph focused on deriving CF for forest fires.*

Response: Since this sentence is duplicated and appeared in "For grasslands…" part. Please refer to Line 241. Therefore, we removed the incorrect description.

*P7 L189: "Here, EF in Tabel 2 was assigned according to the LCT"*

*Correct the typo "Tabel" to "Table"*

*Although the data sources in Table 2 have been added in the revised manuscript, a description of the method (e.g., the original data approach, the aggregation algorithm) is still needed here.*

Response: We changed the "Tabel" to the "Table", and added the description of the method and data. We combined the question "P7 L189", "P7 L191", and "P7 L194-195" and as follows. Please refer to the Line 266.

Revision: "The measurements of EFs in different regions for grasslands and savannas, woody savannas or shrubs, tropical forests, temperate forests, temperate evergreen forests, and crops were reviewed and tabulated by Akagi et al. (2011), whereas those for boreal forest fires were obtained from the averages reported by Akagi et al. (2011) and Urbanski (2014). The EFs for maize, sugar, and rice crop fires were taken from the averages reported by Akagi et al. (2011), Fang et al. (2017), Liu et al. (2016), Santiago-De La Rosa et al. (2018), and Stockwell et al. (2015). The BC EFs of BC for crop fires were sourced from Kanabkaew and Kim Oanh (2011) and those for wheat fires were obtained from Cao et al. (2008). In addition, the emission factors of $NO_2$, $PM_{2.5}$, and $PM_{10}$ for the crop fire were derived from Li et al. (2007), and the EF from the crop was the average of maize, sugar, rice, and wheat. The EFs values are presented in Table 1.".

*P7 L191: "However, other EF measurements were also used when locally measured EF data were not available."*

*Specify which dataset contains the locally measured EF data and clarify the meaning of "other EF measurements" for better understanding.*

Response: We added the description of the method and all data used of EF. We combined the question "P7 L189", "P7 L191", and "P7 L194-195" and as follows. Please refer to the Response to Specific Comment P7 L189 above and Line 266.

*P7 L194-195: "Finally, the EF for the following seven land types of other database were updated"*

*Clarify the reference data used for updating EF values and provide additional context on the update process for clarity.*

Response: We clarified the reference data used and added context on the update process for clarity. We combined the

question "P7 L189", "P7 L191", and "P7 L194-195" and as follows. Please refer to the Response to Specific Comment P7 L189 above and Line 266.

I appreciate the efforts the authors have made in this revised manuscript. Overall, the readability of the manuscript has been greatly improved, and the details about the data and methods have been added and described more clearly. I only have some minor suggestions:

We thank the Reviewer for the constructive comments and suggestions. We have revised the manuscript accordingly, and we address the comments as follows.

**Specific comments:**

*Page 1, Line 17: "high-resolution satellite fire detection"*
*While 1 km resolution can be considered "high resolution" for a global emissions dataset, it is inaccurate to call it "high-resolution" in terms of remote sensing. "High-resolution" remote sensors generally have a spatial resolution finer than 10 meters. I suggest removing "high-resolution" here.*
Response: We revised the manuscript according to the reviewer's comment. We removed "high-resolution".

*Page 1, Line 33-34: "carbon emissions exhibited significant seasonal variability, peaking in …, with an average of …, which was higher than the monthly average of…"*
*It is evident that emissions during the peak-burning months are higher than the all-month average. If you just want to make a comparison, consider changing "which was higher than…" to "substantially higher than…".*
Response: We revised the manuscript according to the reviewer's comment. We changed the sentence to "substantially higher than…". Please refer to the Line 34.

*Page 2, Line 63: "Similar to the MERSI-2 instrument, the Fengyun-3D (FY-3D) satellite has spatial resolutions of 250 and 1000 m at the nadir"*
*"MERSI-2 instrument" is not mentioned earlier in the text, so readers may be unfamiliar with it. Also, it would be better to indicate which channels in FY-3D have 250 m resolution and which have 1000 m resolution.*
Response: We revised the manuscript according to the reviewer's comment. We newly added the channels in FY-3D with 250 m resolution and 1000 m resolution as follows. Please refer to the Line 65.
"Similar to the MERSI-2 instrument, the Fengyun-3D (FY-3D) satellite has spatial resolutions of 250 m (0.47–0.86 μm and 10.80–12.02 μm) and 1000 m (1.38–8.55 μm) at the nadir (Yin et al., 2020)".

*Page 3, Line 73-75: "Given these limitations in the monitoring frequency with the LFTA product, employing the FY-3D GFR product and allocation approaches for short fires are expected to yield reliable estimates of OBB emissions."*
*A shortcoming in other products does not guarantee good performance of the FY-3D based OBB emissions estimates. The entire logic chain should be provided by saying something like "The shorter revisit time of FY-3D allows for…".*

*Also, the meaning of "short fires" is unclear to me.*

Response: We revised the manuscript according to the reviewer's comment. We revised this expression into following sentence and deleted the misrepresentation of "short fires". Please refer to Line 76.

"The shorter revisit time of FY-3D allows for monitoring more fires lasting for one day, which are expected to yield reliable estimates of OBB emissions.".

*Page 3, Line 81: "However, this approach leads to …"*
*You mentioned previous approaches to derive F and CF in the previous sentences. However, it is unclear what the phrase "this approach" refers to.*

Response: We revised the manuscript according to the reviewer's comment. Please refer to Line 86.

"This approach of calculating CF leads to …".

*Page 4, Line 102*
*There's a typo here: Ei (g) should be Ei (x)*

Response: Here, (g) is the unit of Ei and represents pollutant type i emissions at location x for Ei (x). We wrote Ei (x) in equation (1), please refer to the Line 106.

*Page 4, Line 115-116: "This approach eliminates the limitations posed by fixed thresholds in the MODIS and VIIRS algorithms, which set T4 to greater than 360 K (320 K at night) and fixed the moving window size at 21×21"*
*I believe the MODIS/VIIRS active fire algorithms used variable window sizes (from 3x3 to 21x21), not fixed sizes.*

Response: We revised the manuscript according to the reviewer's comment. We changed the mis misdescription, please refer to Line 120.

"This approach eliminates the limitations in the MODIS and VIIRS algorithms, which set T4 to greater than fixed 360 K (320 K at night) and the variable moving window size to a maximum of 21×21 (Giglio et al., 2016)".

*Page 4, Line 117: "re-identification index"*
*It's hard to understand what "re-identification index" is and how its use may have led to improvement. Please provide a brief explanation.*

Response: We revised the manuscript according to the reviewer's comment. We newly provided a brief explanation. Please refer to Line 123.

"Based on the initially identified fire spots, FY-3D employed the re-identification index to further remove false fire spots at cloud edges, water body edges and other high-reflection underlying surfaces. (Chen et al., 2022)".

*Page 4, Line 120-121: "far-infrared band with a high resolution of 250 m, and channels 24 and 25, which has a higher resolution than MODIS (1 km)"*
*Are channels 24 and 25 part of the 'far-infrared bands'? Why are you mentioning them here?*

Response: We deleted this sentence, as it does not directly relate to the subsequent content. Please refer to Line 127.

"FY-3D employs a far-infrared band with a high resolution of 250 m, which has a higher resolution than MODIS (1 km)".

*Page 4, Line 128: "...for accuracy and accuracy without omission"*
*I noticed the authors have tried to clarify the term "accuracy without omission" in the revision. But I still don't understand what exactly it means. Please provide a better explanation.*
Response: We revised the manuscript according to the reviewer's comment. We newly provided a better explanation. Please refer to Line 136.
"… for accuracy (omitted fire) and accuracy without omission (misidentified fire).".

*Page 4, Line 129-131*
*This sentence seems to have grammatical errors. Please revise for clarity and correctness.*
Response: We revised the manuscript according to the reviewer's comment. We newly provided a better explanation. Please refer to Line 136.
"These accuracies were determined by comparing the results of a large-scale field experiment conducted jointly by the State Grid Corporation of China and China Meteorological Administration with the GFR product, including omitted and misidentified fire (Chen et al., 2022).".

Page 5, Line 135: "The location, timing and burned area of the fire events used in the GEIOBB were determined globally using the FY-3D GFR"
A major assumption in the method of this study is to use active fire detection as a proxy for the burned area. This assumption is accompanied by large uncertainties and may lead to lower quality than the BA products derived using the change-detection method. Active fires can be obscured by thick clouds and smoke. Burning between satellite overpass gaps is often omitted. The diurnal cycle cannot be sufficiently represented using observations from polar orbiting satellites. I highly recommend discussing these uncertainties in the 'Discussion' section.
Response: We revised the manuscript according to the reviewer's comment. We newly added uncertainties. Please refer to Line 518.
"The detected active fires were also underestimated due to cloud cover/thick smoke and omitted between satellite overpass, with an omission error of approximately from 10%–30% (Giglio et al., 2006; Schroeder et al., 2008; Roberts et al., 2009). Furthermore, the diurnal cycle cannot be sufficiently represented using observations from polar orbiting satellites, as these satellites have limited temporal coverage and may not capture the full range of fire activity throughout the day (Huang et al., 2024; Zheng et al., 2021b).".

Page 18, Line 430-433
The overpass times are 10:30 and 13:30 LT for Terra and Aqua, and 14:00 for FY-3D. There are only 30 minutes between the Aqua and FY-3D overpass times. How did you conclude that "However, the use of FY-3D, which captures data at 14:00, was highly effective in capturing such events"? Please provide more details on how this conclusion was reached.
Response: We revised the manuscript according to the reviewer's comment. Please refer to Line 444.

"Additionally, emission estimates during the periods by FINN, GFED, and GFAS were generated using data from the Terra and Aqua satellites, which captured data at 10:30 and 13:30 LT. However, the use of FY-3D, which captures data at 14:00, proved highly effective in capturing such events. Furthermore, fire incidents tend to peak in the afternoon (Mehmood et al., 2022b), with agricultural waste and crop residue burning more frequently occurring during this period due to higher temperatures that enhance burning efficiency (Jurdao et al., 2012).".

We newly added the following references according to the reviewer's comment.

Akagi, S. K., Yokelson, R. J., Wiedinmyer, C., Alvarado, M. J., Reid, J. S., Karl, T., Crounse, J. D., and Wennberg, P. O.: Emission factors for open and domestic biomass burning for use in atmospheric models, Atmospheric Chemistry and Physics, 11, 4039–4072, https://doi.org/10.5194/acp-11-4039-2011, 2011.

Anon: A review of biomass burning: Emissions and impacts on air quality, health and climate in China, Science of The Total Environment, 579, 1000–1034, https://doi.org/10.1016/j.scitotenv.2016.11.025, 2017.

Burke, M., Driscoll, A., Heft-Neal, S., Xue, J., Burney, J., and Wara, M.: The changing risk and burden of wildfire in the 
[revised manuscript text omitted]